# DeepEthogram, a machine learning pipeline for supervised behavior classification from raw pixels

**James P Bohnslav[1], Nivanthika K Wimalasena[1,2], Kelsey J Clausing[3,4], Yu Y Dai[3,4], David A Yarmolinsky[1,2], Tomás Cruz[5], Adam D Kashlan[1,2], M Eugenia Chiappe[5], Lauren L Orefice[3,4], Clifford J Woolf[1,2], Christopher D Harvey[1]***

[1]Department of Neurobiology, Harvard Medical School, Boston, United States; [2]F.M. Kirby Neurobiology Center, Boston Children's Hospital, Boston, United States; [3]Department of Molecular Biology, Massachusetts General Hospital, Boston, United States; [4]Department of Genetics, Harvard Medical School, Boston, United States; [5]Champalimaud Neuroscience Programme, Champalimaud Center for the Unknown, Lisbon, Portugal

**Abstract** Videos of animal behavior are used to quantify researcher-defined behaviors of interest to study neural function, gene mutations, and pharmacological therapies. Behaviors of interest are often scored manually, which is time-consuming, limited to few behaviors, and variable across researchers. We created DeepEthogram: software that uses supervised machine learning to convert raw video pixels into an ethogram, the behaviors of interest present in each video frame. DeepEthogram is designed to be general-purpose and applicable across species, behaviors, and video-recording hardware. It uses convolutional neural networks to compute motion, extract features from motion and images, and classify features into behaviors. Behaviors are classified with above 90% accuracy on single frames in videos of mice and flies, matching expert-level human performance. DeepEthogram accurately predicts rare behaviors, requires little training data, and generalizes across subjects. A graphical interface allows beginning-to-end analysis without end-user programming. DeepEthogram's rapid, automatic, and reproducible labeling of researcher-defined behaviors of interest may accelerate and enhance supervised behavior analysis. Code is available at: https://github.com/jbohnslav/deepethogram.

*For correspondence:
harvey@hms.harvard.edu

Competing interest: The authors declare that no competing interests exist.

## Introduction

The analysis of animal behavior is a common approach in a wide range of biomedical research fields, including basic neuroscience research (*Krakauer et al., 2017*), translational analysis of disease models, and development of therapeutics. For example, researchers study behavioral patterns of animals to investigate the effect of a gene mutation, understand the efficacy of potential pharmacological therapies, or uncover the neural underpinnings of behavior. In some cases, behavioral tests allow quantification of behavior through tracking an animal's location in space, such as in the three-chamber assay, open-field arena, Morris water maze, and elevated plus maze (EPM) (*Pennington, 2019*). Increasingly, researchers are finding that important details of behavior involve subtle actions that are hard to quantify, such as changes in the prevalence of grooming in models of anxiety (*Peça et al., 2011*), licking a limb in models of pain (*Browne, 2017*), and manipulation of food objects for fine sensorimotor control (*Neubarth, 2020*; *Sauerbrei et al., 2020*). In these cases, researchers often closely observe videos of animals and then develop a list of behaviors they want to measure. To quantify these observations, the most commonly used approach, to our knowledge, is for researchers to manually watch videos

with a stopwatch to count the time each behavior of interest is exhibited (*Figure 1A*). This approach takes immense amounts of researcher time, often equal to or greater than the duration of the video per individual subject. Also, because this approach requires manual viewing, often only one or a small number of behaviors are studied at a time. In addition, researchers often do not label the video frames when specific behaviors occur, precluding subsequent analysis and review of behavior bouts, such as bout durations and the transition probability between behaviors. Furthermore, scoring of behaviors can vary greatly between researchers especially as new researchers are trained (*Segalin, 2020*) and can be subject to bias. Therefore, it would be a significant advance if a researcher could define a list of behaviors of interest, such as face grooming, tail grooming, limb licking, locomoting, rearing, and so on, and then use automated software to identify when and how frequently each of the behaviors occurred in a video.

Researchers are increasingly turning to computational approaches to quantify and analyze animal behavior (*Datta et al., 2019*; *Anderson and Perona, 2014*; *Gomez-Marin et al., 2014*; *Brown and de Bivort, 2017*; *Egnor and Branson, 2016*). The task of automatically classifying an animal's actions into user-defined behaviors falls in the category of supervised machine learning. In computer vision, this task is called 'action detection,' 'temporal action localization,' 'action recognition,' or 'action segmentation.' This task is distinct from other emerging behavioral analysis methods based on unsupervised learning, in which machine learning models discover behavioral modules from the data, irrespective of researcher labels. Although unsupervised methods, such as Motion Sequencing (*Wiltschko, 2015*; *Wiltschko et al., 2020*), MotionMapper (*Berman et al., 2014*), BehaveNet (*Batty, 2019*), B-SOiD (*Hsu and Yttri, 2019*), and others (*Datta et al., 2019*), can discover behavioral modules not obvious to the researcher, their outputs are not designed to perfectly match up to behaviors of interest in cases in which researchers have strong prior knowledge about the specific behaviors relevant to their experiments.

Pioneering work, including JAABA (*Kabra et al., 2013*), SimBA (*Nilsson, 2020*), MARS (*Segalin, 2020*), Live Mouse Tracker (*de Chaumont et al., 2019*), and others (*Segalin, 2020*; *Dankert et al., 2009*; *Sturman et al., 2020*), has made important progress toward the goal of supervised classification of behaviors. These methods track specific features of an animal's body and use the time series of these features to classify whether a behavior is present at a given timepoint. In computer vision, this is known as 'skeleton-based action detection.' In JAABA, ellipses are fit to the outline of an animal's body, and these ellipses are used to classify behaviors. SimBA classifies behaviors based on the positions of 'keypoints' on the animal's body, such as limb joints. MARS takes a similar approach with a focus on social behaviors (*Segalin, 2020*). These approaches have become easier with recent pose estimation methods, including DeepLabCut (*Mathis, 2018*; *Nath, 2019*; *Lauer, 2021*) and similar algorithms (*Pereira, 2018a*; *Graving et al., 2019*). Thus, these approaches utilize a pipeline with two major steps. First, researchers reduce a video to a set of user-defined features of interest (e.g., limb positions) using pose estimation software. Second, these pose estimates are used as inputs to classifiers that identify the behaviors of interest. This approach has the advantage that it provides information beyond whether a behavior of interest is present or absent at each timepoint. Because the parts of the animal's body that contribute to the behavior are tracked, detailed analyses of movement and how these movements contribute to behaviors of interest can be performed.

Here, we took a different approach based on models that classify behaviors directly from the raw pixel values of videos. Drawing from extensive work in this area in computer vision (*He et al., 2015*; *Piergiovanni and Ryoo, 2018*; *Zhu et al., 2017*; *Simonyan and Zisserman, 2014*), this approach has the potential to simplify the pipeline for classifying behaviors. It requires only one type of human annotation – labels for behaviors of interest – instead of labels for both pose keypoints and behaviors. In addition, this approach requires only a single model for behavior classification instead of models for pose estimation and behavior classification. Classification from raw pixels is in principle generally applicable to any dataset that has video frames and training data for the model in the form of frame-by-frame binary behavior labels. Some recent work has performed behavior classification from pixels but only focused on motor deficits (*Ryait, 2019*) or one species and setup (*van Dam et al., 2020*). Other recent work uses image and motion features, similar to the approaches we developed here, except with a focus on classifying the timepoint at which a behavior starts, instead of classifying every frame into one or more behaviors (*Kwak et al., 2019*).

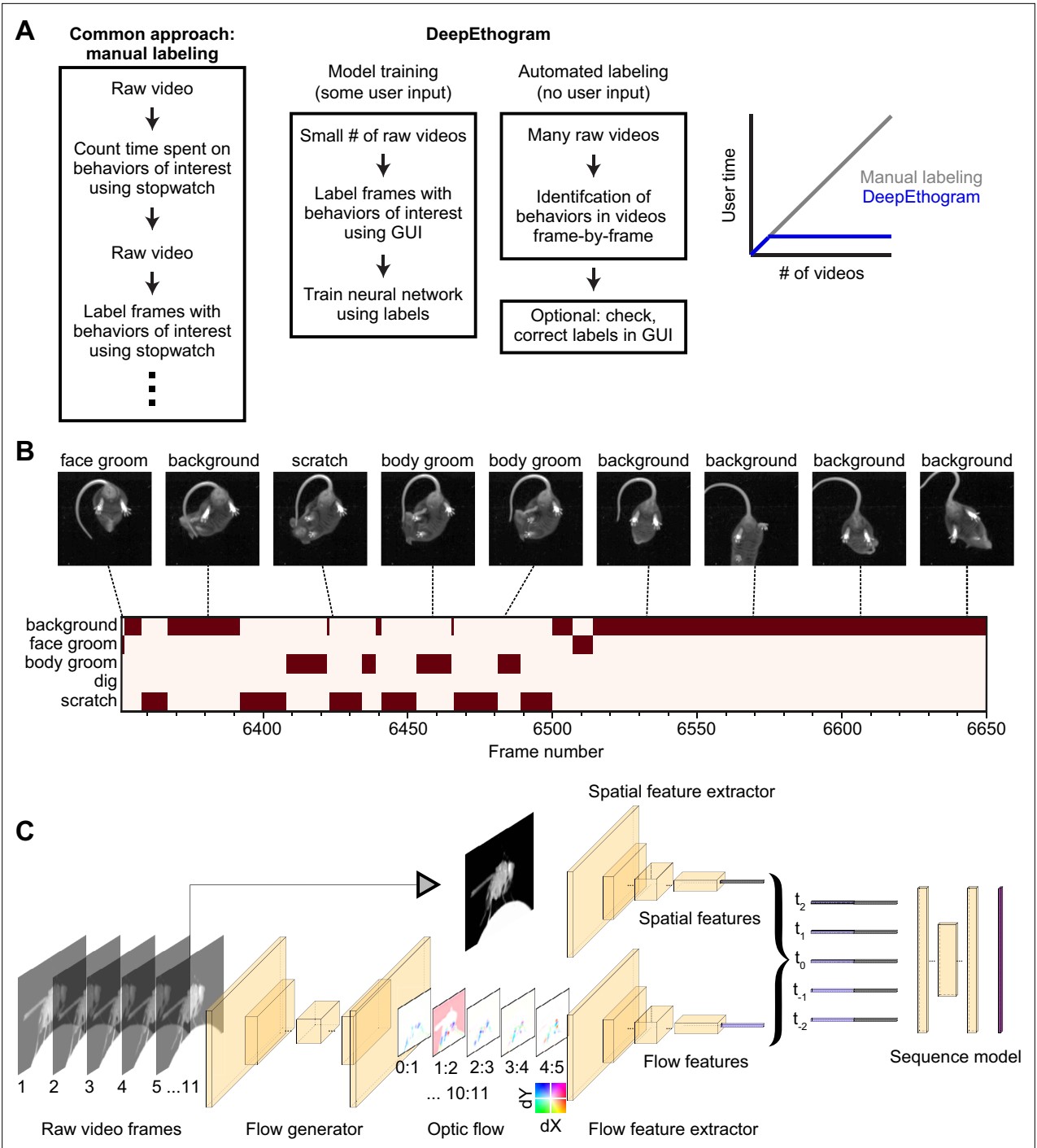

**Figure 1.** DeepEthogram overview. (**A**) Workflows for supervised behavior labeling. Left: a common traditional approach based on manual labeling. Middle: workflow with DeepEthogram. Right: Schematic of expected scaling of user time for each workflow. (**B**) Ethogram schematic. Top: example images from Mouse-Ventral1 dataset. Bottom: ethogram with human labels. Dark colors indicate which behavior is present. Example shown is from Mouse-Ventral1 dataset. Images have been cropped, brightened, and converted to grayscale for clarity. (**C**) DeepEthogram-fast model schematic. Example images are from the Fly dataset. Left: a sequence of 11 frames is converted into 10 optic flows. Middle: the center frame and the stack of 10 optic flows are converted into 512-dimensional representations via deep convolutional neural networks (CNNs). Right: these features are converted into probabilities of each behavior via the sequence model.

The online version of this article includes the following figure supplement(s) for figure 1:

**Figure supplement 1.** Optic flow.

Our method, called DeepEthogram, is a modular pipeline for automatically classifying each frame of a video into a set of user-defined behaviors. It uses a supervised deep-learning model that, with minimal user-based training data, takes a video with $T$ frames and a user-defined set of $K$ behaviors and generates a binary $[T, K]$ matrix (*Figure 1A*; *Piergiovanni and Ryoo, 2018*; *Zhu et al., 2017*). This matrix indicates whether each behavior is present or absent on each frame, which we term an 'etho-gram': the set of behaviors that are present at a given timepoint (*Figure 1B*). We use convolutional neural networks (CNNs), specifically Hidden Two-Stream Networks (*Zhu et al., 2017*) and Temporal Gaussian Mixture (TGM) networks (*Piergiovanni and Ryoo, 2018*), to detect actions in videos, and we pretrained the networks on large open-source datasets (*Deng, 2008*; *Carreira et al., 2019*). Previous work has introduced the methods we use here (*He et al., 2015*; *Piergiovanni and Ryoo, 2018*; *Zhu et al., 2017*; *Simonyan and Zisserman, 2014*), and we have adapted and extended these methods for application to biomedical research of animal behavior. We validated our approach's performance on nine datasets from two species, with each dataset posing distinct challenges for behavior classifi-cation. DeepEthogram automatically classifies behaviors with high performance, often reaching levels obtained by expert human labelers. High performance is achieved with only a few minutes of positive example data and even when the behaviors occur at different locations in the behavioral arena and at distinct orientations of the animal. Importantly, specialized video recording hardware is not required, and the entire pipeline requires no programming by the end-user because we developed a graphical user interface (GUI) for annotating videos, training models, and generating predictions.

## Results

### Modeling approach

Our goal was to take a set of video frames as input and predict the probability that each behavior of interest occurs on a given frame. This task of automated behavior labeling presents several chal-lenges that framed our solution. First, in many cases, the behavior of interest occurs in a relatively small number of video frames, and the accuracy must be judged based on correct identification of these low-frequency events. For example, if a behavior of interest is present in 5% of frames, an algorithm could guess that the behavior is 'not present' on every frame and still achieve 95% overall accuracy. Critically, however, it would achieve 0% accuracy on the frames that matter, and an algo-rithm does not know a priori which frames matter. Second, ideally a method should perform well after being trained on only small amounts of user-labeled video frames, including across different animals, and thus require little manual input. Third, a method should be able to identify the same behavior regardless of the position and orientation of the animal when the behavior occurs. Fourth, methods should require relatively low computational resources in case researchers do not have access to large compute clusters or top-level GPUs.

We modeled our approach after temporal action localization methods used in computer vision aimed to solve related problems (*Zeng, 2019*; *Xie et al., 2019*; *Chao, 2018*; *El-Nouby and Taylor, 2018*). The overall architecture of our solution includes (1) estimating motion (optic flow) from a small snippet of video frames, (2) compressing a snippet of optic flow and individual still images into a lower dimensional set of features, and (3) using a sequence of the compressed features to estimate the probability of each behavior at each frame in a video (*Figure 1C*). We implemented this architecture using large, deep CNNs. First, one CNN is used to generate optic flow from a set of images. We incorporate optic flow because some behaviors are only obvious by looking at the animal's movements between frames, such as distinguishing standing still and walking. We call this CNN the *flow generator* (*Figure 1C*, *Figure 1—figure supplement 1*). We then use the optic flow output of the flow generator as input to a second CNN to compress the large number of optic flow snippets across all the pixels into a small set of features called *flow features* (*Figure 1C*). Separately, we use a distinct CNN, which takes single video frames as input, to compress the large number of raw pixels into a small set of *spatial features*, which contain information about the values of pixels relative to one another spatially but lack temporal information (*Figure 1C*). We include single frames separately because some behaviors are obvious from a single still image, such as identifying licking just by seeing an extended tongue. Together, we call these latter two CNNs *feature extractors* because they compress the large number of raw pixels into a small set of features called a *feature vector* (*Figure 1C*). Each of these feature extractors is trained to produce a

probability for each behavior on each frame based only on their input (optic flow or single frames). We then fuse the outputs of the two feature extractors by averaging (Materials and methods). To produce the final probabilities that each behavior was present on a given frame – a step called inference – we use a *sequence model*, which has a large temporal receptive field and thus utilizes long timescale information (*Figure 1C*). We use this sequence model because our CNNs only 'look at' either 1 frame (spatial) or about 11 frames (optic flow), but when labeling videos, humans know that sometimes the information present seconds ago can be useful for estimating the behavior of the current frame. The final output of DeepEthogram is a $[T, K]$ matrix, in which each element is the probability of behavior $k$ occurring at time $t$. We threshold these probabilities to get a binary prediction for each behavior at each timepoint, with the possibility that multiple behaviors can occur simultaneously (*Figure 1B*).

For the flow generator, we use the MotionNet (*Zhu et al., 2017*) architecture to generate 10 optic flow frames from 11 images. For the feature extractors, we use the ResNet family of models (*He et al., 2015*; *Hara et al., 2018*) to extract both flow features and spatial features. Finally, we use TGM (*Piergiovanni and Ryoo, 2018*) models as the sequence model to perform the ultimate classification. Each of these models has many variants with a large range in the number of parameters and the associated computational demands. We therefore created three versions of DeepEthogram that use variants of these models, with the aim of trading off accuracy and speed: DeepEthogram-fast, DeepEthogram-medium, and DeepEthogram-slow. DeepEthogram-fast uses TinyMotionNet (*Zhu et al., 2017*) for the flow generator and ResNet18 (*He et al., 2015*) for the feature extractors. It has the fewest parameters, the fastest training of the flow generator and feature extractor models, the fastest inference time, and the smallest requirement for computational resources. As a tradeoff for this speed, DeepEthogram-fast tends to have slightly worse performance than the other versions (see below). In contrast, DeepEthogram-slow uses a novel architecture TinyMotionNet3D for its flow generator and 3D-ResNet34 (*He et al., 2015*; *Simonyan and Zisserman, 2014*; *Hara et al., 2018*) for its feature extractors. It has the most parameters, the slowest training and inference times, and the highest computational demands, but it has the capacity to produce the best performance. DeepEthogram-medium is intermediate and uses MotionNet (*Zhu et al., 2017*) and ResNet50 (*He et al., 2015*) for its flow generator and feature extractors. All versions of DeepEthogram use the same sequence model. All variants of the flow generators and feature extractors are pretrained on the Kinetics700 video dataset (*Carreira et al., 2019*), so that model parameters do not have to be learned from scratch (Materials and methods). TGM networks represent the state of the art on various action detection benchmarks as of 2019 (*Piergiovanni and Ryoo, 2018*). However, recent work based on multiple temporal resolutions (*Feichtenhofer et al., 2019*; *Kahatapitiya and Ryoo, 2021*), graph convolutional networks (*Zeng, 2019*), and transformer architectures (*Nawhal and Mori, 2021*) has exceeded this performance. We carefully chose DeepEthogram's components based on their performance, parameter count, and hardware requirements. DeepEthogram as a whole and its component parts are not aimed to be the state of the art on standard computer vision temporal action localization datasets and instead are focused on practical application to biomedical research of animal behavior.

In practice, the first step in running DeepEthogram is to train the flow generator on a set of videos, which occurs without user input (*Figure 1A*). In parallel, a user must label each frame in a set of training videos for the presence of each behavior of interest. These labels are then used to train independently the spatial feature extractor and the flow feature extractor to produce separate estimates of the probability of each behavior. The extracted feature vectors for each frame are then saved and used to train the sequence models to produce the final predicted probability of each behavior at each frame. We chose to train the models in series, rather than all at once from end-to-end, due to a combination of concerns about backpropagating error across diverse models, overfitting with extremely large models, and computational capacity (Materials and methods).

## Diverse datasets to test DeepEthogram

To test the performance of our model, we used nine different neuroscience research datasets that span two species and present distinct challenges for computer vision approaches. Please see the examples in *Figure 2*, *Figure 2—figure supplements 1–6*, and *Videos 1–9* that demonstrate the behaviors of interest and provide an intuition for the ease or difficulty of identifying and distinguishing particular behaviors.

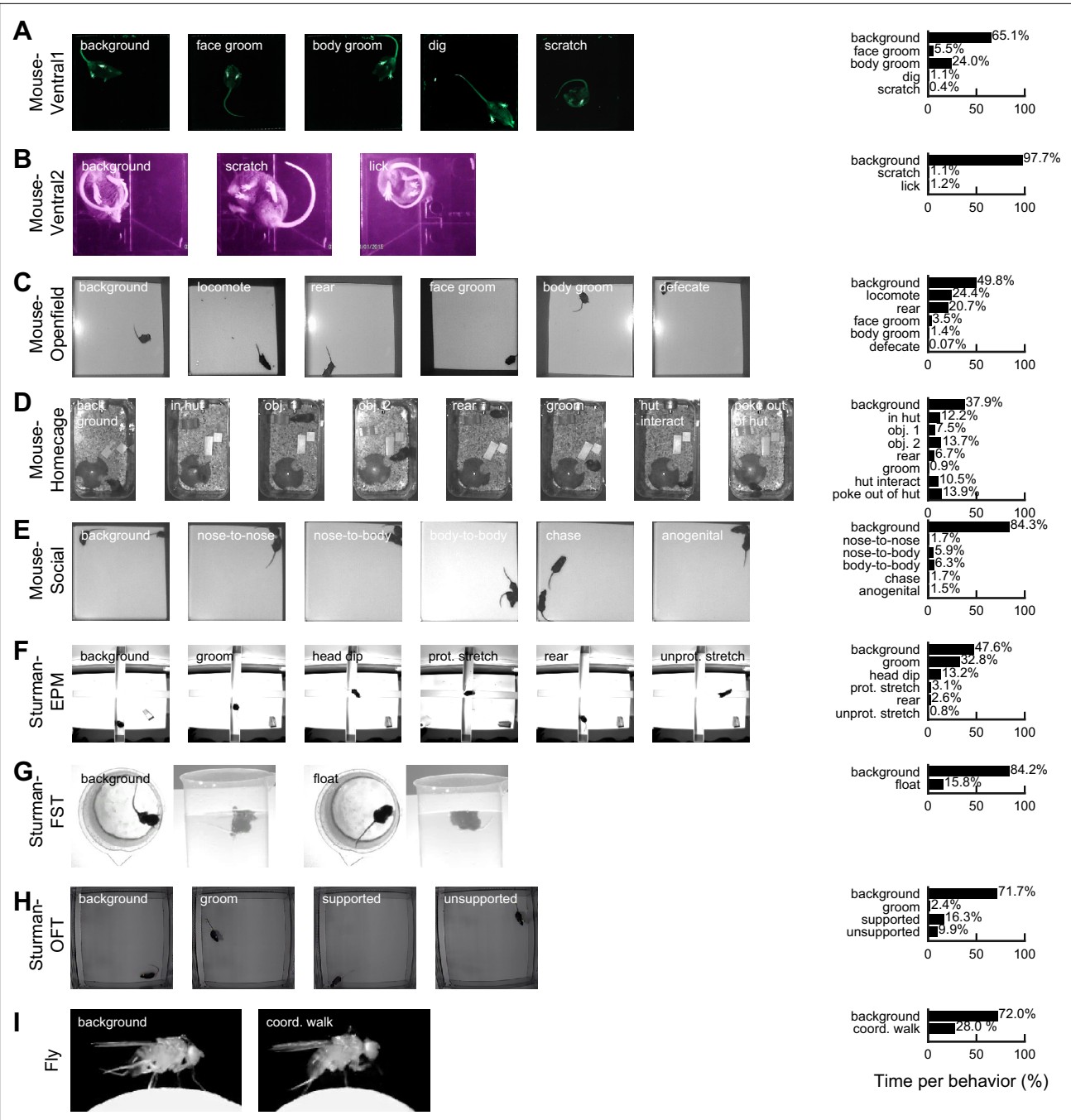

**Figure 2.** Datasets and behaviors of interest. (**A**) Left: raw example images from the Mouse-Ventral1 dataset for each of the behaviors of interest. Right: time spent on each behavior, based on human labels. Note that the times may add up to more than 100% across behaviors because multiple behaviors can occur on the same frame. Background is defined as when no other behaviors occur. (**B–I**) Similar to (**A**), except for the other datasets.

The online version of this article includes the following figure supplement(s) for figure 2:

**Figure supplement 1.** Example images from the datasets, part 1.

**Figure supplement 2.** Example images from the datasets, part 2.

**Figure supplement 3.** Example images from the datasets, part 3.

**Figure supplement 4.** Example images from the datasets, part 4.

**Figure supplement 5.** Example images from the datasets, part 5.

**Figure supplement 6.** Example images from the datasets, part 6.

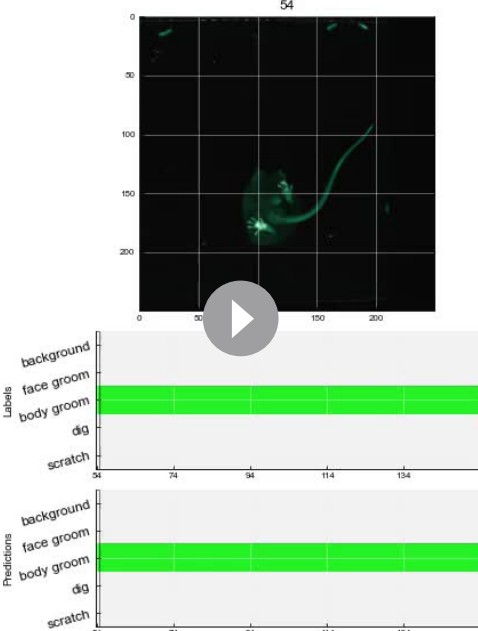 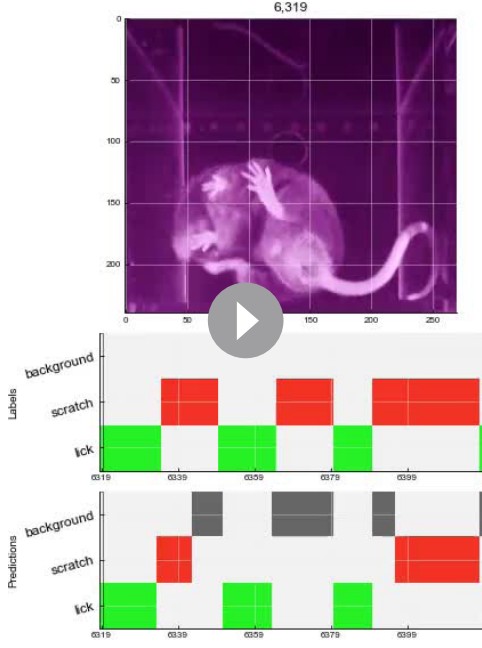

**Video 1.** DeepEthogram example from the Mouse-Ventral1 dataset. Video is from the test set. Top: raw image. Title indicates frame number in video. Tick legends indicate pixels. Middle: human labels. Black box indicates the current frame. Bottom: DeepEthogram predictions from a trained DeepEthogram-medium model.

https://elifesciences.org/articles/63377/figures#video1

**Video 2.** DeepEthogram example from the Mouse-Ventral2 dataset. Video is from the test set. Top: raw image. Title indicates frame number in video. Tick legends indicate pixels. Middle: human labels. Black box indicates the current frame. Bottom: DeepEthogram predictions from a trained DeepEthogram-medium model.

https://elifesciences.org/articles/63377/figures#video2

We collected five datasets of mice in various behavioral arenas. The 'Mouse-Ventral1' and 'Mouse-Ventral2' datasets are bottom-up videos of a mouse in an open field and small chamber, respectively (*Figure 2A,B*, *Figure 2—figure supplement 1A, B*, *Videos 1–2*). The 'Mouse-Openfield' dataset includes commonly used top-down videos of a mouse in an open arena (*Figure 2C*, *Figure 2—figure supplement 2A*, *Video 3*). The 'Mouse-Homecage' dataset are top-down videos of a mouse in its home cage with bedding, a hut, and two objects (*Figure 2D*, *Figure 2—figure supplement 3*, *Video 4*). The 'Mouse-Social' dataset are top-down videos of two mice interacting in an open arena (*Figure 2E*, *Figure 2—figure supplement 4*, *Video 5*). We also tested three datasets from published work by *Sturman et al., 2020* that consist of mice in common behavior assays: the EPM, forced swim test (FST), and open field test (OFT) (*Figure 2F–H*, *Figure 2—figure supplements 5–6*, *Videos 6–8*). Finally, we tested a different species in the 'Fly' dataset that includes side view videos of a *Drosophila melanogaster* and aims to identify a coordinated walking pattern (*Fujiwara et al., 2017*; *Figure 2D*, *Figure 2—figure supplement 2B*, *Video 9*).

Collectively, these datasets include distinct view angles, a variety of illumination levels, and different resolutions and video qualities. They also pose various challenges for computer vision, including the subject occupying a small fraction of pixels (Mouse-Ventral1, Mouse-Openfield, Mouse-Homecage, Sturman-EPM, Sturman-OFT), complex backgrounds with non-uniform patterns (bedding and objects in Mouse-Homecage) or irrelevant motion (moving water in Sturman-FST), objects that occlude the subject (Mouse-Homecage), poor contrast of body parts (Mouse-Openfield, Sturman-EPM, Sturman-OFT), little motion from frame-to-frame (Fly, due to high video rate), and few training examples (Sturman-EPM, only five videos and only three that contain all behaviors). Furthermore, in most datasets, some behaviors of interest are rare and occur in only a few percent of the total video frames.

In each dataset, we labeled a behavior as present regardless of the location where it occurred and the orientation of the subject when it occurred. We did not note position or direction information,

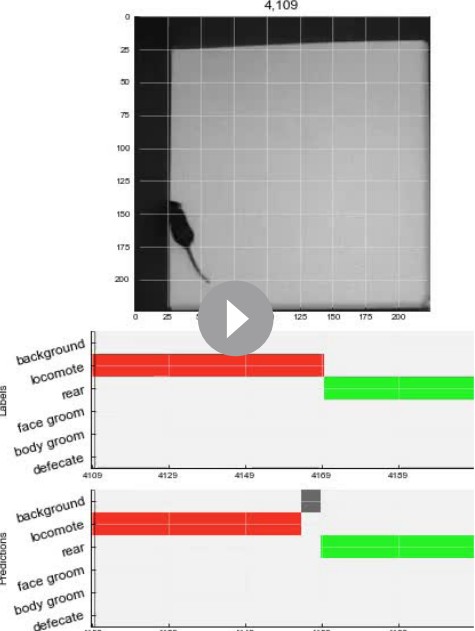

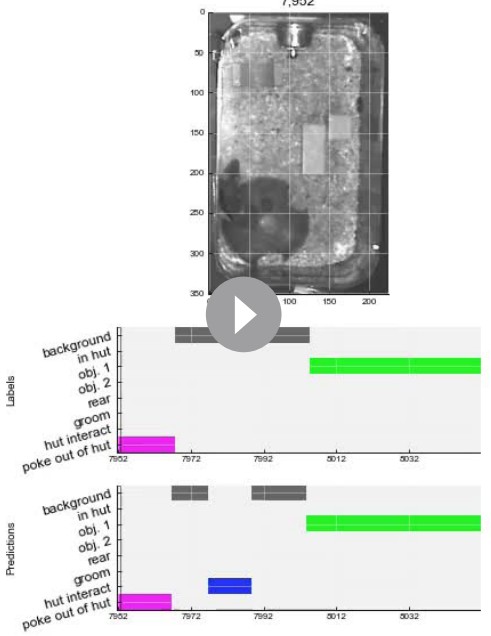

**Video 3.** DeepEthogram example from the Mouse-Openfield dataset. Video is from the test set. Top: raw image. Title indicates frame number in video. Tick legends indicate pixels. Middle: human labels. Black box indicates the current frame. Bottom: DeepEthogram predictions from a trained DeepEthogram-medium model.

https://elifesciences.org/articles/63377/figures#video3

**Video 4.** DeepEthogram example from the Mouse-Homecage dataset. Video is from the test set. Top: raw image. Title indicates frame number in video. Tick legends indicate pixels. Middle: human labels. Black box indicates the current frame. Bottom: DeepEthogram predictions from a trained DeepEthogram-medium model.

https://elifesciences.org/articles/63377/figures#video4

and we did not spatially crop the video frames or align the animal before training our model. In all datasets, we labeled the frames on which none of the behaviors of interest were present as 'background,' following the convention in computer vision. Each video in a dataset was recorded using a different individual mouse or fly, and thus training and testing the model across videos measured generalization across individual subjects. The video datasets and researcher annotations are available at the project website: https://github.com/jbohnslav/deepethogram (copy archived at swh:1:rev:ffd7e6bd91f52c7d1dbb166d1fe8793a26c4cb01), *Bohnslav, 2021*.

## DeepEthogram achieves high performance approaching expert-level human performance

We split each dataset into three subsets: training, validation, and test (Materials and methods). The training set was used to update model parameters, such as the weights of the CNNs. The validation set was used to set appropriate hyperparameters, such as the thresholds used to turn the probabilities of each behavior into binary predictions about whether each behavior was present. The test set was used to report performance on new data not used in training the model. We generated five random splits of the data into training, validation, and test sets and averaged our results across these five splits, unless noted otherwise (Materials and methods). We computed three complementary metrics of model performance using the test set. First, we computed the accuracy, which is the fraction of elements of the $[T, K]$ ethogram that were predicted correctly. We note that in theory accuracy could be high even if the model did not perform well on each behavior. For example, in the Mouse-Ventral2 dataset, some behaviors were incredibly rare, occurring in only ~2% of frames (*Figure 2B*). Thus, the model could in theory achieve ~98% accuracy simply by guessing that the behavior was absent on all frames. Therefore, we also computed the F1 score, a metric ranging from 0 (bad) to 1 (perfect) that takes into account the rates of false positives and false negatives. The F1 score is the geometric mean of the precision and recall of the model. Precision is the fraction of frames labeled by the model as

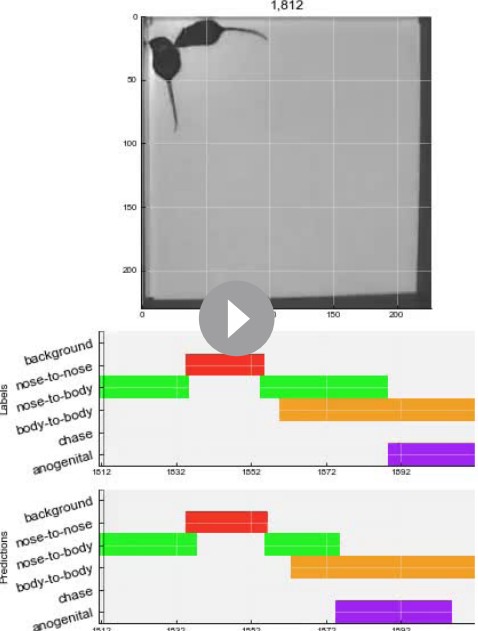

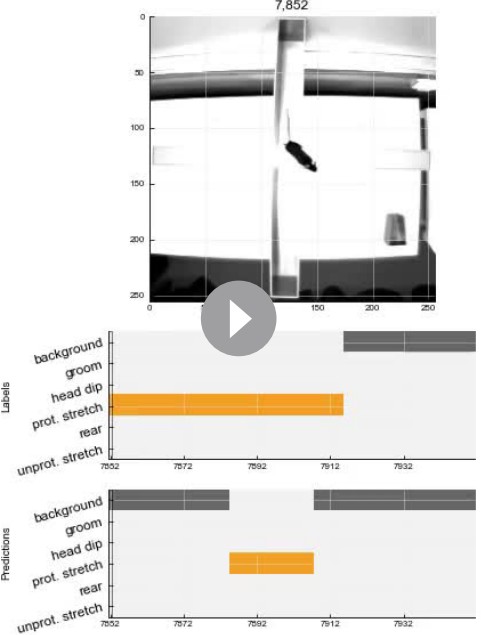

**Video 5.** DeepEthogram example from the Mouse-Social dataset. Video is from the test set. Top: raw image. Title indicates frame number in video. Tick legends indicate pixels. Middle: human labels. Black box indicates the current frame. Bottom: DeepEthogram predictions from a trained DeepEthogram-medium model.

https://elifesciences.org/articles/63377/figures#video5

**Video 6.** DeepEthogram example from the Sturman-EPM dataset. Video is from the test set. Top: raw image. Title indicates frame number in video. Tick legends indicate pixels. Middle: human labels. Black box indicates the current frame. Bottom: DeepEthogram predictions from a trained DeepEthogram-medium model.

https://elifesciences.org/articles/63377/figures#video6

a given behavior that are actually that behavior (true positives/(true positives + false positives)). Recall is the fraction of frames actually having a given behavior that are correctly labeled as that behavior by the model (true positives/(true positives + false negatives)). We report the F1 score in the main figures and show precision and recall performance in the figure supplements. Because the accuracy and F1 score depend on our choice of a threshold to turn the probability of a given behavior on a given frame into a binary prediction about the presence of that behavior, we also computed the area under the receiver operating curve (AUROC), which summarizes performance as a function of the threshold.

We first considered the entire ethogram, including all behaviors. DeepEthogram performed with greater than 85% accuracy on the test data for all datasets (*Figure 3A*). Overall metrics were calculated for each element of the ethogram. The model achieved high overall F1 scores, with high precision and recall (*Figure 3B*, *Figure 3—figure supplement 1A*, *Figure 3—figure supplement 2A*). Similarly, high overall performance was observed with the AUROC measures (*Figure 3—figure supplement 3A*). These results indicate that the model was able to capture the overall patterns of behavior in videos.

We also analyzed the model's performance for each individual behavior. The model achieved F1 scores of 0.7 or higher for many behaviors, even reaching F1 scores above 0.9 in some cases (*Figure 3C–K*). DeepEthogram's performance significantly exceeded chance levels of performance on nearly all behaviors across datasets (*Figure 3C–K*). Given that F1 scores may not be intuitive to understand in terms of their values, we examined individual snippets of videos with a range of F1 scores and found that F1 scores similar to the means for our datasets were consistent with overall accurate predictions (*Figure 3P*, *Figure 3—figure supplements 4–11*). We note that the F1 score is a demanding metric, and even occasional differences on single frames or a small number of frames can substantially decrease the F1 score. Relatedly, the model achieved high precision, recall, and AUROC

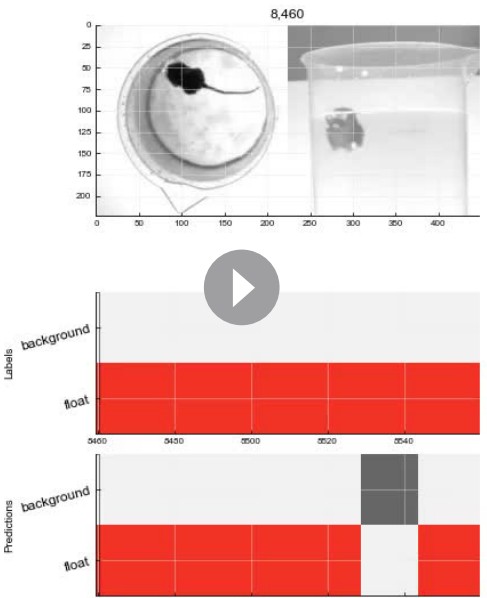

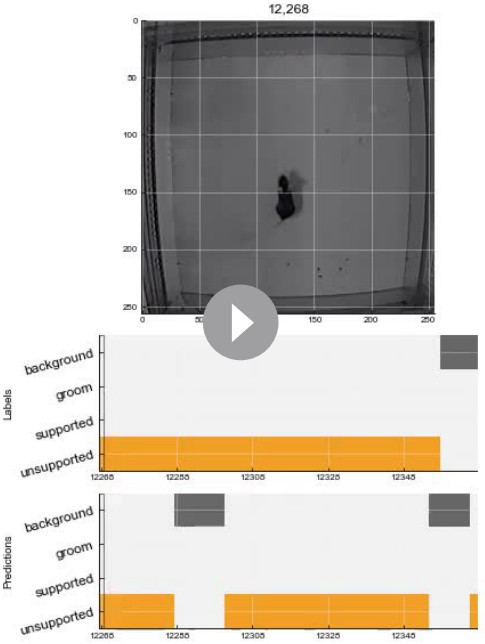

**Video 7.** DeepEthogram example from the Sturman-FST dataset. Video is from the test set. Top: raw image. Title indicates frame number in video. Tick legends indicate pixels. Middle: human labels. Black box indicates the current frame. Bottom: DeepEthogram predictions from a trained DeepEthogram-medium model.

https://elifesciences.org/articles/63377/figures#video7

**Video 8.** DeepEthogram example from the Sturman-OFT dataset. Video is from the test set. Top: raw image. Title indicates frame number in video. Tick legends indicate pixels. Middle: human labels. Black box indicates the current frame. Bottom: DeepEthogram predictions from a trained DeepEthogram-medium model.

https://elifesciences.org/articles/63377/figures#video8

values for individual behaviors (*Figure 3—figure supplement 1B–J*, *Figure 3—figure supplement 2B–J*, *Figure 3—figure supplement 3B–J*). The performance of the model depended on the frequency with which a behavior occurred (c.f. *Figure 2* right panels and *Figure 3C–J*). Strikingly, however, performance was relatively high even for behaviors that occurred rarely, that is, in less than 10% of video frames (*Figure 3M*, *Figure 3—figure supplement 1L*, *Figure 3—figure supplement 2L*, and *Figure 3—figure supplement 3K*). The performance tended to be highest for DeepEthogram-slow and worst for DeepEthogram-fast, but the differences between model versions were generally small and varied across behaviors (*Figure 3A,B*, *Figure 3—figure supplement 1A*, *Figure 3—figure supplement 2A*, *Figure 3—figure supplement 3A*). The high-performance values are, in our opinion, impressive given that they were calculated based on single-frame predictions for each behavior, and thus performance will be reduced if the model misses the onset or offset of a behavior bout by even a single frame. These high values suggest that the model not only correctly predicted which behaviors happened and when but also had the resolution to correctly predict the onset and offset of bouts.

To better understand the performance of DeepEthogram, we benchmarked the model by comparing its performance to the degree of agreement between expert human labelers. Multiple researchers with extensive experience in monitoring and analyzing mouse behavior videos independently labeled the same set of videos for the Mouse-Ventral1, Mouse-Ventral2, Mouse-Openfield, Mouse-Social, and Mouse-Homecage datasets, allowing us to measure the consistency across human experts. Also, Sturman et al. released the labels of each of three expert human labelers (*Sturman et al., 2020*). The Fly dataset has more than 3 million frames and thus was too large to label multiple times. Human-human performance was calculated by defining one labeler as the 'ground truth' and the other labelers as 'predictions' and then computing the same performance metrics as for DeepEthogram. In this way, 'human accuracy' is the same as the percentage of scores on which two humans agreed. Strikingly, the overall accuracy, F1 scores, precision, and recall for DeepEthogram approached that of expert

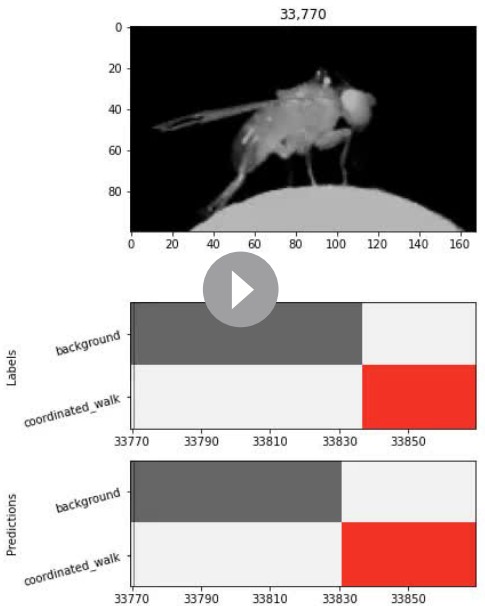

**Video 9.** DeepEthogram example from the Flies dataset. Video is from the test set. Top: raw image. Title indicates frame number in video. Tick legends indicate pixels. Middle: human labels. Black box indicates the current frame. Bottom: DeepEthogram predictions from a trained DeepEthogram-medium model.

https://elifesciences.org/articles/63377/figures#video9

human labelers (*Figure 3A, B, C, E, H, I, J and L*, *Figure 3—figure supplement 1A,B,D,G,H,I,K*, *Figure 3—figure supplement 2*). In many cases, DeepEthogram's performance was statistically indistinguishable from human-level performance, and in the cases in which humans performed better, the difference in performance was generally small. Notably, the behaviors for which DeepEthogram had the lowest performance tended to be the behaviors for which humans had less agreement (lower human-human F1 score) (*Figure 3L*, *Figure 3—figure supplement 1K*, *Figure 3—figure supplement 2K*). Relatedly, DeepEthogram performed best on the frames in which the human labelers agreed and did more poorly in the frames in which humans disagreed (*Figure 3N and O*, *Figure 3—figure supplement 1M*, *Figure 3—figure supplement 2M*). Thus, there is a strong correlation between DeepEthogram and human performance, and the values for DeepEthogram's performance approach those of expert human labelers.

The behavior with the worst model performance was 'defecate' from the Mouse-Openfield dataset (*Figures 2C and 3E*). Notably, defecation was incredibly rare, occurring in only 0.1% of frames. Furthermore, the act of defecation was not actually visible from the videos. Rather, human labelers marked the 'defecate' behavior when new fecal matter appeared, which involved knowledge of the foreground and background, tracking objects, and inferring unseen behavior. This type of behavior is expected to be challenging for DeepEthogram because the model is based on images and local motion and thus will fail when the behavior cannot be directly observed visually.

The model was able to accurately predict the presence of a behavior even when that behavior happened in different locations in the environment and with different orientations of the animal (*Figure 3—figure supplement 12*). For example, the model predicted face grooming accurately both when the mouse was in the top-left quadrant of the chamber and facing north and when the mouse was in the bottom-right quadrant facing west. This result is particularly important for many analyses of behavior that are concerned with the behavior itself, rather than where that behavior happens.

One striking feature was DeepEthogram's high performance even on rare behaviors. From our preliminary work building up to the model presented here, we found that simpler models performed well on behaviors that occurred frequently and performed poorly on the infrequent behaviors. Given that, in many datasets, the behaviors of interest are infrequent (*Figure 2*), we placed a major emphasis on performance in cases with large class imbalances, meaning when some behaviors only occurred in a small fraction of frames. In brief, we accounted for class imbalances in the initialization of the model parameters (Materials and methods). We also changed the cost function to weight errors on rare classes more heavily than errors on common classes. We used a form of regularization specific to transfer learning to reduce overfitting. Finally, we tuned the threshold for converting the model's probability of a given behavior into a classification of whether that behavior was present. Without these added features, the model simply learned to ignore rare classes. We consider these steps toward identifying rare behaviors to be of major significance for effective application in common experimental datasets.

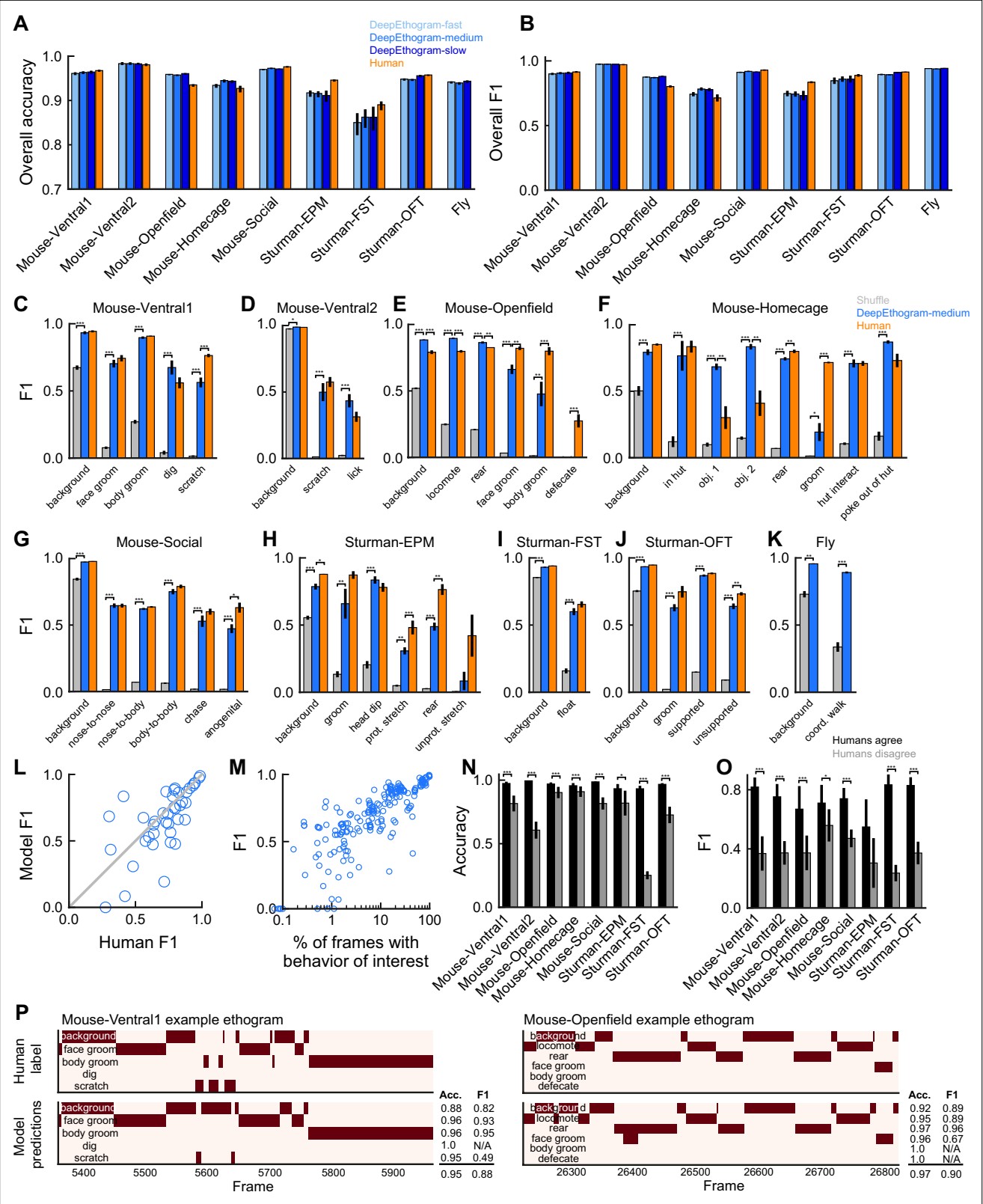

**Figure 3.** DeepEthogram performance. All results are from the test sets only. (**A**) Overall accuracy for each model size and dataset. Error bars indicate mean ± SEM across five random splits of the data (three for Sturman-EPM). (**B**) Similar to (**A**), except for overall F1 score. (**C**) F1 score for DeepEthogram-medium for individual behaviors on the Mouse-Ventral1 dataset. Gray bars indicate shuffle (Materials and methods). *p≤0.05, **p≤0.01, ***p≤0.001, repeated measures ANOVA with a post-hoc Tukey's honestly significant difference test. (**D**) Similar to (**C**), but for Mouse-Ventral2. Model and shuffle

*Figure 3 continued on next page*

*Figure 3 continued*

were compared with paired t-tests with Bonferroni correction. (**E**) Similar to (**C**), but for Mouse-Openfield. (**F**) Similar to (**D**), but for Mouse-Homecage. (**G**) Similar to (**D**), but for Mouse-Social. (**H**) Similar to (**C**), but for Sturman-EPM. (**I**) Similar to (**C**), but for Sturman-FST. (**J**) Similar to (**C**), but for Sturman-OFT. (**K**) Similar to (**D**), but for Fly dataset. (**L**) F1 score on individual behaviors (circles) for DeepEthogram-medium vs. human performance. Circles indicate the average performance across splits for behaviors in datasets with multiple human labels. Gray line: unity. Model vs. human performance: p=0.067, paired t-test. (**M**) Model F1 vs. the percent of frames in the training set with the given behavior. Each circle is one behavior for one split of the data. (**N**) Model accuracy on frames for which two human labelers agreed or disagreed. Paired t-tests with Bonferroni correction. (**O**) Similar to (**N**), but for F1. (**P**) Ethogram examples. Dark color indicates the behavior is present. Top: human labels. Bottom: DeepEthogram-medium predictions. The accuracy and F1 score for each behavior, and the overall accuracy and F1 scores are shown. Examples were chosen to be similar to the model's average by behavior.

The online version of this article includes the following figure supplement(s) for figure 3:

**Figure supplement 1.** DeepEthogram performance, precision.

**Figure supplement 2.** DeepEthogram performance, recall.

**Figure supplement 3.** DeepEthogram performance, area under the receiver operating characteristic curve (AUROC).

**Figure supplement 4.** Ethogram examples for the Mouse-Ventral1 dataset.

**Figure supplement 5.** Ethogram examples for the Mouse-Ventral2 dataset.

**Figure supplement 6.** Ethogram examples for the Mouse-Openfield dataset.

**Figure supplement 7.** Ethogram examples for the Mouse-Homecage dataset.

**Figure supplement 8.** Ethogram examples for the Mouse-Social dataset.

**Figure supplement 9.** Ethogram examples for the Sturman-EPM dataset.

**Figure supplement 10.** Ethogram examples for the Sturman-FST dataset.

**Figure supplement 11.** Ethogram examples for the Sturman-OFT dataset.

**Figure supplement 12.** DeepEthogram exhibits position and heading invariance.

## DeepEthogram accurately predicts behavior bout statistics

Because DeepEthogram produces predictions on individual frames, it allows for subsequent analyses of behavior bouts, such as the number of bouts, the duration of bouts, and the transition probability from one behavior to another. These statistics of bouts are often not available if researchers only record the overall time spent on a behavior with a stopwatch, rather than providing frame-by-frame labels. We found a strong correspondence for the statistics of behavior bouts between the predictions of DeepEthogram and those from human labels. We first focused on results at the level of individual videos for the Mouse-Ventral1 dataset, comparing the model predictions and human labels for the percent of time spent on each behavior, the number of bouts per behavior, and the mean bout duration (*Figure 4A–C*). Note that the model was trained on the labels from Human 1. For the time spent on each behavior, the model predictions and human labels were statistically indistinguishable (one-way ANOVA, p>0.05; *Figure 4A*). For the number of bouts and bout duration, the model was statistically indistinguishable from the labels of Human 1, on which it was trained. Some differences were present between the model predictions and the other human labels not used for training (*Figure 4B,C*). However, the magnitude of these differences was within the range of differences between the multiple human labelers (*Figure 4B,C*).

To summarize the performance of DeepEthogram on bout statistics for each behavior in all datasets, we averaged the time spent, number of bouts, and bout duration for each behavior across the five random splits of the data into train, validation, and test sets. This average provides a quantity similar to an average across multiple videos, and an average across multiple videos is likely how some end-users will report their results. The values from the model were highly similar to the those from the labels on which it was trained (Human 1 labels) for the time spent per behavior, the number of bouts, and the mean bout duration (*Figure 4D–F*). Together, these results show that DeepEthogram accurately predicts bout statistics that might be of interest to biologist end-users.

## DeepEthogram approaches expert-level human performance for bout statistics and transitions

Next, we benchmarked DeepEthogram's performance on bout statistics by comparing its performance to the level of agreement between expert human labelers. We started by looking at the time spent on

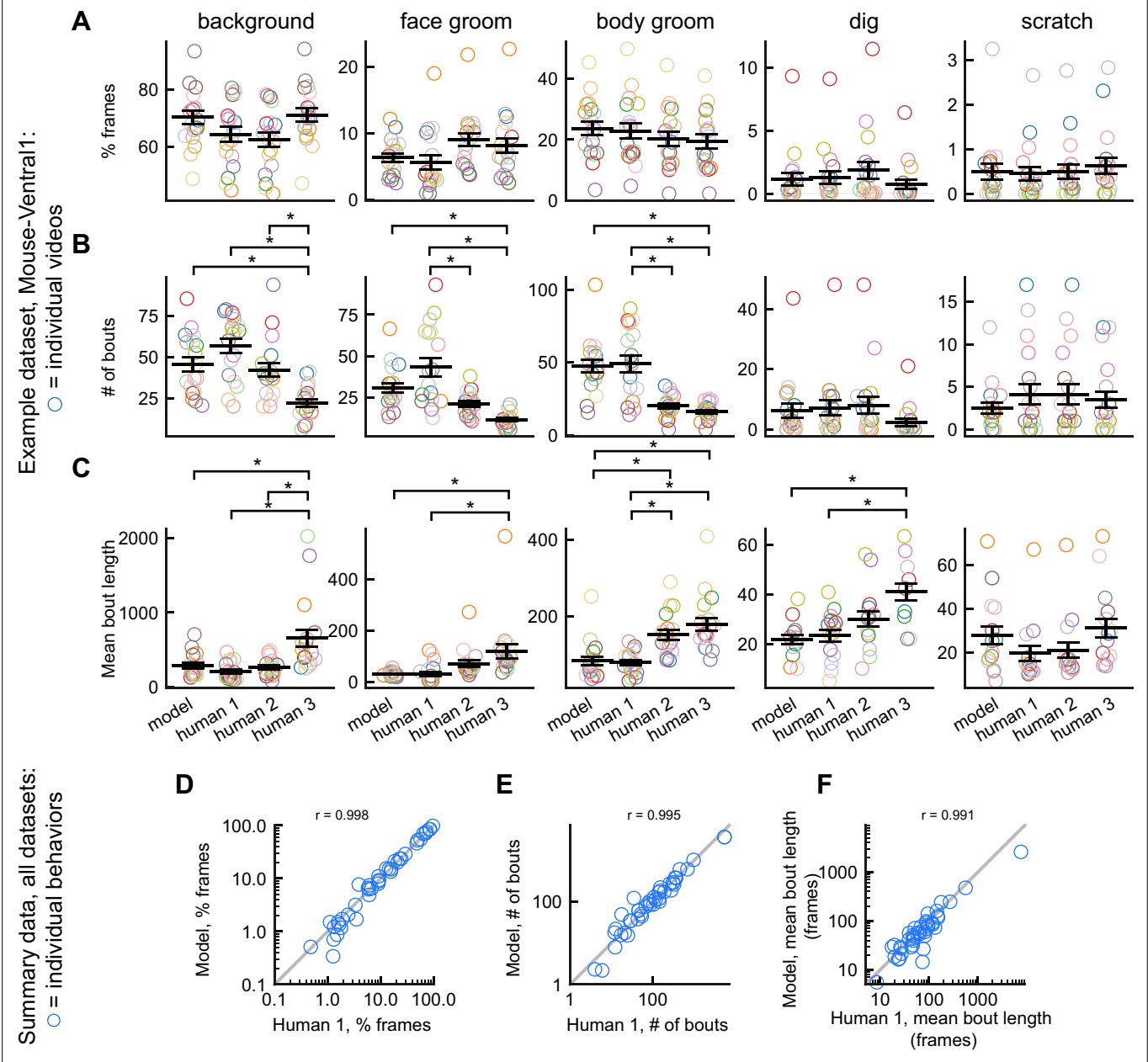

**Figure 4.** DeepEthogram performance on bout statistics. All results from DeepEthogram-medium, test set only. (**A–C**) Comparison of model predictions and human labels on individual videos from the Mouse-Ventral1 dataset. Each point is one behavior from one video. Colors indicate video ID. Error bars: mean ± SEM (n = 18 videos). Asterisks indicate p<0.05, one-way ANOVA with Tukey's multiple comparison test. No asterisk indicates p>0.05. (**D–F**) Comparison of model predictions and human labels on all behaviors for all datasets. Each circle is one behavior from one dataset, averaged across splits of the data. Gray line: unity.

each behavior in single videos for the Mouse-Ventral1 and Sturman-OFT datasets. We compared the labels from Human 1 to the model predictions and to the labels from Humans 2 and 3 (*Figure 5A,B*). In general, there was strong agreement between the model and Human 1 and among human labelers (*Figure 5A,B*, left and middle). To directly compare model performance to human-human agreement, we plotted the absolute difference between the model and Human 1 versus the absolute difference between Human 1 and Humans 2 and 3 (*Figure 5A,B*, right). Model agreement was significantly worse than human-human agreement when considering individual videos. However, the magnitude of this difference was small, implying that discrepancies in behavior labels introduced by the model were only marginally larger than the variability between multiple human labelers.

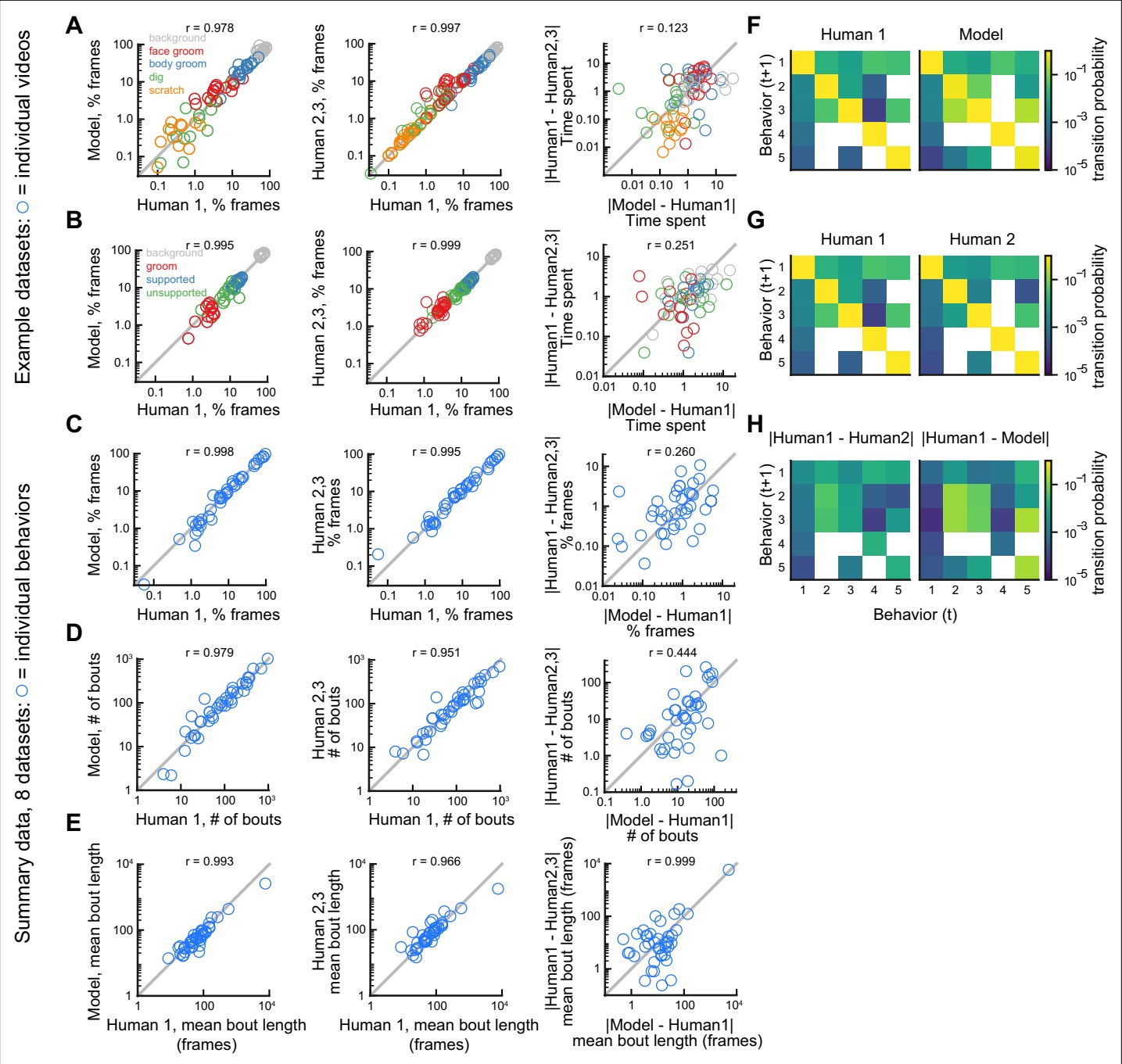

**Figure 5.** Comparison of model performance to human performance on bout statistics. All model data are from DeepEthogram-medium, test set data. r values indicate Pearson's correlation coefficient. (**A**) Performance on Mouse-Ventral1 dataset for time spent. Each circle is one behavior from one video. Left: Human 1 vs. model. Middle: Human 1 vs. Humans 2 and 3. Both Humans 2 and 3 are shown on the y-axis. Right: absolute error between Human 1 and model vs. absolute error between Human 1 and each of Humans 2 and 3. Model difference vs. human difference: p<0.001, paired t-test. (**B**) Similar to (**A**), but for Sturman-OFT dataset. Right: model difference vs. human difference: p<0.001, paired t-test. (**C–E**) Performance on all datasets with multiple human labelers (Mouse-Ventral1, Mouse-Openfield, Sturman-OFT, Sturman-EPM, Sturman-FST). Each point is one behavior from one dataset, averaged across data splits. Performance for Humans 2 and 3 were averaged. Similar to *Figure 4D–F*, but only for datasets with multiple labelers. Left: Human 1 vs. model. Middle: Human 1 vs. Humans 2 and 3. Right: absolute error between Human 1 and model vs. absolute error between Human 1 and each of Humans 2 and 3. p>0.05, paired t-test with Bonferroni correction, in (**C–E**) right panels. (**F–H**) Example transition matrices for Mouse-Ventral1 dataset. For humans and models, transition matrices were computed for each data split and averaged across splits.

The online version of this article includes the following figure supplement(s) for figure 5:

**Figure supplement 1.** Performance of keypoint-based behavior classification on the Mouse-Openfield dataset.

*Figure 5 continued on next page*

*Figure 5 continued*

**Figure supplement 2.** Comparison with unsupervised methods.

To summarize our benchmarking of model performance on bout statistics for each behavior in all datasets with multiple human labelers, we again averaged the time spent, number of bouts, and bout duration for each behavior across the five random splits of the data to obtain a quantity similar to an average across multiple videos (*Figure 5C–E*). For time spent per behavior, number of bouts, and mean bout length, the human-model differences were similar, and not significantly different, in magnitude to the differences between humans (*Figure 5C–E*, right column). The transition probabilities between behaviors were also broadly similar between Human 1, Human 2, and the model (*Figure 5F–H*). Furthermore, model-human differences and human-human differences were significantly correlated (*Figure 5C–E*, right column), again showing that DeepEthogram models are more reliable for situations in which multiple human labelers agree (see *Figure 3N–O*, *Figure 3—figure supplement 1M*, *Figure 3—figure supplement 2M*).

Therefore, the results from *Figure 5A,B* indicate that the model predictions are noisier than human-human agreement on the level of individual videos. However, when averaged across multiple videos (*Figure 5C–E*), this noise averages out and results in similar levels of variability for the model and multiple human labelers. Given that DeepEthogram performed slightly worse on F1 scores relative to expert humans but performed similarly to humans on bout statistics, it is possible that for rare behaviors DeepEthogram misses a small number of bouts, which would minimally affect bout statistics but could decrease the overall F1 score.

Together, our results from *Figures 3–5* and *Figure 3—figure supplements 1–3* indicate that DeepEthogram's predictions match well the labels defined by expert human researchers. Further, these model predictions allow easy post-hoc analysis of additional statistics of behaviors, which may be challenging to obtain with traditional manual methods.

## Comparison to existing methods based on keypoint tracking

While DeepEthogram operates directly on the raw pixel values in the videos, other methods exist that first track body keypoints and then perform behavior classification based on these keypoints (*Segalin, 2020*; *Kabra et al., 2013*; *Nilsson, 2020*; *Sturman et al., 2020*). One such approach that is appealing due to its simplicity and clarity was developed by Sturman et al. and was shown to be superior to commercially available alternatives (*Sturman et al., 2020*). In their approach, DeepLabCut (*Mathis, 2018*) is used to estimate keypoints and then a multilayer perceptron architecture is used to classify features of these keypoints into behaviors. We compared the performance of DeepEthogram and this alternate approach using our custom implementation of the Sturman et al. methods (*Figure 5—figure supplement 1*). We focused our comparison on the Mouse-Openfield dataset, which is representative of videos used in a wide range of biological studies. We used DeepLabCut (*Mathis, 2018*) to identify the position of the four paws, the base of the tail, the tip of the tail, and the nose. These keypoints could be used to distinguish behaviors. For example, the distance between the nose and the base of the tail was highest when the mouse was locomoting (*Figure 5—figure supplement 1C*). However, the accuracy and F1 scores for DeepEthogram generally exceeded those identified from classifiers based on features of these keypoints (*Figure 5—figure supplement 1D–G*). For bout statistics, the two methods performed similarly well (*Figure 5—figure supplement 1F–J*). Thus, for at least one type of video and dataset, DeepEthogram outperformed an established approach.

There are several reasons why DeepEthogram might have done better on accuracy and F1 score. First, the videos tested were relatively low resolution, which restricted the number of keypoints on the mouse's body that could be labeled. High-resolution videos with more keypoints may improve the keypoint-based classification approach. Second, our videos were recorded with a top-down view, which means that the paw positions were often occluded by the mouse's body. A bottom-up or side view could allow for better identification of keypoints and may result in improved performance for the keypoint-based methods.

An alternative approach to DeepEthogram and other supervised classification pipelines could be to use an unsupervised behavior classification followed by human labeling of behavior clusters. In this approach, an unsupervised algorithm identifies behavior clusters without user input, and then the researcher identifies the cluster that most resembles their behavior of interest (e.g., 'cluster 3 looks

like face grooming'). The advantage of this approach is that it involves less researcher time due to the lack of supervised labeling. However, this approach is not designed to identify predefined behaviors of interest and thus, in principle, might not be well suited for the goal of supervised classification. We tested one such approach starting with the Mouse-Openfield dataset and DeepLabCut-generated keypoints (*Figure 5—figure supplement 1A, C*). We used B-SoID (*Hsu and Yttri, 2019*), an unsupervised classification pipeline for animal behavior, which identified 11 behavior clusters for this dataset (*Figure 5—figure supplement 2A, B*). These clusters were separable in a low-dimensional behavior space (*Figure 5—figure supplement 2B*), and B-SoID's fast approximation algorithm showed good performance (*Figure 5—figure supplement 2C*). For every frame in our dataset, we had human labels, DeepEthogram predictions, and B-SoID cluster assignments. By looking at the joint distributions of B-SoID clusters and human labels, there appeared to be little correspondence (*Figure 5—figure supplement 2D*). To assign human labels to B-SoID clusters, for each researcher-defined behavior, we picked the B-SoID cluster that had the highest overlap with the behavior of interest (red boxes, *Figure 5—figure supplement 2D*, right). We then evaluated these 'predictions' compared to DeepEthogram. For most behaviors, DeepEthogram performed better than this alternative pipeline (*Figure 5—figure supplement 2E*).

We note that the unsupervised clustering with post-hoc assignment of human labels is not the use for which B-SoID (*Hsu and Yttri, 2019*) and other unsupervised algorithms (*Wiltschko, 2015*; *Berman et al., 2014*) were designed. Unsupervised approaches are designed to discover repeated behavior motifs directly from data, without humans predefining the behaviors of interest (*Datta et al., 2019*; *Egnor and Branson, 2016*), and B-SoID succeeded in this goal. However, if one's goal is the automatic labeling of human-defined behaviors, our results show that DeepEthogram or other supervised machine learning approaches are better choices.

## DeepEthogram requires little training data to achieve high performance

We evaluated how much data a user must label to train a reliable model. We selected 1, 2, 4, 8, 12, or 16 random videos for training and used the remaining videos for evaluation. We only required that each training set had at least one frame of each behavior. We trained the feature extractors, extracted the features, and trained the sequence models for each split of the data into training, validation, and test sets. We repeated this process five times for each number of videos, resulting in 30 trained models per dataset. Given the large number of dataset variants for this analysis, to reduce overall computation time, we used DeepEthogram-fast and focused on only the Mouse-Ventral1, Mouse-Ventral2, and Fly datasets. Also, we trained the flow generator only once and kept it fixed for all experiments. For all but the rarest behaviors, the models performed at high levels even with only one labeled video in the training set (*Figure 6A–C*). For all the behaviors studied across datasets, the performance measured as accuracy or F1 score approached seemingly asymptotic levels after training on approximately 12 videos. Therefore, a training set of this size or less is likely sufficient for many cases.

We also analyzed the model's performance as a function of the number of frames of a given behavior present in the training set. For each random split, dataset, and behavior, we had a wide range of the number of frames containing a behavior of interest. Combining all these splits, datasets, and behaviors together, we found that the model performed with more than 90% accuracy when trained with only 80 example frames of a given behavior and over 95% accuracy with only 100 positive example frames (*Figure 6D*). Furthermore, DeepEthogram achieved an F1 score of 0.7 with only 9000 positive example frames, which corresponds to about 5 min of example behavior at 30 frames per second (*Figure 6E*, see *Figure 3P* for an example of ~0.7 F1 score). The total number of frames required to reach this number of positive example frames depends on how frequent the behavior is. If the behavior happens 50% of the time, then 18,000 total frames are required to reach 9000 positive example frames. Instead, if the behavior occurs 10% of the time, then 90,000 total frames are required. In addition, when the sequence model was used instead of using the predictions directly from the feature extractors, model performance was higher (*Figure 6F,G*) and required less training data (data not shown), emphasizing the importance of using long timescale information in the prediction of behaviors. Therefore, DeepEthogram models require little training to achieve high performance. As expected, as more training data are added, the performance of the model improves,

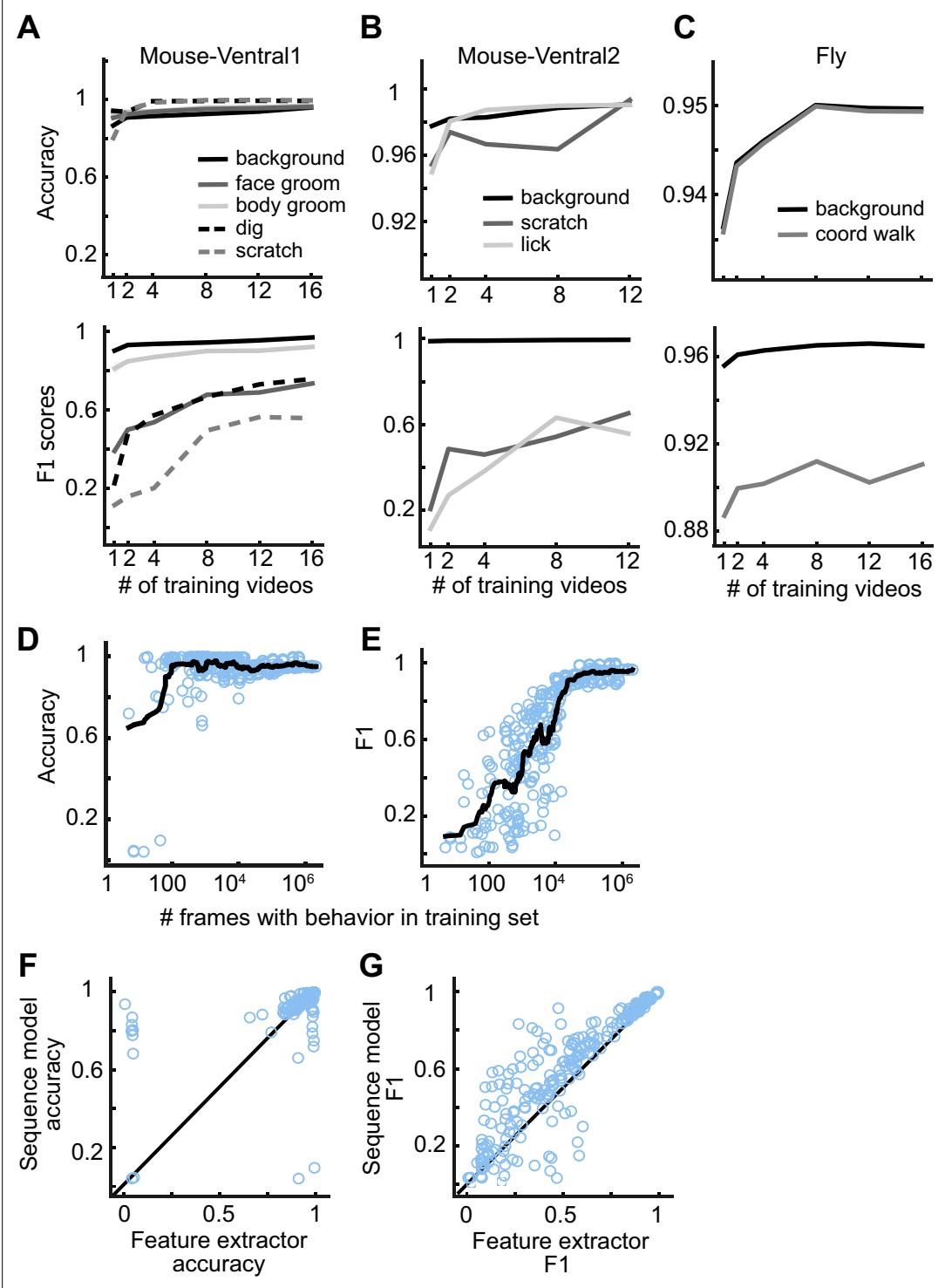

**Figure 6.** DeepEthogram performance as a function of training set size. (**A**) Accuracy (top) and F1 score (bottom) for DeepEthogram-fast as a function of the number of videos in the training set for Mouse-Ventral1, shown for each behavior separately. The mean is shown across five random selections of training videos. (**B, C**) Similar to (**A**), except for the Mouse-Ventral2 dataset and Fly dataset. (**D**) Accuracy of DeepEthogram-fast as a function of the number of frames with the behavior of interest in the training set. Each point is one behavior for one random split of the data, across datasets. The black line shows the running average. For reference, 104 frames is ~5 min of behavior at 30 frames per second.(**E**) Similar to (**D**), except for F1 score. (**F**) Accuracy for the predictions of DeepEthogram-fast using the feature extractors only or using the sequence model. Each point is one behavior from one split of the data, across datasets, for the splits used in (**D, E**). (**G**) Similar to (**F**), except for F1 score.

**Table 1.** Inference speed.

| Dataset | Resolution | Inference time (FPS) | | | | | |
| | | Titan RTX | | | Geforce 1080 Ti | | |
| | | DEG_f | DEG_m | DEG_s | DEG_f | DEG_m | DEG_s |
| --- | --- | --- | --- | --- | --- | --- | --- |
| Mouse-Ventral1 | 256 × 256 | 235 | 128 | 34 | 152 | 76 | 13 |
| Mouse-Ventral2 | 256 × 256 | 249 | 132 | 34 | 157 | 79 | 13 |
| Mouse-Openfield | 256 × 256 | 211 | 117 | 33 | 141 | 80 | 13 |
| Mouse-Homecage | 352 × 224 | 204 | 102 | 28 | 132 | 70 | 11 |
| Mouse-Social | 224 × 224 | 324 | 155 | 44 | 204 | 106 | 17 |
| Sturman-EPM | 256 × 256 | 240 | 123 | 34 | 157 | 83 | 13 |
| Sturman-FST | 224 × 448 | 157 | 75 | 21 | 106 | 51 | 9 |
| Sturman-OFT | 256 × 256 | 250 | 125 | 34 | 159 | 84 | 13 |
| Flies | 128 × 192 | 623 | 294 | 89 | 378 | 189 | 33 |

but this rather light dependency on the amount of training data makes DeepEthogram amenable for even small-scale projects.

## DeepEthogram allows rapid inference time

A key aspect of the functionality of the software is the speed with which the models can be trained and predictions about behaviors made on new videos. Although the versions of DeepEthogram vary in speed, they are all fast enough to allow functionality in typical experimental pipelines. On modern computer hardware, the flow generator and feature extractors can be trained in approximately 24 hr. In many cases, these models only need to be trained once. Afterwards, performing inference to make predictions about the behaviors present on each frame can be performed at ~150 frames per second for videos at 256 × 256 resolution for DeepEthogram-fast, at 80 frames per second for DeepEthogram-medium, and 13 frames per second for DeepEthogram-slow (*Table 1*). Thus, for a standard 30 min video collected at 60 frames per second, inference could be finished in 12 min for DeepEthogram-fast or 2 hr for DeepEthogram-slow. Importantly, the training of the models and the inference involve zero user time because they do not require manual input or observation from the user. Furthermore, this speed is rapid enough to get results quickly after experiments to allow fast analysis and experimental iteration. However, the inference time is not fast enough for online or closed-loop experiments.

## A GUI for beginning-to-end management of experiments

We developed a GUI for labeling videos, training models, and running inference (*Figure 7*). Our GUI is similar in behavior to those for BORIS (*Friard et al., 2016*) and JAABA (*Kabra et al., 2013*). To train DeepEthogram models, the user first defines which behaviors of interest they would like to detect in their videos. Next, the user imports a few videos into DeepEthogram, which automatically calculates video statistics and organizes them into a consistent file structure. Then the user clicks a button to train the *flow generator* model, which occurs without user time. While this model is training, the user can go through a set of videos frame-by-frame and label the presence or absence of all behaviors in these videos. Labeling is performed with simple keyboard or mouse clicks at the onset and offset of a given behavior while scrolling through a video in a viewing window. After a small number of videos have been labeled and the *flow generator* is trained, the user then clicks a button to train the *feature extractors*, which occurs without user input and saves the extracted features to disk. Finally, the *sequence model* can be trained automatically on these saved features by clicking another button. All these training steps could in many cases be performed once per project. With these trained models, the user can import new videos and click the *predict* button, which estimates the probability of each behavior on each frame. This GUI therefore presents a single interface for labeling videos, training models, and generating predictions on new videos. Importantly, this interface requires no programming by the end-user.

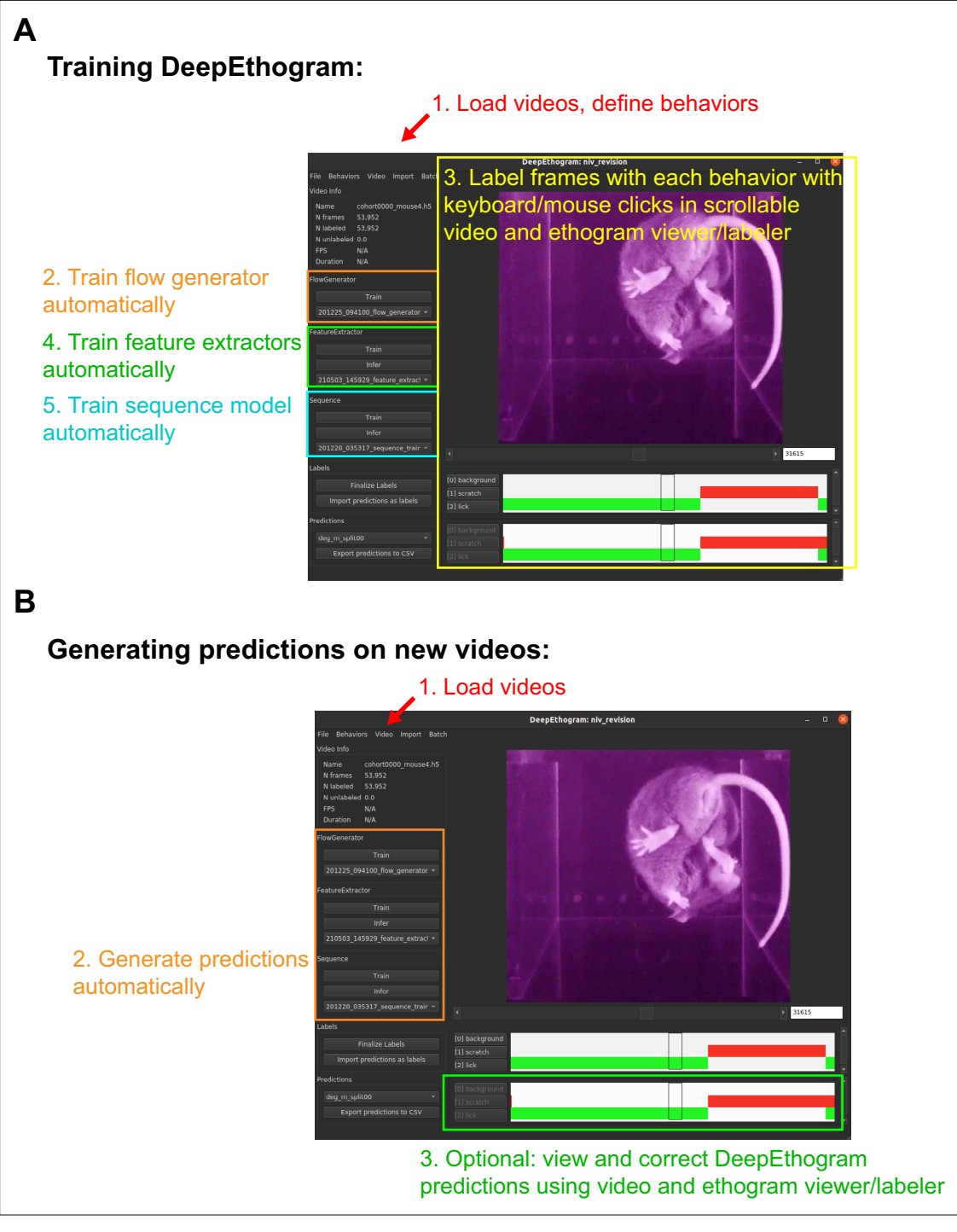

**Figure 7.** Graphical user interface. (**A**) Example DeepEthogram window with training steps highlighted. (**B**) Example DeepEthogram window with inference steps highlighted.

The GUI also includes an option for users to manually check and edit the predictions output by the model. The user can load into the GUI a video and predictions made by the model. By scrolling through the video, the user can see the predicted behaviors for each frame and update the labels of the behavior manually. This allows users to validate the accuracy of the model and to fix errors should they occur. This process is expected to be fast because the large majority of frames are expected to be labeled correctly, based on our accuracy results, so the user can focus on the small number of frames associated with rare behaviors or behaviors that are challenging to detect automatically.

Importantly, these new labels can then be used to retrain the models to obtain better performance on future experimental videos. Documentation for the GUI will be included on the project's website.

## Discussion

We developed a method for automatically classifying each frame of a video into a set of user-defined behaviors. Our open-source software, called DeepEthogram, provides the code and user interface necessary to label videos and train DeepEthogram models. We show that modern computer vision methods for action detection based on pretrained deep neural networks can be readily applied to animal behavior datasets. DeepEthogram performed well on multiple datasets and generalized across videos and animals, even for identifying rare behaviors. Importantly, by design, CNNs ignore absolute spatial location and thus are able to identify behaviors even when animals are in different locations and orientations within a behavioral arena (*Figure 3—figure supplement 12*). We anticipate this software package will save researchers great amounts of time, will lead to more reproducible results by eliminating inter-researcher variability, and will enable experiments that may otherwise not be possible by increasing the number of experiments a lab can reasonably perform or the number of behaviors that can be investigated. DeepEthogram joins a growing community of open-source computer vision applications for biomedical research (*Datta et al., 2019*; *Anderson and Perona, 2014*; *Egnor and Branson, 2016*).

The models presented here performed well for all datasets tested. In general, we expect the models will perform well in cases in which there is a high degree of agreement between separate human labelers, as our results in *Figures 3–5* indicate. As we have shown, the models do better with more training data. We anticipate that a common use of DeepEthogram will be to make automated predictions for each video frame followed by rapid and easy user-based checking and editing of the labels in the GUI for the small number of frames that may be inaccurately labeled. We note that these revised labels can then be used as additional training data to continually update the models and thus improve the performance on subsequent videos.

One of our goals for DeepEthogram was to make it general-purpose and applicable to all videos with behavior labels. DeepEthogram operates directly on the raw video pixels, which is advantageous because preprocessing is not required and the researcher does not need to make decisions about which features of the animal to track. Skeleton-based action recognition models, in which keypoints are used to predict behaviors, require a consistent skeleton as their input. A crucial step in skeleton-based action recognition is feature engineering, which means turning the x and y coordinates of keypoints (such as paws or joints) into features suitable for classification (such as the angle of specific joints). With different skeletons (such as mice, flies, or humans) or numbers of animals (one or more), these features must be carefully redesigned. Using raw pixel values as inputs to DeepEthogram allows for a general-purpose pipeline that can be applied to videos of all types, without the need to tailor preprocessing steps depending on the behavior of interest, species, number of animals, video angles, resolution, and maze geometries. However, DeepEthogram models are not expected to generalize across videos that differ substantially in any of these parameters. For example, models that detect grooming in top-down videos are unlikely to identify grooming in side-view videos.

Because DeepEthogram is a general-purpose pipeline, it will not perform as well as pipelines that are engineered for a specific task, arena, or species. For example, MARS was exquisitely engineered for social interactions between a black and white mouse (*Segalin, 2020*) and thus is expected to outperform DeepEthogram on videos of this type. Moreover, because DeepEthogram operates on raw pixels, it is possible that our models may perform more poorly on zoomed-out videos in which the animal is only a few pixels. Also, if the recording conditions change greatly, such as moving the camera or altering the arena background, it is likely that DeepEthogram will have to be retrained.

An alternate approach is to use innovative methods for estimating pose, including DeepLabCut (*Mathis, 2018*; *Nath, 2019*; *Lauer, 2021*), LEAP (*Pereira, 2018b*), and others (*Graving et al., 2019*), followed by frame-by-frame classification of behaviors based on pose in a supervised (*Segalin, 2020*; *Nilsson, 2020*; *Sturman et al., 2020*) or unsupervised (*Hsu and Yttri, 2019*) way. Using pose for classification could make behavior classifiers faster to train, less susceptible to overfitting, and less demanding of computational resources. Using pose as an intermediate feature could allow the user to more easily assess model performance. Depending on the design, such skeleton-based action recognition could aid multi-animal experiments by more easily predicting

behaviors separately for each animal, as JAABA does (*Kabra et al., 2013*). While we demonstrated that DeepEthogram can accurately identify social interactions, it does not have the ability to track the identities of multiple mice and identify behaviors separately for each mouse. Furthermore, tracking keypoints on the animal gives valuable, human-understandable information for further analysis, such as time spent near the walls of an arena, distance traveled, and measures of velocity (*Pennington, 2019*; *Sturman et al., 2020*). In addition, specific aspects of an animal's movements, such as limb positions and angles derived from keypoint tracking, can be directly related to each behavior of interest, providing an additional layer of interpretation and analysis of the behavior. DeepEthogram does not track parts of the animal's body or position and velocity information, and instead it focuses only on the classification of human-defined behaviors. Finally, because DeepEthogram uses 11 frames at a time for inputs, as well as relatively large models, it is not easily applicable to real-time applications, such as the triggering of optogenetic stimulation based on ongoing behaviors.

DeepEthogram may prove to be especially useful when a large number of videos or behaviors need to be analyzed in a given project. These cases could include drug discovery projects or projects in which multiple genotypes need to be compared. Additionally, DeepEthogram could be used for standardized behavioral assays, such as those run frequently in a behavioral core facility or across many projects with standardized conditions. Importantly, whereas user time scales linearly with the number of videos for manual labeling of behaviors, user time for DeepEthogram is limited to only the labeling of initial videos for training the models and then can involve essentially no time on the user's end for all subsequent movies. In our hands, it took approximately 1–3 hr for an expert researcher to label five behaviors in a 10 min movie from the Mouse-Openfield dataset. This large amount of time was necessary for researchers to scroll back and forth through a movie to mark behaviors that are challenging to identify by eye. If only approximately 10 human-labeled movies are needed for training the model, then only approximately 10–30 hr of user time would be required. Subsequently, tens of movies could be analyzed, across projects with similar recording conditions, without additional user time. DeepEthogram does require a fair amount of computer time (see Inference time above, Materials and methods); however, we believe that trading increasingly cheap and available computer time for valuable researcher effort is worthwhile. Notably, the use of DeepEthogram should make results more reproducible across studies and reduce variability imposed by inter-human labeling differences. Furthermore, in neuroscience experiments, DeepEthogram could aid identification of the starts and stops of behaviors to relate to neural activity measurements or manipulations.

Future extensions could continue to improve the accuracy and utility of DeepEthogram. First, DeepEthogram could be easily combined with an algorithm to track an animal's location in an environment (*Pennington, 2019*), thus allowing the identification of behaviors of interest and where those behaviors occur. Also, it would be interesting to use DeepEthogram's optic flow snippets as inputs to unsupervised behavior pipelines, where they could help to uncover latent structure in animal behavior (*Wiltschko, 2015*; *Berman et al., 2014*; *Batty, 2019*). In addition, while the use of CNNs for classification is standard practice in machine learning, recent works in temporal action detection use widely different sequence modeling approaches and loss functions (*Piergiovanni and Ryoo, 2018*; *Zeng, 2019*; *Monfort, 2020*). Testing these different approaches in the DeepEthogram pipeline could further improve performance. Importantly, DeepEthogram was designed in a modular way to allow easy incorporation of new approaches as they become available. While inference is already fast, further development could improve inference speed by using low-precision weights, model quantization, or pruning. Furthermore, although our model is currently designed for temporal action localization, DeepEthogram could be extended by incorporating models for spatiotemporal action localization, in which there can be multiple actors (i.e., animals) performing different behaviors on each frame.

## Materials and methods
### DeepEthogram pipeline
Along with this publication, we are releasing open-source Python code for labeling videos, training all DeepEthogram models, and performing inference on new videos. The code, associated documentation, and files for the GUI can be found at https://github.com/jbohnslav/deepethogram.

## Implementation

We implemented DeepEthogram in the Python programming language (version 3.7 or later; *Rossum et al., 2010*). We used PyTorch (*Paszke, 2018*; version 1.4.0 or greater) for all deep-learning models. We used PyTorch Lightning for training (*Falcon, 2019*). We used OpenCV (*Bradski, 2008*) for image reading and writing. We use Kornia (*Riba et al., 2019*) for GPU-based image augmentations. We used scikit-learn (*Pedregosa, 2021*) for evaluation metrics, along with custom Python code. CNN diagram in *Figure 1* was generated using PlotNeuralNet (*Iqbal, 2018*). Other figures were generated in Matplotlib (*Caswell, 2021*). For training, we used one of the following Nvidia GPUs: GeForce 1080Ti, Titan RTX, Quadro RTX6000, or Quadro RTX8000. Inference speed was evaluated on a computer running Ubuntu 18.04, an AMD Ryzen Threadripper 2950 X CPU, an Nvidia Titan RTX, an Nvidia Geforce 1080Ti, a Samsung 970 Evo hard disk, and 128 GB DDR4 memory.

## Datasets

All experimental procedures were approved by the Institutional Animal Care and Use Committees at Boston Children's Hospital (protocol numbers 17-06-3494R and 19-01-3809R) or Massachusetts General Hospital (protocol number 2018N000219) and were performed in compliance with the Guide for the Care and Use of Laboratory Animals.

For human-human comparison, we relabeled all videos for Mouse-Ventral1, Mouse-Ventral2, Mouse-Openfield, Mouse-Social, and Mouse-Homecage using the DeepEthogram GUI. Previous labels were not accessible during relabeling. Criteria for relabeling were written in detail by the original experimenters, and example labeled videos were viewed extensively before relabeling. Mouse-Ventral1 was labeled three times and the other datasets were labeled twice.

Videos and human annotations are available at the project website: https://github.com/jbohnslav/deepethogram.

### Kinetics700

To pretrain our models for transfer to neuroscience datasets, we use the Kinetics700 (*Carreira et al., 2019*) dataset. The training split of this dataset consisted of 538,523 videos and 141,677,361 frames. We first resized each video so that the short side was 256 pixels. During training, we randomly cropped 224 × 224 pixel images, and during validation, we used the center crop.

### Mouse-Ventral1

Recordings of voluntary behavior were acquired for 14 adult male C57BL/6J mice on the PalmReader device (Roberson et al., submitted). In brief, images were collected with infrared illumination and frustrated total internal reflectance (FTIR) illumination on alternate frames. The FTIR channel highlighted the parts of the mouse's body that were in contact with the floor. We stacked these channels into an RGB frame: red corresponded to the FTIR image, green corresponded to the infrared image, and blue was the pixel-wise mean of the two. In particular, images were captured as a ventral view of mice placed within an opaque 18 cm long × 18 cm wide × 15 cm high chamber with a 5 -mm-thick borosilicate glass floor using a Basler acA2000-50gmNIR GigE near-infrared camera at 25 frames per second. Animals were illuminated from below using nonvisible 850 nm near-infrared LED strips. All mice were habituated to investigator handling in short (~5 min) sessions and then habituated to the recording chamber in two sessions lasting 2 hr on separate days. On recording days, mice were habituated in a mock recording chamber for 45 min and then moved by an investigator to the recording chamber for 30 min. Each mouse was recorded in two of these sessions spaced 72 hr apart. The last 10 min of each recording was manually scored on a frame-by-frame basis for defined actions using a custom interface implemented in MATLAB. The 28 approximately 10 min videos totaled 419,846 frames (and labels) in the dataset. Data were recorded at 1000 × 1000 pixels and down-sampled to 250 × 250 pixels. We resized to 256 × 256 pixels using bilinear interpolation during training and inference.

### Mouse-Ventral2

Recordings of voluntary behavior were acquired for 16 adult male and female C57BL/6J mice. These data were collected on the iBob device. Briefly, the animals were enclosed in a device containing an opaque six-chambered plastic enclosure atop a glass floor. The box was dark and illuminated with only infrared light. Animals were habituated for 1 hr in the device before being removed to clean

the enclosure. They were then habituated for another 30 min and recorded for 30 min. Recorded mice were either wild type or contained a genetic mutation predisposing them to dermatitis. Thus, scratching and licking behavior were scored. Up to six mice were imaged from below simultaneously and subsequently cropped to a resolution of 270 × 240 pixels. Images were resized to 256 × 256 pixels during training and inference. Data were collected at 30 frames per second. There were 16 approximately 30 min videos for a total of 863,232 frames.

### Mouse-Openfield

Videos for the Mouse-Openfield dataset were obtained from published studies (*Orefice, 2019*; *Orefice, 2016*) and unpublished work (Clausing et al., unpublished). Video recordings of voluntary behavior were acquired for 20 adult male mice.

All mice were exposed to a novel empty arena (40 cm × 40 cm × 40 cm) with opaque plexiglass walls. Animals were allowed to explore the arena for 10 min, under dim lighting. Videos were recorded via an overhead-mounted camera at either 30 or 60 frames per second. Videos were acquired with 2–4 mice simultaneously in separate arenas and cropped with a custom Python script such that each video contained the behavioral arena for a single animal. Prior to analysis, some videos were brightened in FIJI (*Schindelin, 2012*), using empirically determined display range cutoffs that maximized the contrast between the mouse's body and the walls of the arena. Twenty of the 10 min recordings were manually scored on a frame-by-frame basis for defined actions in the DeepEthogram interface. All data were labeled by an experimenter. The 20 approximately 10 min videos totaled 537,534 frames (and labels).

### Mouse-Homecage

Videos for the mouse home cage behavior dataset were obtained from unpublished studies (Clausing et al., unpublished). Video recordings of voluntary behavior were acquired for 12 adult male mice. All animals were group-housed in cages containing four total mice. On the day of testing, all mice except for the experimental mouse were temporarily removed from the home cage for home cage behavior testing. For these sessions, experimental mice remained alone in their home cages, which measured 28 cm × 16.5 cm × 12.5 cm and contained bedding and nesting material. For each session, two visually distinct novel wooden objects and one novel plastic igloo were placed into the experimental mouse's home cage. Animals were allowed to interact with the igloo and objects for 10 min, under dim lighting. Videos were recorded via an overhead-mounted camera at 60 frames per second. Videos were acquired of two mice simultaneously in separate home cages. Following recordings, videos were cropped using a custom Python script such that each video contained the home cage for a single animal. Prior to analysis, all videos were brightened in FIJI48, using empirically determined display range cutoffs that maximized the contrast between each mouse's body and the bedding and walls of the home cage. Twelve of the 10 min recordings were manually scored on a frame-by-frame basis for defined actions in the DeepEthogram interface. Data were labeled by two experimenters. The 12 approximately 10 min videos totaled 438,544 frames (and labels).

### Mouse-Social

Videos for the mouse reciprocal social interaction test dataset were obtained from unpublished studies (Clausing et al., unpublished; Dai et al., unpublished). Video recordings of voluntary behavior were acquired for 12 adult male mice. All mice were first habituated to a novel empty arena (40 cm × 40 cm × 40 cm) with opaque plexiglass walls for 10 min per day for two consecutive days prior to testing. For each test session, two sex-, weight-, and age-matched mice were placed into the same arena. Animals were allowed to explore the arena for 10 min under dim lighting. Videos were recorded via an overhead-mounted camera at 60 frames per second. Videos were acquired with 2–4 pairs of mice simultaneously in separate arenas. Following recordings, videos were cropped using a custom Python script, such that each video only contained the behavioral arena for two interacting animals. Prior to analysis, all videos were brightened in FIJI48 using empirically determined display range cutoffs that maximized the contrast between each mouse's body and the walls of the arena. Twelve of the 10 min recordings were manually scored on a frame-by-frame basis for defined actions in the DeepEthogram interface. Data were labeled by two experimenters. The 12 approximately 10 minvideos totaled 438,544 frames (and labels).

## Sturman datasets

All Sturman datasets are from *Sturman et al., 2020*. For more details, please read their paper. Videos were downloaded from an online repository (https://zenodo.org/record/3608658#.YFt8-f4pCEA). Labels were downloaded from GitHub (https://github.com/ETHZ-INS/DLCAnalyzer, *Lukas von, 2021*). We arbitrarily chose 'Jin' as the labeler for model training; the other labelers were used for human-human evaluation (*Figures 4 and 5*).

## Sturman-EPM

This dataset consists of five videos. Only three contain one example of all behaviors. Therefore, we could only perform three random train-validation-test splits for this dataset (as our approach requires at least one example in each set). Images were resized to 256 × 256 during training and inference. Images were flipped up-down and left-right each with a probability of 0.5.

## Sturman-FST

This dataset consists of 10 recordings. Each recording has two videos, one top-down and one side view. To make this multiview dataset suitable for DeepEthogram, we closely cropped the mice in each view, resized each to 224 × 224, and concatenated them horizontally so that the final resolution was 448 × 224. We did not perform flipping augmentation.

## Sturman-OFT

This dataset consists of 20 videos. Images were resized to 256 × 256 for training and inference. Images were flipped up-down and left-right with probability 0.5 during training.

## Fly

Wild type DL adult male flies (*D. melanogaster*), 2–4 days post-eclosion were reared on a standard fly medium and kept on a 12 hr light-dark cycle at 25°. Flies were cold anesthetized and placed in a fly sarcophagus. We glued the fly head to its thorax and finally to a tungsten wire at an angle around 60° (UV cured glue, Bondic). The wire was placed in a micromanipulator used to position the fly on top of an air-suspended ball. Side-view images of the fly were collected at 200 Hz with a Basler A602f camera. Videos were down-sampled to 100 Hz. There were 19 approximately 30 min videos for a total of 3,419,943 labeled frames. Images were acquired at 168 × 100 pixels and up-sampled to 192 × 128 pixels during training and inference. Images were acquired in grayscale but converted to RGB (cv2. cvtColor, cv2.COLOR_GRAY2RGB) so that input channels were compatible with pretrained networks and other datasets.

## Models

### Overall setup

#### Problem statement

Our input features were a set of images with dimensions $[T, C, H, W]$, and our goal was to output the probability of each behavior on each frame, which is a matrix with dimensions $[T, K]$. $T$ is the number of frames in a video. $C$ is the number of input channels – in typical color images, this number is 3 for the red, green, and blue (RGB) channels. $H, W$ are the height and width of the images in pixels. $K$ is the number of user-defined behaviors we aimed to estimate from our data.

#### Training protocol

We used the ADAM optimizer (*Kingma and Ba, 2017*) with an initial learning rate of $1 \times 10^{-4}$ for all models. When validation performance saturated for 5000 training steps, we decreased the learning rate by a factor of $\frac{1}{\sqrt{10}}$ on the Kinetics700 dataset, or by a factor of 0.1 for neuroscience datasets (for speed), $5e - 7$. For Kinetics700, we used the provided train and validation split. For neuroscience datasets, we randomly picked 60% of videos for training, 20% for validation, and 20% for test (with the exception of the subsampling experiments for *Figure 5*, wherein we only used training and validation sets to reduce overall training time). Our only restriction on random splitting was ensuring that at least one frame of each class was included in each split. We were limited to five random splits of the data for most experiments due to the computational and time demands of retraining models. We save

both the final model weights and the best model weights; for inference, we load the best weights. Best is assessed by the minimum validation loss for flow generator models or the mean F1 across all non-background classes for feature extractor and sequence models.

### Stopping criterion

For Kinetics700 models, we stopped when the learning rate dropped below 5$e$-7. This required about 800,000 training steps. For neuroscience dataset flow generators, we stopped training at 10,000 training steps. For neuroscience dataset feature extractors, we stopped when the learning rate dropped below 5$e$-7, when 20,000 training steps were complete, or 24 hr elapsed, whichever came first. For sequence models, we stopped when the learning rate dropped below 5$e$-7, or when 100,000 training steps were complete, whichever came first.

### End-to-end training

We could, in theory, train the entire DeepEthogram pipeline end-to-end. However, we chose to train the flow generator, feature extractor, and then sequence models sequentially. By backpropagating the classification loss into the flow generator (*Zhu et al., 2017*), we risk increasing the overall number of parameters and overfitting. Furthermore, we designed the sequence models to have a large temporal receptive window. We therefore train on long sequences (see below). Very long sequences of raw video frames take large amounts of VRAM and exceed our computational limits. By illustration, to train on sequences of, for example, 180 frame snippets of 11 images, our tensor would be of shape [N × 33 × 180 × 256 × 256]. This corresponds to 24 GB of VRAM at a batch size of 16, just for the data and none of the neural activations or gradients, which is impractical. Therefore, we first extract features to disk and subsequently train sequence models.

### Augmentations

To improve the robustness and generalization of our models, we augmented the input images with random perturbations for all datasets during training. We used Kornia (*Riba et al., 2019*) for GPU-based image augmentation to improve training speed. We perturbed the image brightness and contrast, rotated each image by up to 10° , and flipped horizontally and vertically (depending on the dataset). The input to the flow generator model is a set of 11 frames; the same augmentations were performed on each image in this stack. On Mouse-Ventral1 and Mouse-Ventral2, we also flipped images vertically with a probability of 0.5. We calculated the mean and standard deviation of the RGB input channels and standardized the input channel-wise.

### Pretraining + transfer learning

All flow generators and feature extractors were first trained to classify videos in the Kinetics700 dataset (see below). These weights were used to initialize models on neuroscience datasets. Sequence models were trained from scratch.

## Flow generators

For optic flow extraction, a common algorithm to use is TV-L1 (*Carreira and Zisserman, 2017*). However, common implementations of this algorithm (*Bradski, 2008*) require compilation of C++, which would introduce many dependencies and make installation more difficult. Furthermore, recent work (*Zhu et al., 2017*) has shown that even simple neural-network-based optic flow estimators outperform TV-L1 for action detection. Therefore, we used CNN-based optic flow estimators. Furthermore, we found that saving optic flow as JPEG images, as is common, significantly degraded performance. Therefore, we computed optic flows from a stack of RGB images at runtime for both training and inference. This method is known as Hidden Two-Stream Networks (*Zhu et al., 2017*).

### Architectures

For summary, see *Table 2*.

### TinyMotionNet

For every timepoint, we extracted features based on one RGB image and up to 10 optic flow frames. Furthermore, for large datasets like Kinetics700 (*Carreira et al., 2019*), it was time-consuming and

**Table 2.** Model summary.

| Model name | Flow generator (parameters) | Feature extractor (parameters) | Sequence model (parameters) | # frames input to flow generator | # frames input to RGB feature extractor | Total parameters |
| --- | --- | --- | --- | --- | --- | --- |
| DeepEthogram-fast | TinyMotionNet (1.9 M) | ResNet18 × 2 (22.4M) | TGM (250K) | 11 | 1 | ~24.5 M |
| DeepEthogram-medium | MotionNet (45.8 M) | ResNet50 × 2 (49.2M) | TGM (250 K) | 11 | 1 | ~ 95.2 M |
| DeepEthogram-slow | TinyMotionNet3D (0.4 M) | ResNet3D-34 × 2 (127M) | TGM (250 K) | 11 | 11 | ~ 127.6 M |

required a large amount of disk space to extract and save optic flow frames. Therefore, we implemented TinyMotionNet (*Zhu et al., 2017*) to extract 10 optic flow frames from 11 RGB images 'on the fly,' as we extracted features. TinyMotionNet is a small and fast optic flow model with 1.9 million parameters. Similar to a U-Net (*Ronneberger et al., 2015*), it consists of a downward branch of convolutional layers of decreasing resolution and increasing depth. It is followed by an upward branch of increasing resolution. Units from the downward branch are concatenated to the upward branch. During training, estimated optic flows were output at 0.5, 0.25, and 0.125 of the original resolution.

### MotionNet

MotionNet is similar to TinyMotionNet except with more parameters and more feature maps per layer. During training, estimated optic flows were output at 0.5, 0.25, and 0.125 of the original resolution. See the original paper (*Zhu et al., 2017*) for more details.

### TinyMotionNet3D

This novel architecture is based on TinyMotionNet (*Zhu et al., 2017*), except we replaced all 2D convolutions with 3D convolutions. We maintained the height and width of the kernels. On the encoder and decoder branches, we used a temporal kernel size of 3, meaning that each filter spanned three images. On the last layer of the encoder and the *iconv* layers that connect the encoder and decoder branches, we used a temporal kernel of 2, meaning the kernels spanned two consecutive images. We aimed to have the model learn the displacement between two consecutive images (i.e., the optic flow). Due to the large memory requirements of 3D convolutional layers, we used 16, 32, and 64 filter maps per layer. For this architecture, we noticed large estimated flows in texture-less regions in neuroscience datasets after training on Kinetics. Therefore, we added a L1 sparsity penalty on the flows themselves (see 'Loss functions,' below).

### Modifications

For the above models, we deviated from the original paper. First, each time the flows were up-sampled by a factor of 2, we multiplied the values of the neural activations by 2. If the flow size increased from 0.25 to 0.5 of the original resolution, a flow estimate of 1 corresponds to four pixels and two pixels in the original image, respectively. To compensate for this distortion, we multiplied the up-sampled activations by 2. Secondly, when used in combination with the CNN feature extractors (see below), we did not compress the flow values to the discrete values between 0 and 255 (*Zhu et al., 2017*). In fact, we saw performance increases when keeping the continuous float32 values. Third, we did not backpropagate the classifier loss function into the flow generators as the neuroscience datasets likely did not have enough training examples to make this a sensible strategy. Finally, for MotionNet, we only output flows at three resolutions (rather than five) for consistency.

## Loss functions

In brief, we train flow generators to minimize reconstruction errors and minimize high-frequency components (to encourage smooth flow outputs).

### MotionNet loss

For full details, see original paper (*Zhu et al., 2017*). For clarity, we reproduce the loss functions here. We estimate the current frame given the next frame and an estimated optic flow as follows:

$$\hat{I}_0\left(i,j\right) = I_1\left(i + V^x\left(i,j\right), j + V^y\left(i,j\right)\right)$$

where $I_0, I_1$ are the current and next image. $i, j$ are the indices of the given pixel in rows and columns. $V^x\left(i,j\right), V^y\left(i,j\right)$ are the estimated x and y displacements between $I_0, I_1$, which means $V$ is the optic flow. We use Spatial Transformer Networks (*Jaderberg et al., 2015*) to perform this sampling operation in a differentiable manner (PyTorch function *torch.nn.functional.grid_sample*).

The image loss is the error between the reconstructed $\hat{I}_0$ and original $I_0$.

$$L_{pixel} = \frac{1}{N}\sum_{i,j}^{N}\rho\left(I_0 - \hat{I}_0\right)$$

where $\rho$ is the generalized Charbonnier penalty $\rho\left(x\right) = \left(x^2 + \epsilon^2\right)$, which reduces the influence of outliers compared to a simple L1 loss. Following (**Zhu et al., 2017**), we use $\alpha = 0.4, \epsilon = 1e^{-7}$.

The structural similarity (SSIM, **Wang et al., 2004**) loss encourages the reconstructed $\hat{I}_0$ and original $I_0$ to be perceptually similar:

$$L_{SSIM} = \frac{1}{N}\sum 1 - SSIM\left(I_0, \hat{I}_0\right)$$

The smoothness loss encourages smooth flow estimates by penalizing the $x$ and $y$ gradients of the optic flow:

$$L_{smooth} = \frac{1}{N}\sum \rho\left(\nabla V_x^x\right) + \rho\left(\nabla V_x^x\right) + \rho\left(\nabla V_x^y\right) + \rho\left(\nabla V_y^y\right)$$

We set the Charbonnier $\alpha = 0.3$.

For the TinyMotionNet3D architecture only, we added a flow sparsity loss that penalizes unnecessary flows:

$$L_{sparsity} = \frac{1}{N}\sum |V|$$

## Regularization

With millions of parameters and far fewer data points, it is likely that our models will overfit to the training data. Transfer learning (see above) ameliorates this problem somewhat, as does using dropout (see below). However, increasing dropout to very high levels reduces the representational space of the feature vector. To reduce overfitting, we used $L^2SP$ regularization (**Li et al., 2018**). A common form of regularization is weight decay, in which the sum of squared weights is penalized. However, this simple term could cause the model to 'forget' its initial knowledge from transfer learning. Therefore, $L^2SP$ regularization uses the initial weights from transfer learning as the target.

$$L_{regularization}\left(w\right) = \frac{\alpha}{2}\left\|w_s - w_s^0\right\|_2^2 + \frac{\beta}{2}\left\|w_s\right\|_2^2$$

For details, see the L2-SP paper (**Li et al., 2018**). $w$ are all trainable parameters of the network, excluding biases and batch normalization parameters. $\alpha$ is a hyperparameter governing how much to decay weights towards their initial values (from transfer learning). $w_s$ are current model weights, and $w_s^0$ are their values from pre-training. $\beta$ is a hyperparameter decaying new weights $w_s$ (such as the final linear readout layers in feature extractors) towards zero. For flow generator models, $\alpha = 1e^{-5}$. There are no new weights, so $\beta$ is unused.

The final loss is the weighted sum of the previous components:

$$L_{data} = \lambda_0 L_{pixel} + \lambda_1 L_{SSIM} + \lambda_2 L_{smooth} + \lambda_3 L_{sparsity} + L_{regularization}$$

Following **Zhu et al., 2017**, we set $\lambda_0 = 1, \lambda_1 = 1$. During training, the flow generator's output flows at multiple resolutions. From largest to smallest, we set $\lambda_2$ to be 0.01, 0.02, 0.04. For TinyMotionNet3D, we set $\lambda_3$ to 0.05 and reduced $\lambda_2$ by a factor of 0.25.

## Feature extractors

The goal of the feature extractor was to model the probability that each behavior was present in the given frame of the video (or optic flow stack). We used two-stream CNNs (**Zhu et al., 2017**; **Simonyan and Zisserman, 2014**) to classify inputs from both RGB frames and optic flow frames. These CNNs reduced an input tensor from $\left(N, C, H, W\right)$ pixels to $\left(N, 512\right)$ features. Our final fully connected layer estimated probabilities for each behavior, with output shape $\left(N, K\right)$. Here, $N$ is the batch size. We trained these CNNs on our labels, and then used the penultimate $\left(N, 512\right)$ *spatial features* or *flow features* as inputs to our sequence models (below).

### Architectures

For summary, see *Table 2*. We used the ResNet (**He et al., 2015**; **Hara et al., 2018**) family of models for our feature extractors, one for the spatial stream and one for the flow stream. For DeepEthogram-fast, we used ResNet18 with ~11 million parameters. For DeepEthogram-medium, we used a

ResNet50 with ~23 million parameters. We added dropout (**Hinton et al., 2012**) layers before the final fully connected layer. For DeepEthogram-medium, we added an extra fully connected layer of shape $(2048, 512)$ after the global average pooling layer to reduce the file size of stored features. For DeepEthogram-slow, we used a 3D ResNet34 (**Hara et al., 2018**) with ~63 million parameters. For DeepEthogram-fast and DeepEthogram-medium, these models were pretrained on ImageNet (**Deng, 2008**) with three input channels (RGB). We stacked 10 optic flow frames, for 20 input channels. To leverage ImageNet weights with this new number of channels, we used the mean weight across all three RGB channels and replicated it 20 times (**Wang et al., 2015**). This was only performed when adapting ImageNet weights to Kinetics700 models to resolve the input-frame-number discrepancy; the user will never need to perform this averaging.

## Loss functions

Our problem is a multi-label classification task. Each timepoint can have multiple positive examples. For example, if a mouse is licking its forepaw and scratching itself with its hindlimb, both 'lick' and 'scratch' should be positive. Therefore, we used a focal binary loss (**Lin et al., 2018**; **Marks, 2020**). The focal loss is the binary cross-entropy loss, weighted by probability, to de-emphasize the loss for already well-classified examples and to encourage the model to 'focus' on misclassified examples. Combined with up-weighting rare, positive examples, the data loss function is

$$L_{data} = \sum_{t,k} w_k \left(1 - p\right)^{\gamma} \cdot y_{t,k} \, log \left(p\left(k|x_t\right)\right) + p^{\gamma} \left(1 - y_{t,k}\right) log \left(1 - p\left(k|x_t\right)\right)$$

where $y_{t,k}$ is the ground truth label and was 1 if class $k$ occurred at time $t$, or otherwise was 0. $p\left(k|x_t\right)$ is our model output for class $k$ at time $t$. Note that for the feature extractor we only considered one timepoint at a time, so $t = 0$. $\gamma$ is a focal loss term (**Lin et al., 2018**); if $\gamma = 0$, this equation is simply the weighted binary cross-entropy loss. The larger the $\gamma$, the more the model down-weights correctly classified but insufficiently confident predictions. See the focal loss paper for more details (**Lin et al., 2018**). We chose $\gamma = 1$ for all feature extractor and sequence models for all datasets; see 'Hyperparameter optimization' section. We also use label smoothing (**Müller et al., 2019**), so that the target was 0.05 if $y_{t,k} = 0$ and 0.95 if $y_{t,k} = 1$. $w_k$ is a weight given to positive examples – note that there was no corresponding weight in the second term when our ground truth is 0. Intuitively, if we had a very rare behavior, we wanted to penalize the model more for an error on positive examples because there were so few examples of the behavior. We calculated the weight as follows:

$$w_k = \left(\frac{\sum_{i=1:N} y_{i,k}}{\sum_{i=1:N} 1 - y_{i,k}}\right)^{\beta}$$

The numerator is the total number of positive examples in our training set, and the denominator is the total number of negative examples in our training set. $\beta$ is a hyperparameter that we tuned manually. If $\beta = 1$, positive examples were weighted fully by their frequency in the training set. If $\beta = 0$, all training examples were weighted equally. By illustration, if only 1% of our training set had a positive example for a given behavior, with $\beta = 1$ our weight was 100 and with $\beta = 0$ this $w_k = 1$. We empirically found that with rare behaviors $\beta = 1$ drastically increased the levels of false positives, while with $\beta = 0$ many false negatives occurred. For all datasets, we set $\beta = 0.25$. This $w_k$ argument corresponds to *pos_weight* in *torch.nn.BCEWithLogitsLoss*.

We used L2-SP regularization as above. For feature extractors, we used $\alpha = 1e^{-5}$ and $\beta = 1e^{-3}$. See 'Hyperparameter optimization' section.

The final loss term is the sum of the data term and the regularization term:

$$L = L_{data} + L_{regularization}$$

## Bias initialization

To combat the effects of class imbalance, we set the bias parameters on the final layer to approximate the class imbalance (https://www.tensorflow.org/tutorials/structured_data/imbalanced_data). For example, if we had 99 negative examples and 1 positive example, we wanted to set our initial biases such that the model guessed 'positive' around 1% of the time. Therefore, we initialized the bias term as the log ratio of positive examples to negative examples:

$$b_k = log_e \frac{\sum_{i=1:\ N} y_{i,k}}{\sum_{i=1:\ N} 1 - y_{i,k}}$$

## Fusion

There are many ways to fuse together the outputs of the spatial and motion stream in two-stream CNNs (*Feichtenhofer et al., 2016*). For simplicity, we used late, average fusion. We averaged together the K-dimensional output vectors of the CNNs before the sigmoid function:

$$p\left(K|x_t\right) = \sigma\left(\frac{f_{spatial}\left(x_t\right) + f_{motion}\left(x_t\right)}{2}\right)$$

## Inference time

To improve inference speed, we use a custom inference video pipeline that uses only sequential video reading, batched model predictions, and multiprocessed data loading. Inference speed time is strongly related to input resolution and GPU hardware. We report timing on both a Titan RTX graphics card and a GeForce 1080 Ti graphics card.

## Sequence models

### Architecture

For summary, see *Table 2*. The goal of the sequence model was to have a wide temporal receptive field for classifying timepoints into behaviors. For human labelers, it is much easier to classify the behavior at time $t$ by watching a short clip centered at $t$ rather than viewing the static image. Therefore, we used a sequence model that takes as input a sequence of *spatial features* and *flow features* output by the feature extractors. Our criteria were to find a model that had a large temporal receptive field as context can be useful for classifying frames. However, we also wanted a model that had relatively few parameters as this model was trained from scratch on small neuroscience datasets. Therefore, we chose TGM (*Piergiovanni and Ryoo, 2018*) models, which are designed for temporal action detection. Unless otherwise noted, we used the following hyperparameters:

- Filter length: $L = 15$
- Number of input layers: $C = 1$
- Number of output layers: $C_{out} = 8$
- Number of TGM layers: 3
- Input dropout: 0.5
- Dropout of output features: 0.5
- Input dimensionality (concatenation of flow and spatial): $D = 1024$
- Number of filters: 8
- Sequence length: 180
- Soft attention, not 1D convolution
- We do not use super-events
- For more details, see *Piergiovanni and Ryoo, 2018*.

### Modifications

TGM models use two main features to make the final prediction: the $[T, D]$ input features (in our case, spatial and flow features from the feature extractors); and the $[T, D]$ learned features output by the TGM layers. The original TGM model performed 'early fusion' by concatenating these two features into shape $[T, 2D]$ before the 1D convolution layer. We found in low-data regimes that the model ignored the learned features, and therefore reduced to a simple 1D convolution. Therefore, we performed 'late fusion' – we used separate 1D convolutions on the input features and on the learned features. We averaged the output of these two layers (both $[T, K]$ activations before the sigmoid function). Secondly, in the original TGM paper, the penultimate layer was a standard 1D convolutional layer with 512 output channels. We found that this dramatically increased the number of parameters without improving performance significantly. Therefore, we reduced output channels to 128. The total number of parameters was ~264,000.

## Loss function

The data loss term is the weighted, binary focal loss as for the feature extractor above. For the regularization loss, we used simple L2 regularization because we do not pretrain the sequence models.

$$L_{regularization}\left(w\right) = \frac{\alpha}{2}\left\|w_s\right\|_2^2$$

We used $\alpha = 0.01$ for all datasets.

# Keypoint-based classification

We compared pixel-based (DeepEthogram) and skeleton-based behavioral classification on the Mouse-Openfield dataset. Our goal was to replicate *Sturman et al., 2020* as closely as possible. We chose this dataset because videos with this resolution of mice in an open field arena are a common form of behavioral measurement in biology. We first used DeepLabCut (*Mathis, 2018*) to label keypoints on the mouse and train pose estimation models. Due to the low resolution of the videos (200–300 pixels on each side), we could only reliably estimate seven keypoints: nose, left and right forepaw (if visible, or shoulder area), left and right hindpaw (if visible, or hip area), the base of the tail, and the tip of the tail. See *Figure 5—figure supplement 1A* for details. We labeled 1800 images and trained models using the DeepLabCut Colab notebook (ResNet50). Example performance on held-out data for unlabeled frames can be seen in *Figure 5—figure supplement 1B*. We used linear interpolation for keypoints with confidence below 0.9.

Using these seven keypoints for all videos, we computed a number of pose and behavioral features (Python, NumPy). As a check, we plotted the distribution of these features for each human-labeled behavior (*Figure 5—figure supplement 1C*). These features contained signal that can reliably discriminate behaviors. For example, the distance between the nose and the tailbase is larger during locomotion than during face grooming (*Figure 5—figure supplement 1C*, left).

We attempted to replicate *Sturman et al., 2020* as closely as possible. However, due to technical considerations, using the exact codebase was not possible. Our dataset is multilabel, meaning that two behaviors can present, and be labeled, on a single frame. Therefore, we could not use cross-entropy loss. Our videos are lower resolution, and therefore we used seven keypoints instead of 10. We normalized pixels by the width and height of the arena. We computed the centroid as the mean of all paw locations. Due to the difference in keypoints we selected, we had to perform our own behavioral feature expansion. (See 'Time-resolved skeleton representation,' *Sturman et al., 2020* supplementary methods.) We used the following features:

- x and y coordinates of all keypoints in the arena
- x and y coordinates after aligning relative to the body axis, such that the nose was to the right and the tailbase to the left
- Angles between
  - tail and body axis
  - each paw and the body axis
- Distances between
  - nose and tailbase
  - tail base and tip
  - left forepaw and left hindpaw, right forepaw and hindpaw, averaged
  - forepaw and nose
  - left and right forepaw
  - left and right hindpaw
- The area of the body (polygon enclosed by the paws, nose, and tailbase)

This resulted in 44 behavioral features for each frame. Following Sturman et al., we used T-15 frames to T + 15 frames as input to our classifier, which is 1364 features in total (44 * 31). We also used the same model architecture as Sturman et al.: a multilayer perceptron with 1364 neurons in the input layer, two hidden layers with 256 and 128 neurons, respectively, and ReLU activations. For simplicity, we used Dropout (*Srivastava et al., 2014*) with probability 0.35 between each layer.

To make the comparison as fair as possible, we implemented training tricks from DeepEthogram sequence models to train these keypoint-based models. These include the loss function (binary focal loss with up-weighting of rare behaviors), the stopping criterion (100 epochs or when learning rate reduces below 5e-7), learning rate scheduling based on validation F1 saturation, L2 regularization,

thresholds optimized based on F1, postprocessing based on bout length statistics, and inference using the best weights during training (as opposed to the final weights).

## Unsupervised classification

To compare DeepEthogram to unsupervised classification, we used B-SoID (*Hsu and Yttri, 2019*; version 2.0, downloaded January 18, 2021). We used the same DeepLabCut outputs as for the supervised classifiers. We used the Streamlit GUI for feature computation, UMAP embedding, model training, and classification. Our Mouse-Openfield dataset contained videos with a mixture of 30 and 60 frames-per-second videos. The B-SoID app assumed constant framerates; therefore, we downsampled the poses from 60 Hz to 30 Hz, performed all embedding and classification, and then up-sampled classified behaviors back to 60 Hz using nearest-neighbor up-sampling (PyTorch, see *Figure 5—figure supplement 2A*). B-SoID identified 11 clusters in UMAP space (*Figure 5—figure supplement 2B*, left). To compare unsupervised with post-hoc assignment to DeepEthogram, we first computed a simple lookup table that mapped human annotations to B-SoID clusters by counting the frames on which they co-occurred (*Figure 5—figure supplement 2D*). For each human label, we picked the B-SoID cluster with the maximum number of co-occurring labels; this defines a mapping between B-SoID clusters and human labels. We used this mapping to 'predict' human labels on the test set (*Figure 5—figure supplement 2E*). We compared B-SoID to the DeepEthogram-fast model for a fair comparison as B-SoID inference is relatively fast.

## Hyperparameter optimization

There are many hyperparameters in DeepEthogram models that can dramatically affect performance. We optimized hyperparameters using Ray Tune (*Liaw, 2018*), a software package for distributed, asynchronous model selection and training. Our target for hyperparameter optimization was the F1 score averaged over classes on the validation set, ignoring the background class. We used random search to select hyperparameters, and the Asynchronous Successive Halving algorithm (*Li, 2020*) to terminate poor runs. We did not perform hyperparameter optimization on flow generator models. For feature extractors, we optimized the following hyperparameters: learning rate, $\alpha$ and $\beta$ from the regularization loss, $\gamma$ from the focal loss, $\beta$ from the positive example weighting, whether or not to add a batch normalization layer after the final fully connected layer (*Kocaman et al., 2020*), dropout probability, and label smoothing. For sequence models, we optimized learning rate, regularization $\alpha$, $\gamma$ from the focal loss, $\beta$ from the positive example weighting, whether or not to add a batch normalization layer after the final fully connected layer (*Kocaman et al., 2020*), input dropout probability, output dropout probability, filter length, number of layers, whether or not to use soft attention (*Piergiovanni and Ryoo, 2018*), whether or not to add a nonlinear classification layer, and number of features in the nonlinear classification layer.

The absolute best performance could have been obtained by picking the best hyperparameters for each dataset, model size (DeepEthogram-f, DeepEthogram-m, or DeepEthogram-s), and split of the data. However, this would overstate performance for subsequent users that do not have the computational resources to perform such an optimization themselves. Therefore, we manually selected hyperparameters that had good performance on average across all datasets, models, and splits, and used the same parameters for all models. We will release the Ray Tune integration code required for users to optimize their own models, should they choose.

We performed this optimization on the O2 High Performance Compute Cluster, supported by the Research Computing Group, at Harvard Medical School. See http://rc.hms.harvard.edu for more information. Specifically, we used a cluster consisting of 8 RTX6000 GPUs.

## Postprocessing

The output of the feature extractor and sequence model is the probability of behavior $k$ occurring on frame $t$: $p_{t,k} = f(x_{t,k})$. To convert these probabilities into binary predictions, we thresholded the probabilities:

$$\hat{y_{t,k}} = p_{t,k} > \tau_k$$

We picked the threshold $\tau_k$ for each behavior $k$ that maximized the F1 score (below). We picked the threshold independently on the training and validation sets. On test data, we used the validation thresholds.

We found that these predictions overestimated the overall number of bouts. In particular, very short bouts were over-represented in model predictions. For each behavior $k$, we removed both 'positive' and 'negative' bouts (binary sequences of 1 s and 0 s, respectively) shorter than the first percentile of the bout length distribution in the training set.

Finally, we computed the 'background' class as the logical not of the other predictions.

## Evaluation and metrics

We used the following metrics: overall accuracy, F1 score, and the AUROC by class. Accuracy was defined as

$$Accuracy = \frac{TP+TN}{TP+TN+FP+FN}$$

where $TP$ is the number of true positives, $TN$ is the number of true negatives, $FP$ is the number of false positives, and $FN$ is the number of false negatives. We reported overall accuracy, not accuracy for each class.

F1 score was defined as

$$F_1 = \frac{1}{K} \sum_{k=1}^{K} 2 \frac{precision \cdot recall}{precision+recall}$$

where $precision = \frac{TP}{TP+FP}$ and $recall = \frac{TP}{TP+FN}$ . The above was implemented by *sklearn.metrics.f1_score* with argument *average='macro'*.

AUROC was computed by taking the AUROC for each class and averaging the result. This was implemented by *sklearn.metrics.roc_auc_score* with argument *average='macro'*.

## Shuffle

To compare model performance to random chance, we performed a shuffling procedure. For each model and random split, we randomly circularly permuted each video's labels 100 times. This means that the distribution of labels was kept the same, but the correspondence between predictions and labels was broken. For each of these 100 repetitions, we computed all metrics and then averaged across repeats; this results in one chance value per split of the data (gray bars, *Figure 3C–K*, *Figure 3—figure supplement 1B–J*, *Figure 3—figure supplement 2B–J*, *Figure 3—figure supplement 3B–J*).

## Statistics

We randomly assigned input videos to train, validation, and test splits (see above). We then trained flow generator models, feature extractor models, performed inference, and trained sequence models. We repeated this process five times for all datasets. For Sturman-EPM, because only three videos had at least one behavior, we split this dataset three times. When evaluating DeepEthogram performance, this results in N = 5 samples. For each split of the data, the videos in each subset were different; the fully connected layers in the feature extractor were randomly initialized with different weights; and the sequence model was randomly initialized with different weights. When comparing the means of multiple groups (e.g., shuffle, DeepEthogram, and human performance for a single behavior), we used a one-way repeated measures ANOVA, with subjects being splits. If this was significant, we performed a post-hoc Tukey's honestly significant difference test to compare means pairwise. For cases in which only two groups were being compared (e.g., model and shuffle without human performance), we performed paired t-tests with Bonferroni correction.

## Acknowledgements

We thank Woolf lab members Rachel Moon for data acquisition and scoring and Victor Fattori for scoring, and David Roberson and Lee Barrett for designing and constructing the PalmReader and iBob mouse viewing platforms. We thank the Harvey lab for helpful discussions and feedback on the manuscript. We thank Sturman et al. 2020 for making their videos and human labels publicly available. This work was supported by NIH grants R01 MH107620 (CDH), R01 NS089521 (CDH), R01 NS108410

(CDH), DP1 MH125776 (CDH), F31 NS108450 (JPB), R35 NS105076 (CJW), R01 AT011447 (CJW), R00 NS101057 (LLO), K99 DE028360 (DAY), European Research Council grant ERC-Stg-759782 (EC), an NSF GRFP (NKW), FCT fellowship PD/BD/105947/2014 (TC), a Harvard Medical School Dean's Innovation Award (CDH), and a Harvard Medical School Goldenson Research Award (CDH).

## Additional information

### Funding

| Funder | Grant reference number | Author |
|---|---|---|
| National Institutes of Health | R01MH107620 | Christopher D Harvey |
| National Institutes of Health | R01NS089521 | Christopher D Harvey |
| National Institutes of Health | R01NS108410 | Christopher D Harvey |
| National Institutes of Health | F31NS108450 | James P Bohnslav |
| National Institutes of Health | R35NS105076 | Clifford J Woolf |
| National Institutes of Health | R01AT011447 | Clifford J Woolf |
| National Institutes of Health | R00NS101057 | Lauren L Orefice |
| National Institutes of Health | K99DE028360 | David A Yarmolinsky |
| European Research Council | ERC-Stg-759782 | M Eugenia Chiappe |
| National Science Foundation | GRFP | Nivanthika K Wimalasena |
| Ministry of Education | PD/BD/105947/2014 | Tomás Cruz |
| Harvard Medical School | Dean's Innovation Award | Christopher D Harvey |
| Harvard Medical School | Goldenson Research Award | Christopher D Harvey |
| National Institutes of Health | DP1 MH125776 | Christopher D Harvey |

The funders had no role in study design, data collection and interpretation, or the decision to submit the work for publication.

### Author contributions

James P Bohnslav, Conceptualization, Developed the software and analyzed the data. Performed video labeling, Formal analysis, Funding acquisition, Investigation, Methodology, Performed video labeling, Software, Visualization, Writing - original draft, Writing – review and editing; Nivanthika K Wimalasena, Conceptualization, Funding acquisition, Investigation, Performed experiments. Performed video labeling, Performed experiments. Performed video labeling, Performed experiments. Performed video labeling, Performed experiments. Performed video labeling, Performed experiments. Performed video labeling, Performed video labeling, Writing – review and editing; Kelsey J Clausing, Yu Y Dai, Investigation, Performed experiments. Performed video labeling, Performed experiments. Performed video labeling, Performed experiments. Performed video labeling, Performed experiments. Performed video labeling, Performed experiments. Performed video labeling, Performed video labeling, Writing – review and editing; David A Yarmolinsky, Funding acquisition, Investigation, Performed experiments. Performed video labeling, Performed experiments. Performed video

labeling, Performed experiments. Performed video labeling, Performed experiments. Performed video labeling, Performed experiments. Performed video labeling, Performed experiments. Performed video labeling, Performed video labeling, Writing – review and editing; Tomás Cruz, Investigation, Performed experiments. Performed video labeling, Performed experiments. Performed video labeling, Performed experiments. Performed video labeling, Performed experiments. Performed video labeling, Performed experiments. Performed video labeling, Performed experiments. Performed video labeling, Writing – review and editing; Adam D Kashlan, Investigation, Performed video labeling; M Eugenia Chiappe, Lauren L Orefice, Funding acquisition, Supervised experiments, Supervised experiments, Supervised experiments, Supervision, Writing – review and editing; Clifford J Woolf, Conceptualization, Funding acquisition, Supervised experiments, Supervised experiments, Supervised experiments, Supervision, Writing – review and editing; Christopher D Harvey, Conceptualization, Funding acquisition, Methodology, Software, Supervised the software development and data analysis, Supervision, Writing - original draft, Writing – review and editing

### Author ORCIDs
James P Bohnslav (iD) http://orcid.org/0000-0002-9359-8907
M Eugenia Chiappe (iD) http://orcid.org/0000-0003-1761-0457
Christopher D Harvey (iD) http://orcid.org/0000-0001-9850-2268

### Ethics
All experimental procedures were approved by the Institutional Animal Care and Use Committees at Boston Children's Hospital (protocol numbers 17-06-3494R and 19-01-3809R) or Massachusetts General Hospital (protocol number 2018N000219) and were performed in compliance with the Guide for the Care and Use of Laboratory Animals.

### Decision letter and Author response
Decision letter https://doi.org/10.7554/eLife.63377.sa1
Author response https://doi.org/10.7554/eLife.63377.sa2

---

## Additional files

### Supplementary files
• Transparent reporting form

### Data availability
Code is posted publicly on Github and linked in the paper. Video datasets and human annotations are publicly available and linked in the paper.

The following previously published datasets were used:

| Author(s) | Year | Dataset title | Dataset URL | Database and Identifier |
|---|---|---|---|---|
| von Ziegler L, Sturman O, Bohacek J | 2020 | Videos for deeplabcut, noldus ethovision X14 and TSE multi conditioning systems comparisons | https://zenodo.org/record/3608658#.YFt8-f4pCEA | Zenodo, 10.5281/zenodo.3608658 |

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
