## [Decision Letter]

**Acceptance summary:**

DeepEthogram introduces a new tool to the neuroscience and behavior community that allow direct from-video-to-actions to be automatically identified. The authors comprehensively benchmark and provide data that demonstrates the tool's high utility in many common laboratory scenarios.

**Decision letter after peer review:**

Thank you for submitting your article "DeepEthogram: a machine learning pipeline for supervised behavior classification from raw pixels" for consideration by*eLife*. Your article has been reviewed by 3 peer reviewers, including Mackenzie Mathis as the Reviewing Editor and Reviewer #1, and the evaluation has been overseen by Timothy Behrens as the Senior Editor. The following individual involved in review of your submission has agreed to reveal their identity: Johannes Bohacek (Reviewer #3).

The reviewers have discussed the reviews with one another and the Reviewing Editor has drafted this decision to help you prepare a revised submission.

As the editors have judged that your manuscript is of interest, but as described below that additional experiments are required before it is published, we would like to draw your attention to changes in our revision policy that we have made in response to COVID-19 (https://elifesciences.org/articles/57162). First, because many researchers have temporarily lost access to the labs, we will give authors as much time as they need to submit revised manuscripts. We are also offering, if you choose, to post the manuscript to bioRxiv (if it is not already there) along with this decision letter and a formal designation that the manuscript is "in revision at eLife". Please let us know if you would like to pursue this option. (If your work is more suitable for medRxiv, you will need to post the preprint yourself, as the mechanisms for us to do so are still in development.)

Summary:

Bohnslav et al., present a new toolkit and GUI for using video input to extract behavioral states (ethograms) using a set of established deep neural networks. They show their pipeline works on range of laboratory datasets, and provide metrics comparing network performance to humans. However, the reviewers all agreed there are several key revisions needed in order to support the main claims of the paper. These revolve around benchmarking, datasets, and a more careful handling of related work, limitations of such a software, and clarifying methods. We have collectively decided to send the individual reviews from each reviewer, and ask you address those (and perhaps combine where you see fit), but we urge you to focus in on the following points for your revision.

Datasets:

The reviewers each expressed concern over the simplicity of the datasets and the potentially limited scope of DeepEthogram in relation. For example, the authors claim these are difficult datasets, but in fact we feel they are not representative of the laboratory videos often collected: they have very static backgrounds, no animals have cables or other occluders. We would urge the authors to use other datasets, even those publically available, to more thoroughly benchmark performance in a broader collection of behaviors.

Benchmarking:

While DeepEthogram could be an important tool to the growing toolbox of deep learning tools for behavior, we felt that there are sufficiently other options available that the authors should directly compare performance. While we do appreciate that comparing to the "gold standard" of human-labeled data, the real challenge with such datasets is even humans tend not to agree on a semantic label. Here, the authors only use two humans for ground-truth annotation, but there is a concern of an outlier. Typically, 3 humans are used to overcome a bit of this limitation. Therefore, we suggest carefully benchmarking against humans (i.e., increase the number of ground truth annotations), and please see the individual reviewer comments with specific questions related to other published/available code bases where you can directly compare your pipelines performance.

Methods, Relation to other packages, and Limitations:

The reviewers raised several points where methods are unclear, or how an analysis was performed was not clear. In particular, we ask you to check reviewer #3's comments carefully regarding methods. Moreover, we think a more nuanced discussion about when to do some "pre-processing" (like pose estimation) would be beneficial vs. straight to an ethogram, and visa versa. In particular, it's worth nothing that often times having an intermediate bottleneck such as key points allows the user to more easily assess network performance (keypoints are a defined ground truth vs. semantic action labels).

In total, the reviews are certainly enthusiastic about this work, and do hope you find these suggestions helpful. We look forward to reading your revision.*Reviewer #1:*

Bohnslav et al., present a new tool to quantify behavior actions directly from video. I think this is a nice addition to the growing body of work using video to analyze behavior. The paper is well written, clear for a general audience, and takes nice innovations in computer vision into life sciences and presents a usable tool for the community. I have a few critical points that I believe need addressed before publication, mostly revolving around benchmarking, but overall I am enthusiastic about this work being in*eLife*.

In the following sections I highlight areas I believe can be improved upon.

In relation to prior work: The authors should more explicitly state their contribution, and the field's contributions, to action recognition. The introduction mostly highlights limitations of unsupervised methods to perform behavioral analysis (which to note, produces the same outputs as this paper, i.e. an ethogram) and key point estimation alone, which of course is tackling a different problem. What I would like to see is a more careful consideration of the state-of-the-field in computer vision for action recognition, and clearly defining what the contribution is in this paper the cover letter alludes to them developing novel computer vision aspects of the package, but from the code base, etc, it seems they utilize (albeit nicely!) pre-existing works from ~3 years ago, begging the question if this is truly state-of-the-art performance. Moreover, and this does hurt novelty a bit, this is not the first report in life science of such a pipeline, so this should be clearly stated. I don't think it's required to compare this tool to every other tool available, but I do think discussing this in the introduction is of importance (but again, I am still enthusiastic for this being in*eLife*).

"Our model operates directly on the raw pixel values of videos, and thus it is generally applicable to any case with video data and binary behavior labels and further does not require pre-specification of the body features of interest, such as keypoints on limbs or fitting the body with ellipses." – please include references to the many other papers that do this as well. For example, please see:

Data-driven analyses of motor impairments in animal models of neurological disorders https://journals.plos.org/plosbiology/article?id=10.1371/journal.pbio.3000516

LSTM Self-Supervision for Detailed Behavior Analysis https://openaccess.thecvf.com/content_cvpr_2017/html/Brattoli_LSTM_Self-Supervision_for_CVPR_2017_paper.html

Facial expressions of emotion states and their neuronal correlates in mice https://science.sciencemag.org/content/368/6486/89/tab-figures-data (not deep learning, but similar workflow; also extract features as the authors here do, and gets good performance using old CV techniques)

Deep learning improves automated rodent behavior recognition within a specific experimental setup https://www.sciencedirect.com/science/article/pii/S0165027019303930

I think Figure 1A is a bit misleading, it's not clear anymore that manual annotation is the only or most common other alternative pipeline (discussed below in benchmarking)- many tools for automated analysis now exist, and tools like JAABA and MotionMapper have been around for 5+ years; I would rather like to see a comparison workflow to "unsupervised methods," and/or keypoint estimation + classification with supervised or unsupervised means.

Lastly, they do not discuss key papers in life science for automated animal ethogram building, such as Live Mouse Tracker (https://livemousetracker.org/), BORIS and related Behatrix. Not only should these important papers be discussed, they should likely be benchmarked if the authors want to claim SOTA (see below).

Datasets: the authors claim they picked challenging datasets ("Diverse and challenging datasets to test DeepEthogram"), but I don't believe this is the case and they should tone down this statement. In fact, the datasets presented are rather easy to solve (the camera is orthogonal to the animal, i.e. top or bottom, or the animal's position is fixed, and the background is homogeneous, rarely the case even for laboratory experiments). I would urge them to use another more challenging dataset, and/or discuss the limitations of this work. For example, a mouse in a standard home cage with bedding, nests, huts, etc would pose more challenges, or they could report their performance on the Kinect700 dataset, which they pretrain on anyhow.

Benchmarking: The authors don't directly compare their work to that of other tools available in the field. Is their approach better (higher performance) than:

(1) unsupervised learning methods

(2) pose estimation plus classifiers or unsupervised clustering (as done in LEAP, DeepLabCut, B-SOiD, SIMBA, and the ETH DLC-Analyzer)

(3) tools that automate ethogram building, such as JAABA, BORIS/Behatrix.

Therefore, more results should be presented in relation to key works, and/or a more clear introduction on this topic should be presented.

– For example, they claim it's hard to match the resulting clusters from unsupervised learning to their "label:" i.e., "their outputs can be challenging to match up to behaviors of interest in cases in which researchers have strong prior knowledge about the specific behaviors relevant to their experiments". But this is not really a fair statement; one can simply look at the clusters and post-hoc assign a label, which has been nicely done in MotionMapper, for example.

– In pose estimation, one gets an animal-centric lower dimensional representation, which can be mapped onto behavioral states (ethograms), or used for kinematic analysis if desired. However, there is the minimal number of key points needed to make a representation that can still be used for ethogram building. Is the raw-pixel input truly better than this for all behaviors? For example, on the simple datasets with black backgrounds presented in this work, the background pixels are useless, and don't hinder the analysis. However, if the background dynamically changed (camera is moving, or background changes (lighting, bedding etc)), then the classification task from raw pixels becomes much harder than the task of extracted keypoints to classification task. Therefore, I think the authors should do the following: (1) discuss this limitation clearly in the paper, and (2) if they want to claim their method has universally higher performance, they need to show this on both simple and more challenging data.

Moveover, the authors discuss 4 limitations of other approaches, but do not address them in their work, i.e.:

– "First, the user must specify which features are key to the behavior (e.g. body position or limb position), but many behaviors are whole-body activities that could best be classified by full body data." – can they show an example where this is true? It seems from their data each action could be easily defined by kinematic actions of specific body parts a priori.

– "Second, errors that occur in tracking these features in a video will result in poor input data to the classification of behaviors, potentially decreasing the accuracy of labeling." – but is poor video quality not an issue for your classification method? The apple-to-apple comparison here is having corrupted video data as "bad" inputs – of course any method will suffer with bad data input.

– "Third, users might have to perform a pre-processing step between their raw videos and the input to these algorithms, increasing pipeline complexity and researcher time." – can they elaborate here? What preprocessing is needed for pose estimation, that is not needed for this, for example? (Both require manual labor, and given the time estimates, DEG takes longer to label than key point estimation due to the human needing to be able to look at video clips (see their own discussion)).

– "Fourth, the selection of features often needs to be tailored to specific video angles, behaviors (e.g. social behaviors vs. individual mice), species, and maze environments, making the analysis pipelines often specialized to specific experiments." – this is absolutely true, but also a limitation to the authors work, where the classifiers are tailored, the video should be a fixed perspective, background static, etc. So again I don't see this as a major limitation that makes pose estimation a truly invalid option.

Benchmarking and Evaluation:

– "We evaluated how many video frames a user must label to train a reliable model. We selected 1, 2, 4, 8, 12, or 16 random videos for training and used the remaining videos for evaluation. We only required that each training set had at least one frame of each behavior. We trained the feature extractors, extracted the features, and trained the sequence models for each split of the data." – it is not clear how many FRAMES are used here; please state in # of frames in Figure 5 and in the text (not just video #'s).

Related: "Combining all these models together, we found that the model performed with more than 90% accuracy when trained with only 80 example frames" This again is a bit misleading, as the user wants to know the total # of frames needed for your data, i.e. in this case this means that a human needs to annotate at least 80-100 frames per behavior, which for 5 states is ~500 frames; this should be made more explicit.

– "We note that here we used DEG-fast due to the large numbers of splits of the data, and we anticipate that the more complex DEG-medium and DEG-slow models might even require less training data." – this would go against common assumptions in deep learning; the deeper the models, the more prone to overfitting you are with less data. Please revise, or show the data that this statement is true.

– "Human-human performance was calculated by defining one labeler as the "ground truth" and the other labeler as "predictions", and then computing the same performance metrics as for DEG. " – this is a rather unconventional way to measure ground truth performance of humans. Shouldn't the humans be directly compared for % agreement and % disagreement on the behavioral state? (i.e., add a plot to the row that starts with G in figure 3).

To note, this is a limitation of such approaches, compared to pose-estimation, as humans can disagree on what a "behavior" is, whereas key points have a true GT, so I think it's a really important point that the authors address this head on (thanks!), and could be expanded in the discussion. Notably, MARS puts a lot of effort into measuring human performance, and perhaps this could be discussed in the context of this work as well.

*Reviewer #2:*

It was a pleasure reviewing the methodological manuscript describing DeepEthogram, a software developed for supervised behavioral classification. The software is intended to allow users to automate classification/quantification of complex animal behaviors using a set of supervised deep learning algorithms. The manuscript combines a few state-of-art neural networks into a pipeline to solve the problem of behavior classification in a supervised way. The pipeline uses well-established CNN to extract spatial features from each still frame of the videos that best predicts the user-provided behavior labels. In parallel, optical flow for each frame is estimated through another CNN, providing information about the "instantaneous" movement for each pixel. The optical flow "image" is then passed to another feature extractor that has the same architecture as the spatial feature extractor, and meaningful patterns of pixel-wise movements are extracted. Finally, the spatial feature stream and the optical flow feature stream are combined and fed into a temporal Gaussian mixture CNN, which can pool together information across long periods of time, mimicking human classifiers who can use previous frames to inform classification of behavior in current frame. The resulting pipeline provides a supervised classification algorithm that can directly operate on raw videos, while maintaining a relatively small computational demands on the hardware.

While I think something like DeepEthogram is needed in the field, I think the authors could do substantially more to validate that DeepEthogram is the ticket. In particular, I find the range of datasets validated in the manuscript poorly representative of the range of behavioral tracking circumstances that researchers routinely face. First, in all exemplar datasets, the animals are recorded in a completely empty environment. The animals are not interacting with any objects as they might in routine behavioral tests; there are no cables attached to them (which is routine for optogenetic studies, physiological recording studies, etc); they are alone (Can DeepEthogram classify social behaviors? the github page lists this as a typical use case); there isn't even cage bedding.

The authors also tout the time saving benefits of using deep ethogram. However, with their best performing implementation (DEG slow), with a state of the art computer, with a small video (256 x 256 pixels, width by height), the software runs at 15 frames per second (nearly 1/2 the speed of the raw video). My intuition is that this is on the slow side, given that many behaviors can be scored by human observers in near real time if the observer is using anything but a stopwatch. It would be nice to see benchmarks on larger videos that more accurately reflect the range of acquisition frames. If it is necessary for users to dramatically downsample videos, this should be made clear.

Specific comments:

– It would be nice to see if DeepEthogram is capable of accurately scoring a behavior across a range of backgrounds. For example, if the model is trained on a sideview recording of an animal grooming in its cage, can it accurately score an animal in an open field doing the same from an overhead view, or a side view? If the authors provided guidance on such issues to the reader this would be helpful.

– The authors should highlight that human scoring greatly outperforms DEG on a range of behaviors when comparing the individual F1 scores in Figure 3. Why aren't there any statistics for these comparisons?

– Some of the F1 scores for individual behaviors look very low (~0.5). It would be nice to know what chance performance is in these situations and if the software is performing above chance.

– I find it hard to understand the size of the data sets used in the analyses. For instance, what is 'one split of the data', referenced in Figure 3? Moreover, the authors state "We selected 1, 2, 4, 8, 12, or 16 random videos for training and used the remaining videos for evaluation" I have no idea what this means. What is the length and fps of the video?

– Are overall F1 scores in Figure 3 computed as the mean of the individual scores on each component F1 score, or the combination of all behaviors (such that it weights high frequency behaviors)? It's also difficult to understand what the individual points in Figure 4 (a-c) correspond to.

– The use of the names Mouse-1, Mouse-2 etc for experiments are confusing because it can appear that these experiments are only looking at single mice. I would change the nomenclature to highlight that these reflect experiments with multiple mice.

– It is not clear why the image has to be averaged across RGB channels and then replicated 20 times for the spatial stream. The author mentioned "To leverage ImageNet weights with this new number of channels", and I assume this means the input to the spatial stream has to have same shape (number of weights) as the input to the flow stream. However why this is the case is not clear, especially considering two feature extractor networks are independently trained for spatial and flow streams. Lastly this might raise the question of whether there will be valuable information in the RGB channels separately that will be lost from the averaging operation (for example, certain part of an animal's body has different color than others but is equal-luminous).

– It is not intuitive why simple average pooling is sufficient for fusing the spatial and flow streams. It can be speculated that classification of certain behavior will benefit much more from optical flow features while other behaviors benefits from still image features. I'm curious to see whether an additional layer at the fusing stage that has behavior-specific weights could improve performance.

– Since computational demands is one of the major concern in this article, I'm wondering whether exploiting the sparse nature of the input images would further improve the performance of the algorithm. Often times the animal of interests only occupies a small number of pixels in the raw images, and some simple thresholding of the images, or even user-defined masking of the images, together with use of sparse data backends and operations should in theory significantly reduce the computational demands for both the spatial and flow feature extractor networks.

*Reviewer #3:*

The paper by Bohnslav et al., presents a software tool that integrates a supervised machine learning algorithm for detecting and quantifying behavior directly from raw video input. The manuscript is well-written, the results are clear. Strengths and weaknesses of the approach are discussed and the work is appropriately placed in the bigger context of ongoing research in the field. The algorithms demonstrate high performance and reach human-level accuracy for behavior recognition. The classifiers are embedded in an excellent user-friendly interface that eliminates the need of any programming skills on the end of the user. Labeled datasets can even be imported. We suggest additional analyses to strengthen the manuscript.

1) Although the presented metrics for accuracy and F1 are state of the art it would be useful to also report absolute numbers for some of the scored behaviors for each trial, because most behavioral neuroscience studies actually report behavior in absolute numbers and/or duration of individual behaviors (rears, face grooms, etc.). Correlation of human and DEG data should also be presented on this level. This will speak to many readers more directly than the accuracy and F1 statistics. For this, we would like to see a leave-one-out cross-validation or a k-fold cross-validation (ensure that each trial ends up exactly once in a cross validation set) that enables a final per-trial readout. This can be done with only one of the DEG types (e.g "fast"). The current randomization approach of 60/20/20% (train/validate/test) with a n of 3 repeats is insufficient, since it a) allows per-trial data for at most 60% of all files and b) is susceptible to artefacts due to random splits (i.e one abnormal trial can be over or under represented in the cross validation sets).

2) In line with comment 1) we propose to update Figure 4, which at the moment uses summed up data from multiple trials. We would rather like to see each trial represented by a single data-point in this figure (#bouts/#frames by behavior). As alternative to individual scatterplots, correlation-matrix-heatmaps could be used to compare different raters.

3) Direct benchmarking against existing datasets is necessary. With many algorithms being published these days, it is important to pick additional (published) datasets and test how well the classifiers perform on those videos. Their software package already allows import of labeled datasets, some are available online. For example, how well can DeepEthogram score…

a. grooming in comparison to Hsu and Yttri (REF #17) or van den Boom et al., (2017, J Neurosci Methods).

b. rearing in comparison to Sturman et al., (REF #21).

c. social interactions compared to (Segalin et al., (REF #7) or Nilsson et al., (REF #19)).

4) In the discussion on page 19 the authors state: "Subsequently, tens to hundreds to thousands of movies could be analyzed, across projects and labs, without additional user-time, which would normally cost additionally hundreds to thousands of hours of time from researchers." This sentence suggests that a network trained on the e.g. the open field test in one lab can be transferred across labs. This key issue of "model transferability" should be tested. E.g. the authors could use the classifier from mouse#3 and test is on another available top-view recording dataset recorded in a different lab with different open-field setup (datasets are available online, e.g. REF #21).

5) Figure 5D/E: Trendline is questionable, we would advise to fit a sigmoid trendline, not an arbitrarily high order polynomial. Linear trend lines (such as shown in Figure 4) should include R^2^values on the plot or in the legend.

6) In the discussion, the authors do a very good job highlighting the limitations and advantages of their approach. The following limitations should however be expanded:

a. pose-estimation-based approaches (e.g. DLC) are going to be able to track multiple animals at the same time (thus allowing e.g. better read-outs of social interaction). It seems this feature cannot be incorporated in DeepEthogram.

b. Having only 2 human raters is a weakness that should briefly be addressed. Triplicates are useful for assessing outlier values, this could be mentioned in light of the fact that the F1 score of DeepEthogram occasionally outperforms the human raters (e.g. Figure 3C,E).

c. Traditional tracking measures such as time in zone, distance moved and velocity cannot be extracted with this approach. These parameters are still very informative and require a separate analysis with different tools (creating additional work).

d. The authors are correct that the additional time required for behavior analysis (due to the computationally demanding algorithms) is irrelevant for most labs. However, they should add (1) that the current system will not be able to perform behavior recognition in real time (thus preventing the use of closed-loop systems, which packages such as DLC have made possible) and (2) that the speed they discuss on page 16 is based on an advanced computer system (GPU, RAM) and will not be possible with a standard lab computer (or provide an estimate how long training would require if it is possible).

[Editors' note: further revisions were suggested prior to acceptance, as described below.]

Thank you for resubmitting your work entitled "DeepEthogram, a machine learning pipeline for supervised behavior classification from raw pixels" for further consideration by*eLife*. Your revised article has been evaluated 3 peer reviewers, one of whom is a member of our Board of Reviewing Editors, and the evaluation has been overseen by Timothy Behrens as the Senior Editor. The following individual involved in review of your submission has agreed to reveal their identity: Johannes Bohacek (Reviewer #3).

The manuscript has been improved but there are some remaining issues that need to be addressed, as outlined below:

The reviewers all felt the manuscript was improved, and thank the authors for the additional datasets and analysis. We would just like to see two items before the publication is accepted fully.

(1) Both reviewer #1 and #2 note the new data is great, but lacks human ground truth. Both for comparison, and releasing the data for others to benchmark on, it would be please include the data. We also understand that obtaining ground truth from 3 persons is a large time commitment, but even if there is one person, this data should be included for all datasets shown in Figure 3.

(2) Please include links for the raw videos used in this work; it is essential for others to benchmark and use to validate the algorithm presented here (see Reviewer 3: "raw videos used in this work (except the ones added during the revision) are – it appears – not accessible online").

Lastly, reviewer 3 notes that perhaps, still, some use-cases are best suited for DeepEthogram, while others more for pose-estimation plus other tools, but this of course cannot be exhaustively demonstrated here; at your discretion you might want to address in the discussion, but we leave that up to your judgement.*Reviewer #1:*

I thank the authors for the revisions and clarifications, and I think the manuscript is much improved. Plus, the new datasets and comparisons to B-iOD and R-Analyzer (Sturman) are a good additions.

One note is that is not clear which datasets have ground truth data; namely, in the results 5 datasets they use for testing are introduced:

"Mouse-Ventral1"

"Mouse-Ventral2"

"Mouse-Openfield"

"Mouse-Homecage"

"Mouse-Social"

plus three datasets from published work by Sturman et al.,

and "fly"

Then it states that all datasets were labeled; yet, Figure 3 has no ground truth for "Mouse-Ventral2" , "Mouse-Homecage" , "Mouse-Social" or 'Fly" -- please correct and include the ground truth. I do see that is says that only a subset of each of the 2 datasets in Figure 3 are labeled with 3 humans, but minimally then the rest (1 human?) should be included in Figure 3 (and be made open source for future benchmarking).

It appears from the discussion this was done (i.e., at least 1 human, as this is of course required for the supervised algorithm too):

"In our hands, it took approximately 1-3 hours for an expert researcher to label five behaviors in a ten-minute movie from the Mouse-Openfield dataset" and it appears that labeling is defined in the methods.

*Reviewer #2:*

The authors did a great job addressing our comments, especially with the additional validation work. My only concern is that some of the newly included datasets don't have human-labeled performance for comparison, hence making it hard to judge the actual performance of DeepEthogram. While I understand it is very time-consuming to obtain human labels, I think it will greatly improve the impact of the work if the model comparison can be bench-marked against ground truth. Especially it would be great to see the comparison to human label for the "Mouse-Social" and "Mouse-Homecage" datasets, which presumably represent a large proportion of use cases for DeepEthogram. Otherwise I think it looks good and I would support publication of this manuscript.

*Reviewer #3:*

The authors present a software solution (DeepEthogram) that performs supervised machine-learning analysis of behavior directly from raw videos files. DeepEthogram comes with a graphical user interface and performs behavior identification and quantification with high accuracy, requires modest amounts of pre-labeled training data, and demands manageable computational resources. It promises to be a versatile addition to the ever-growing compendium of open-source behavior analysis platforms and presents an interesting alternative to pose-estimation-based approaches for supervised behavior classification, under certain conditions.

The authors have generated a large amount of additional data and showcase the power of their approach in a wide variety of datasets including their own data as well as published datasets. DeepEthogram is clearly a powerful tool and the authors do an excellent job describing the advantages and disadvantages of their system and provide a nuanced comparison of point-tracking analyses vs. analyses based on raw videos (pixel data). Also their responses to the reviewers comments are very detailed, thoughtful and clear. The only major issue is that the raw videos used in this work (except the ones added during the revision) are – it appears – not accessible online. This problem must be solved, the videos are essential for reproducibility.

A minor caveat is that in order to compare DeepEthogram to existing supervised and unsupervised approaches, the authors have slightly skewed the odds in their favor by picking conditions that benefit their own algorithm. In the comparison with point-tracking data they use a low resolution top-view recording to label the paws of mice (which are obstructed most of the time from this angle). In the comparison with unsupervised clustering, they use the unsupervised approach for an application that it isn't really designed for (performed in response to reviewers requests). But the authors directly address these points in the text, and the comparisons are still valid and interesting and address the reviewers concerns.

---

## [Author Response]

Summary:Bohnslav et al., present a new toolkit and GUI for using video input to extract behavioral states (ethograms) using a set of established deep neural networks. They show their pipeline works on range of laboratory datasets, and provide metrics comparing network performance to humans. However, the reviewers all agreed there are several key revisions needed in order to support the main claims of the paper. These revolve around benchmarking, datasets, and a more careful handling of related work, limitations of such a software, and clarifying methods. We have collectively decided to send the individual reviews from each reviewer, and ask you address those (and perhaps combine where you see fit), but we urge you to focus in on the following points for your revision.

We thank the reviewers for their feedback and constructive suggestions. We have worked hard to incorporate all the suggestions of each reviewer and feel that the manuscript and software are much improved as a result.

Datasets:The reviewers each expressed concern over the simplicity of the datasets and the potentially limited scope of DeepEthogram in relation. For example, the authors claim these are difficult datasets, but in fact we feel they are not representative of the laboratory videos often collected: they have very static backgrounds, no animals have cables or other occluders. We would urge the authors to use other datasets, even those publically available, to more thoroughly benchmark performance in a broader collection of behaviors.

We thank the reviewers for the suggestion to add more datasets that cover a wider range of behavior settings. We have now added five datasets, including three publicly available datasets and two new datasets collected by us that specifically address these concerns. The new datasets we collected include a mouse in a homecage, which contains a complex background and occluders. The second dataset we added is a social interaction dataset that includes two mice and thus complex and dynamic settings. The three publicly available datasets we added feature commonly used behavioral paradigms: the open field test, the forced swim test, and the elevated plus maze. We thank Sturman et al., for making their videos and labels publicly available.

We now have nine datasets in total that span two species, multiple view angles (dorsal, ventral, side), individual and social settings, complex and static backgrounds, and multiple types of occluders (objects and other mice). We feel these datasets cover a wide range of common lab experiments and typical issues for behavior analysis. It is of course not possible to cover all types of videos, but we have made a sincere and substantial effort to demonstrate the efficacy of our software in a variety of settings.

We think the datasets that we include are representative of the laboratory videos often collected. To our knowledge, datasets for open field behavior are some of the most commonly collected in laboratory settings when one considers behavioral core facilities, biotech/pharmaceutical companies, as well as the fields of mouse disease models, mouse genetic mutations, and behavioral pharmacology. The same is true for the elevated plus maze, which is now included in the revision. A Pubmed search for “open field test” or “elevated plus maze” reveals more than 2500 papers for each in the past five years. We also note that all the datasets we include were not collected only for the purpose of testing our method; rather, they were collected for specific neuroscience research questions, indicating that they are at least reflective of the methods used in some fields, including the large fields of pain research, anxiety research, and autism research. The new additions of the datasets for the forced swim test, elevated plus maze, social interaction, and homecage behavior extend our tests of DeepEthogram to other commonly acquired videos. We agree that these videos do not cover all the possible types of videos that can be collected or that are common in a lab setting, but we are confident these videos cover a wide range of very commonly used behavioral tests and thus will be informative regarding the possible utility of DeepEthogram.

Benchmarking:While DeepEthogram could be an important tool to the growing toolbox of deep learning tools for behavior, we felt that there are sufficiently other options available that the authors should directly compare performance. While we do appreciate that comparing to the "gold standard" of human-labeled data, the real challenge with such datasets is even humans tend not to agree on a semantic label. Here, the authors only use two humans for ground-truth annotation, but there is a concern of an outlier. Typically, 3 humans are used to overcome a bit of this limitation. Therefore, we suggest carefully benchmarking against humans (i.e., increase the number of ground truth annotations), and please see the individual reviewer comments with specific questions related to other published/available code bases where you can directly compare your pipelines performance.

We thank the reviewers for the recommendation to add more benchmarking. We have extended our benchmarking analysis in two important ways. First, we have added a third human labeler. The results are qualitatively similar to before with the third human labeler added. We have also included three datasets from Sturman et al., each of which has three labelers. Second, we have added a comparison to other recent approaches in the field, including the use of keypoint tracking followed by supervised classification into behaviors and the use of unsupervised behavior analysis followed by post-hoc labeling of machine-generated clusters. We find that DeepEthogram performs better than these alternate methods. We think the addition of this new benchmarking combined with the new datasets will provide the reader with an accurate measure of DeepEthogram’s performance.

Methods, Relation to other packages, and Limitations:The reviewers raised several points where methods are unclear, or how an analysis was performed was not clear. In particular, we ask you to check reviewer #3's comments carefully regarding methods. Moreover, we think a more nuanced discussion about when to do some "pre-processing" (like pose estimation) would be beneficial vs. straight to an ethogram, and visa versa. In particular, it's worth nothing that often times having an intermediate bottleneck such as key points allows the user to more easily assess network performance (keypoints are a defined ground truth vs. semantic action labels).

We thank the reviewers for these suggestions. We have clarified the methods and analysis as suggested and detailed below. We have also extended the discussion of our method relative to others, including keypoint approaches, so that the advantages and disadvantages of our approach are clearer. These changes are described in detail below in response to individual reviewer comments.

In total, the reviews are certainly enthusiastic about this work, and do hope you find these suggestions helpful. We look forward to reading your revision.

We appreciate the time that the reviewers spent to provide detailed comments and constructive suggestions. The feedback has been valuable in helping us to improve the paper and the software.

Reviewer #1:Bohnslav et al., present a new tool to quantify behavior actions directly from video. I think this is a nice addition to the growing body of work using video to analyze behavior. The paper is well written, clear for a general audience, and takes nice innovations in computer vision into life sciences and presents a usable tool for the community. I have a few critical points that I believe need addressed before publication, mostly revolving around benchmarking, but overall I am enthusiastic about this work being in eLife.

We thank the reviewer for this positive feedback and for their recommendations that have helped us to improve our work.

In the following sections I highlight areas I believe can be improved upon.In relation to prior work: The authors should more explicitly state their contribution, and the field's contributions, to action recognition. The introduction mostly highlights limitations of unsupervised methods to perform behavioral analysis (which to note, produces the same outputs as this paper, i.e. an ethogram) and key point estimation alone, which of course is tackling a different problem. What I would like to see is a more careful consideration of the state-of-the-field in computer vision for action recognition, and clearly defining what the contribution is in this paper the cover letter alludes to them developing novel computer vision aspects of the package, but from the code base, etc, it seems they utilize (albeit nicely!) pre-existing works from ~3 years ago, begging the question if this is truly state-of-the-art performance. Moreover, and this does hurt novelty a bit, this is not the first report in life science of such a pipeline, so this should be clearly stated. I don't think it's required to compare this tool to every other tool available, but I do think discussing this in the introduction is of importance (but again, I am still enthusiastic for this being in eLife).

We have revised the introduction so that it now focuses on how DeepEthogram differs in design to previous methods used to classify animal behaviors. We try to provide the reader with an understanding of the differences in design and uses between methods that utilize unsupervised behavior classification, keypoints, and classification from raw pixel values. We have removed the text that focused on potential limitations of keypoint-based methods. We have also added sentences that highlight that the approach we take is built on existing methods that have addressed action detection in different settings, making it clear that our software is not built from scratch and is rather an extension and novel use case of earlier, pioneering work in a different field. We now directly state that our contribution is to extend and apply this previous work in the new setting of animal behavior and life sciences research.

"Our model operates directly on the raw pixel values of videos, and thus it is generally applicable to any case with video data and binary behavior labels and further does not require pre-specification of the body features of interest, such as keypoints on limbs or fitting the body with ellipses." -- please include references to the many other papers that do this as well. For example, please see:Data-driven analyses of motor impairments in animal models of neurological disorders https://journals.plos.org/plosbiology/article?id=10.1371/journal.pbio.3000516LSTM Self-Supervision for Detailed Behavior Analysis https://openaccess.thecvf.com/content_cvpr_2017/html/Brattoli_LSTM_Self-Supervision_for_CVPR_2017_paper.htmlFacial expressions of emotion states and their neuronal correlates in mice https://science.sciencemag.org/content/368/6486/89/tab-figures-data (not deep learning, but similar workflow; also extract features as the authors here do, and gets good performance using old CV techniques)Deep learning improves automated rodent behavior recognition within a specific experimental setup https://www.sciencedirect.com/science/article/pii/S0165027019303930

We have added new citations on this topic.

I think Figure 1A is a bit misleading, it's not clear anymore that manual annotation is the only or most common other alternative pipeline (discussed below in benchmarking)- many tools for automated analysis now exist, and tools like JAABA and MotionMapper have been around for 5+ years; I would rather like to see a comparison workflow to "unsupervised methods," and/or keypoint estimation + classification with supervised or unsupervised means.

We have now added direct comparisons to the methods mentioned by the reviewer. In Supplementary Figures 20-21, we have direct comparisons to methods for keypoint estimation + classification and for unsupervised classification followed by post-hoc labeling. We show that DeepEthogram performs better than these approaches, at least for the dataset and behaviors we tested. In the discussion, we now provide an extended section on cases in which we expect DeepEthogram to perform better than other methods and cases in which we anticipate other methods, such as keypoint estimation followed by classification, may outperform DeepEthogram. We hope these new comparisons and new text help readers understand the differences between methods as well as the situations in which they should choose one versus the other.

To our knowledge, manual annotation is still very common, and we do not think automated analysis tools, such as JAABA and MotionMapper, are more commonly used than manual annotation. We have many colleagues in our department, departments at neighboring institutions, and behavior core facilities that solve the problem of behavior classification using manual annotation. In fact, we have had a hard time finding any colleagues who use JAABA and MotionMapper as their main methods. We asked our colleagues why they do not use these automated approaches, and the common response has been that they have been tried but that they do not work sufficiently well or they are challenging to implement. Thus, to our knowledge, unlike DeepLabCut for keypoint estimation, these methods have not caught on for behavior classification. There are of course labs that use the existing automated approaches for this problem, but our informal survey of colleagues and the literature indicates that manual annotation is still very common.

For these reasons, we think that mentioning the idea of manual annotation is valid, and following the reviewer’s suggestion, we now focus our introduction on describing the approaches taken by recent automated pipelines, with an emphasis on how our pipeline differs in design.

Lastly, they do not discuss key papers in life science for automated animal ethogram building, such as Live Mouse Tracker (https://livemousetracker.org/), BORIS and related Behatrix. Not only should these important papers be discussed, they should likely be benchmarked if the authors want to claim SOTA (see below).

We thank the reviewer for these references, and we have added citations to these papers. However, we feel that each of these references solves a different problem than the one DeepEthogram is meant to address. Live Mouse Tracker is an interesting approach. However, it uses specialized hardware (depth sensors and RFID monitoring), which means their approach cannot be applied using typical video recording hardware, and is specific to mouse social behaviors. DeepEthogram is meant to use typical lab hardware and to be general-purpose across species and behaviors. BORIS appears to be an excellent open-source GUI for labeling behaviors in videos. However, it does not appear to include automatic labeling or machine learning. Behatrix is software for generating transition matrices, calculating statistics from behavioral sequences, and generating plots. When combined with BORIS, it is software for data analysis of human-labeled behavioral videos, but it does not appear to perform automatic labeling of videos. We have therefore cited these papers, but we do not see an easy way to benchmark our work against these approaches given that they seem to address different goals. Instead, we have added benchmarking of our work relative to a keypoint estimation + classification approach and in comparison to an unsupervised clustering with post-hoc labeling approach.

Datasets: the authors claim they picked challenging datasets ("Diverse and challenging datasets to test DeepEthogram"), but I don't believe this is the case and they should tone down this statement. In fact, the datasets presented are rather easy to solve (the camera is orthogonal to the animal, i.e. top or bottom, or the animal's position is fixed, and the background is homogeneous, rarely the case even for laboratory experiments). I would urge them to use another more challenging dataset, and/or discuss the limitations of this work. For example, a mouse in a standard home cage with bedding, nests, huts, etc would pose more challenges, or they could report their performance on the Kinect700 dataset, which they pretrain on anyhow.

We have removed our statement about the datasets being challenging in general. We have now clarified the specific challenges that the datasets present to machine learning. In particular, we emphasize the major imbalance in the frequency of classes. Many of the classes of interest are only present in ~1-5% of the frames. In fact, this was the hardest problem for us to solve.

Following the reviewer’s suggestions, we have added new datasets that present the challenges mentioned, including complex backgrounds, occluders, and social interactions. This includes a mouse in the homecage with bedding, objects, and a hut, as suggested. We find that DeepEthogram performs well on these more challenging datasets.

Benchmarking: The authors don't directly compare their work to that of other tools available in the field. Is their approach better (higher performance) than:(1) unsupervised learning methods

It is difficult to compare supervised learning methods to unsupervised ones. The objectives of the two approaches are different. In unsupervised methods, the goal is to identify behavior dimensions or clusters based on statistical regularities, whereas in supervised methods the goal is to label research-defined behaviors. The dimensions or clusters identified in unsupervised approaches are not designed to line up to researcher-defined behaviors of interest. In some cases, they may line up nicely, but in other cases it may be more challenging to identify a researcher’s behavior of interest in the output of an unsupervised algorithm.

Nevertheless, it is possible to try this approach and compare its performance to DeepEthogram (Supplementary Figure 21). We used DeepLabCut to identify keypoints followed by B-SOiD, an unsupervised method, to identify behavior clusters. We then labeled the B-SOiD clusters based on similarity to our pre-defined behaviors of interest. We verified that the DeepLabCut tracking worked well. Also, B-SOiD found statistically meaningful clusters that divided the DeepLabCut-based features into distinct parts of a low dimensional behavior space. However, in many cases, the correspondence of machine-generated clusters to researcher-defined behaviors of interest was poor.

We found that DeepEthogram’s performance was higher than that of the pipeline using unsupervised methods. For example, “rearing” was the behavior that the B-SOiD approach predicted most accurately, at 76% accuracy. In comparison, our worst model, DeepEthogram-fast, scored this behavior with 94% accuracy. The difference in performance was even greater for the other behaviors. Thus, a workflow in which one first generates unsupervised clusters and then treats them as labels for researcher-defined behaviors did not work as well as DeepEthogram.

(2) pose estimation plus classifiers or unsupervised clustering (as done in LEAP, DeepLabCut, B-SOiD, SIMBA, and the ETH DLC-Analyzer)

We benchmarked DeepEthogram relative to pose estimation plus classification using the classification architecture from Sturman et al., (ETH DLC-Analyzer) (Supplementary Figure 20). Note that we re-implemented the Sturman et al. approach for two reasons. First, our datasets can have more than one behavior per timepoint, which necessitates different activation and loss functions. Second, we wanted to ensure the same train/validation/test splits were used for both DeepEthogram and the ETH DLC-Analyzer classification architecture (the multilayer perceptron).

We found that DeepEthogram had higher accuracy and F1 scores than the ETH DLC-Analyzer on the dataset we tested. The two methods performed similarly on bout statistics, including the number of bouts and bout duration.

As mentioned above, we tested unsupervised clustering using a B-SOiD-based pipeline and found that DeepEthogram performed better.

We did not test other pipelines that are based on keypoint estimation plus classifiers, such as JAABA and SIMBA. The reason is that each pipeline takes a substantial effort to get set up and to verify that it is working properly. In a reasonable amount of time, it is therefore not feasible to test many different pipelines for benchmarking purposes. We therefore chose to focus on two pipelines with different approaches, namely the ETH DLC-Analyzer for keypoints + classification and the B-SOiD-based unsupervised approach.

(3) tools that automate ethogram building, such as JAABA, BORIS/Behatrix.

BORIS and Behatrix are excellent tools for annotated videos and creating statistics of behaviors. However, they do not provide automated labeling pipelines, to our knowledge. We therefore did not see how to benchmark our work relative to these approaches.

As mentioned above, it takes a great deal of time and effort to set up and validate each new pipeline. Given that JAABA is based on the idea of keypoint estimation followed by classification, we chose not to implement it and instead focused only on the ETH DLC-Analyzer for this approach. It would be ideal to rapidly benchmark DeepEthogram relative to many approaches, but we were unable to find a way to do so in a reasonable amount of time, especially because we need to ensure that each existing algorithm is operating properly.

Therefore, more results should be presented in relation to key works, and/or a more clear introduction on this topic should be presented.– For example, they claim it's hard to match the resulting clusters from unsupervised learning to their "label:" i.e., "their outputs can be challenging to match up to behaviors of interest in cases in which researchers have strong prior knowledge about the specific behaviors relevant to their experiments". But this is not really a fair statement; one can simply look at the clusters and post-hoc assign a label, which has been nicely done in MotionMapper, for example.

We have now supported this claim by trying out an unsupervised approach. We used the B-SOiD algorithm on DeepLabCut keypoints (Supplementary Figure 21). B-SOiD identified well separated behavior clusters. However, we found it challenging to line up these clusters to the researcher-defined behaviors of interest. In our Mouse-Openfield dataset, we found one cluster that looked like “rearing”, but it was not possible to find clusters that looked like the other behaviors. It is of course possible that in some cases the clusters and behaviors will line up nicely, but that was not apparent in our results. More generally, the goals of supervised and unsupervised clustering are not the same, so in principle there is no reason that post-hoc assignment of labels will work well. It is true that in theory this might work reasonably okay in some cases, but our results in Supplementary Figure 21 show that this approach does not work as well as DeepEthogram.

– In pose estimation, one gets an animal-centric lower dimensional representation, which can be mapped onto behavioral states (ethograms), or used for kinematic analysis if desired. However, there is the minimal number of key points needed to make a representation that can still be used for ethogram building. Is the raw-pixel input truly better than this for all behaviors? For example, on the simple datasets with black backgrounds presented in this work, the background pixels are useless, and don't hinder the analysis. However, if the background dynamically changed (camera is moving, or background changes (lighting, bedding etc)), then the classification task from raw pixels becomes much harder than the task of extracted keypoints to classification task. Therefore, I think the authors should do the following: (1) discuss this limitation clearly in the paper, and (2) if they want to claim their method has universally higher performance, they need to show this on both simple and more challenging data.

We have now added a paragraph in the discussion that mentions cases in which DeepEthogram is expected to perform poorly. This includes cases with, for example, moving cameras and changing lighting. We are not aware that these scenarios are common in laboratory settings, but nevertheless we mention conditions that will help DeepEthogram’s performance, including fixed camera angles, fixed lighting, and fixed backgrounds. In addition, we have added two datasets that have more complex or changing backgrounds. In the Mouse-Homecage dataset, there are bedding, objects, and a hut, which provide occluders and a complex background. In the Sturman-FST, the background is dynamic due to water movement as the animal swims. We also now include a social interaction dataset, which adds a different use case. In all these cases, DeepEthogram performs well.

We have expanded our discussion of the settings in which DeepEthogram may excel and those in which other methods, like keypoint-based methods, might be advantageous. In addition, we now mention that DeepEthogram is intended as a general framework that can be used without extensive fine tuning and customization for each experiment. As with all software focused on general purpose uses, we anticipate that DeepEthogram will perform worse than solutions that are custom for a problem at hand, such as custom pipelines for moving cameras or specific pose estimations. We now note this tradeoff between general purpose and customized solutions and discuss where DeepEthogram falls on this spectrum.

Moveover, the authors discuss 4 limitations of other approaches, but do not address them in their work, i.e.:– "First, the user must specify which features are key to the behavior (e.g. body position or limb position), but many behaviors are whole-body activities that could best be classified by full body data." – can they show an example where this is true? It seems from their data each action could be easily defined by kinematic actions of specific body parts a priori.

We have removed this sentence. Instead, in the introduction, we now focus on the design differences and general goals of the different approaches with less emphasis on potential limitations. In the discussion, we present a balanced discussion of cases in which DeepEthogram might be a good method to choose and cases in which other methods, such as those with keypoint estimation, might be better.

– "Second, errors that occur in tracking these features in a video will result in poor input data to the classification of behaviors, potentially decreasing the accuracy of labeling." – but is poor video quality not an issue for your classification method? The apple-to-apple comparison here is having corrupted video data as "bad" inputs – of course any method will suffer with bad data input.

We have removed this sentence.

– "Third, users might have to perform a pre-processing step between their raw videos and the input to these algorithms, increasing pipeline complexity and researcher time." – can they elaborate here? What preprocessing is needed for pose estimation, that is not needed for this, for example? (Both require manual labor, and given the time estimates, DEG takes longer to label than key point estimation due to the human needing to be able to look at video clips (see their own discussion)).

We have re-phrased this point in the introduction. The main point we wish to make is that a pipeline for behavior classification based on keypoints + classification requires two steps: keypoint estimation followed by behavior classification. This requires two stages of manual annotation, one for labeling the keypoints and one for labeling the behaviors for classification. If one wants to get frame-by-frame predictions for this type of pipeline, then one must have frame-by-frame labeling of training videos. Instead, DeepEthogram only requires the latter step. As the reviewer notes, the frame-by-frame behavior labeling is indeed the more time-consuming step, but it is required for both types of pipelines.

– "Fourth, the selection of features often needs to be tailored to specific video angles, behaviors (e.g. social behaviors vs. individual mice), species, and maze environments, making the analysis pipelines often specialized to specific experiments." – this is absolutely true, but also a limitation to the authors work, where the classifiers are tailored, the video should be a fixed perspective, background static, etc. So again I don't see this as a major limitation that makes pose estimation a truly invalid option.

We have removed this sentence from the introduction and added this point to the discussion where we talk about the advantages and disadvantages of DeepEthogram. We disagree that DeepEthogram requires tailored classifiers. We have used the same model architecture for all nine datasets that involve different backgrounds, species, behaviors, and camera angles. It is true that DeepEthogram might not work in some cases, such as with moving cameras, and we now highlight the cases in which we expect DeepEthogram will fail. In general, though, we have used the same model throughout our study for a range of datasets.

Benchmarking and Evaluation:– "We evaluated how many video frames a user must label to train a reliable model. We selected 1, 2, 4, 8, 12, or 16 random videos for training and used the remaining videos for evaluation. We only required that each training set had at least one frame of each behavior. We trained the feature extractors, extracted the features, and trained the sequence models for each split of the data." – it is not clear how many FRAMES are used here; please state in # of frames in Figure 5 and in the text (not just video #'s).Related: "Combining all these models together, we found that the model performed with more than 90% accuracy when trained with only 80 example frames" This again is a bit misleading, as the user wants to know the total # of frames needed for your data, i.e. in this case this means that a human needs to annotate at least 80-100 frames per behavior, which for 5 states is ~500 frames; this should be made more explicit.

We now report the number of frames to Figure 6 as suggested by the reviewer.

We choose to report the number of example frames because this is what is most relevant for the model, but we realize this is not the most relevant for the user. The number of actual frames (not just examples) varies depending on how frequent the behavior is. If 100 example frames are needed for a behavior, that could mean the user has to label 200 frames if the behavior occurs 50% of the time or 10,000 frames if the behavior occurs 1% of the time. We have now added sentences in the text to help the reader with this conversion. We do not think it is helpful to report the number of actual frames in the figure because then it is impossible for the user to extrapolate to their own use cases. Instead, if they know the number of example frames required and how frequent their behavior of interest is, they can calculate for themselves how many frames will need to be labeled. The new sentences we added help clarify this point and will help the reader calculate for their own cases.

– "We note that here we used DEG-fast due to the large numbers of splits of the data, and we anticipate that the more complex DEG-medium and DEG-slow models might even require less training data." – this would go against common assumptions in deep learning; the deeper the models, the more prone to overfitting you are with less data. Please revise, or show the data that this statement is true.

We have removed this sentence.

– "Human-human performance was calculated by defining one labeler as the "ground truth" and the other labeler as "predictions", and then computing the same performance metrics as for DEG. " – this is a rather unconventional way to measure ground truth performance of humans. Shouldn't the humans be directly compared for % agreement and % disagreement on the behavioral state? (i.e., add a plot to the row that starts with G in figure 3).

We agree that this description was confusing. We have now clarified in the text that the percent agreement between human labelers is identical to the accuracy measure we report.

To note, this is a limitation of such approaches, compared to pose-estimation, as humans can disagree on what a "behavior" is, whereas key points have a true GT, so I think it's a really important point that the authors address this head on (thanks!), and could be expanded in the discussion. Notably, MARS puts a lot of effort into measuring human performance, and perhaps this could be discussed in the context of this work as well.

We appreciate that the reviewer acknowledged the hard work of labeling many videos with multiple human labelers. We have now extended this effort to include a third human labeler for the Mouse-Ventral1 dataset. We have also added three publicly available datasets that each have three human labelers.

Reviewer #2:It was a pleasure reviewing the methodological manuscript describing DeepEthogram, a software developed for supervised behavioral classification. The software is intended to allow users to automate classification/quantification of complex animal behaviors using a set of supervised deep learning algorithms. The manuscript combines a few state-of-art neural networks into a pipeline to solve the problem of behavior classification in a supervised way. The pipeline uses well-established CNN to extract spatial features from each still frame of the videos that best predicts the user-provided behavior labels. In parallel, optical flow for each frame is estimated through another CNN, providing information about the "instantaneous" movement for each pixel. The optical flow "image" is then passed to another feature extractor that has the same architecture as the spatial feature extractor, and meaningful patterns of pixel-wise movements are extracted. Finally, the spatial feature stream and the optical flow feature stream are combined and fed into a temporal Gaussian mixture CNN, which can pool together information across long periods of time, mimicking human classifiers who can use previous frames to inform classification of behavior in current frame. The resulting pipeline provides a supervised classification algorithm that can directly operate on raw videos, while maintaining a relatively small computational demands on the hardware.

Thank you for the positive feedback.

While I think something like DeepEthogram is needed in the field, I think the authors could do substantially more to validate that DeepEthogram is the ticket. In particular, I find the range of datasets validated in the manuscript poorly representative of the range of behavioral tracking circumstances that researchers routinely face. First, in all exemplar datasets, the animals are recorded in a completely empty environment. The animals are not interacting with any objects as they might in routine behavioral tests; there are no cables attached to them (which is routine for optogenetic studies, physiological recording studies, etc); they are alone (Can DeepEthogram classify social behaviors? the github page lists this as a typical use case); there isn't even cage bedding.

We thank the reviewer for this suggestion. We have now added five new datasets that address many of these concerns. We have added a dataset in which the mouse is in its homecage. In these videos, there is bedding, and the mouse interacts with multiple objects and is occluded by these objects and a hut. Second, we added a dataset for the elevated plus maze, in which the mouse interacts with the maze, exhibiting stretches and dips over the side of the maze. Third, we added a social interaction dataset with two mice that specifically tests the model’s performance on social behaviors. Fourth, we added a dataset for the forced swim test in which the mouse is swimming in a small pool. These datasets therefore have more complex backgrounds (e.g. bedding, objects, multiple mice), dynamic backgrounds (moving water in the forced swim test), and social interactions. Unfortunately, we did not have access to a dataset with cables, such as for optogenetics or physiology recordings, but we note that the Mouse-Homecage dataset includes occluders. We now have nine datasets in total, and DeepEthogram performed well across all these datasets.

We feel that the datasets we used are representative of videos that are common in many fields. For example, the open field assay and elevated plus maze are common assays that are typical in behavior core facilities, biotech/pharma companies, and many academic research labs. A Pubmed search for “open field test” or “elevated plus maze” each returns over 2500 papers in the past five years. These types of videos are not necessarily common in the fields of neurophysiology or systems neuroscience, but they are commonly used for phenotyping mutant mice, tests of mouse models of disease, and treatments of disease. However, we agree entirely that we have not tested all common types of behavior videos, including some mentioned by the reviewer. Unfortunately, each dataset takes a long time to test due to extensive labeling by humans for “ground-truth” data, and we did not have access to all types of datasets. In the revision, we made a sincere and substantial effort to extend the datasets tested. In addition, we have added more discussion of the cases in which we expect DeepEthogram to work well and the cases in which we expect it will be worse than other methods. We hope this discussion points readers to the best method for their specific use case.

The authors also tout the time saving benefits of using deep ethogram. However, with their best performing implementation (DEG slow), with a state of the art computer, with a small video (256 x 256 pixels, width by height), the software runs at 15 frames per second (nearly 1/2 the speed of the raw video). My intuition is that this is on the slow side, given that many behaviors can be scored by human observers in near real time if the observer is using anything but a stopwatch. It would be nice to see benchmarks on larger videos that more accurately reflect the range of acquisition frames. If it is necessary for users to dramatically downsample videos, this should be made clear.

We have now clarified in the text that the time savings are in terms of person-hours instead of actual time. A significant advantage is that DeepEthogram can run in the background on a computer with little to no human time (after it is trained), whereas manual labeling of each video continues to require human time with each video. In the case of the DeepEthogram-slow model, it is true that humans might be able to do it faster, but this would take human time, whereas the model can run while the researchers do other things. We think this is a critical difference and have highlighted this difference more clearly. Furthermore, we have made engineering improvements that have sped up inference time by nearly 300%.

We have added a table that measures inference time across the range of image sizes we used in this study. We note that all comparable methods in temporal action localization down-sample videos in resolution. For example, video classification networks commonly use 224 x 224 pixels as input. DeepEthogram works similarly for longer acquisitions (more frames per video). We had access to videos collected for specific neuroscience research questions and were restricted to these acquisition times, but the software will work similarly with all acquisition times.

Specific comments:– It would be nice to see if DeepEthogram is capable of accurately scoring a behavior across a range of backgrounds. For example, if the model is trained on a sideview recording of an animal grooming in its cage, can it accurately score an animal in an open field doing the same from an overhead view, or a side view? If the authors provided guidance on such issues to the reader this would be helpful.

We have tested DeepEthogram in a range of different camera angles and backgrounds across the nine datasets we now include. Our results show that the model can work across this diversity of set ups. However, an important point is that it is unlikely that the model trained on a side view, for example, will work for videos recorded from above. Instead, the user would be required to label the top-view and side-view videos separately and train different models for these cases. We do not think this is a major limitation because often experimenters decide on a camera location and use that for an entire set of experiments. However, this is an important point for the user to understand. We have therefore added a sentence to the discussion describing this point, along with extended points about the cases in which DeepEthogram is expected to excel and when it might not work well.

– The authors should highlight that human scoring greatly outperforms DEG on a range of behaviors when comparing the individual F1 scores in Figure 3. Why aren't there any statistics for these comparisons?

Thank you for this recommendation. We have added statistics for comparing the F1 scores in Figure 3. These analyses reveal that human scoring outperforms DeepEthogram on some behaviors, but the performance of DeepEthogram and humans is statistically indistinguishable on the majority of behaviors tested. We have revised the main text to make it clear that human labelers do better than DeepEthogram in some cases.

We have also compared DeepEthogram to human performance on other measures of behavior, in particular the statistics of behavior bouts (duration and frequency). We expect that these metrics may be more commonly used by the end-user. In the case of bout statistics, DeepEthogram’s performance is worse than human performance at the level of single videos (Figure 5A-B). However, when averaged across videos, the performance of the model and humans is statistically indistinguishable (Figure 5C-E). The reason appears to be that the model’s predictions are more variable on a single-video level, but after averaging across videos, the noise is averaged out to reveal a similar mean to what is obtained by human labelers. These results can also be seen in Figure 4A-C, in which the difference between the bout statistics for the model and labels on which it was trained (Human 1) are similar in magnitude to the difference between human labelers.

We thank the reviewer again for this suggestion. The statistics along with portions of Figure 4-5 are new in response to this suggestion.

– Some of the F1 scores for individual behaviors look very low (~0.5). It would be nice to know what chance performance is in these situations and if the software is performing above chance.

We thank the reviewer for the suggestion to add chance level performance. We have added chance-level F1 scores by performing a shuffle of the actual human labels relative to the video frames. For nearly all behaviors, DeepEthogram’s performance is significantly higher than chance level performance by a substantial margin. We think the addition of the chance level performance helps to interpret the F1 scores, which are not always intuitive to understand in terms of how good or bad they are. We also note that the F1 score is a demanding metric. For example, if the model correctly predicts the presence of a grooming bout, but misses the onset and offset times by several frames, the F1 score will be substantially reduced. For this reason, we also report the bout statistics in Figures 4-5, which are closer to the metrics that end users might want to report and are easier to interpret.

– I find it hard to understand the size of the data sets used in the analyses. For instance, what is 'one split of the data', referenced in Figure 3? Moreover, the authors state "We selected 1, 2, 4, 8, 12, or 16 random videos for training and used the remaining videos for evaluation" I have no idea what this means. What is the length and fps of the video?

We thank the reviewer for this comment and agree that it was not clear in the original submission. We have updated the text and figure legends to clarify this point. A split of the data refers to a random splitting of the data into training, validation, and testing sets. We now describe this in the procedures for the model set up in the main text and clarify the meaning of “splits” when it is used later in the Results section. Videos refer to a single recorded behavior video. Videos can differ in terms of their duration and frame number. However, it is a commonly used division of the data for experimental recordings. We therefore include two sets of plots in Figure 6. One is based on the number of videos, and the other is based on the number of frames with positive examples of the behavior. The former might be more intuitive to some readers because it is a common division of the data, but it might be challenging to understand given that videos can differ in duration and frame rate. Instead, reporting the number of positive examples provides an exact metric that can be applied to any case once one calculates the frame rate and frequency of their behavior of interest. We have now included text that helps readers with this conversion.

– Are overall F1 scores in Figure 3 computed as the mean of the individual scores on each component F1 score, or the combination of all behaviors (such that it weights high frequency behaviors)? It's also difficult to understand what the individual points in Figure 4 (a-c) correspond to.

We have clarified the meaning of these values in text and figure legends. The overall F1 score (Figure 3A-B) reports the “combination” of all behaviors. We report each behavior individually below (Figure 3C-K). In the old Figure 4A-C, each dot represented a single behavior (e.g. “face grooming”). We have clarified the meaning of each marker in each figure in the legends.

– The use of the names Mouse-1, Mouse-2 etc for experiments are confusing because it can appear that these experiments are only looking at single mice. I would change the nomenclature to highlight that these reflect experiments with multiple mice.

We thank the reviewer for this suggestion. We have now updated the naming of each dataset to provide a more descriptive title.

– It is not clear why the image has to be averaged across RGB channels and then replicated 20 times for the spatial stream. The author mentioned "To leverage ImageNet weights with this new number of channels", and I assume this means the input to the spatial stream has to have same shape (number of weights) as the input to the flow stream. However why this is the case is not clear, especially considering two feature extractor networks are independently trained for spatial and flow streams. Lastly this might raise the question of whether there will be valuable information in the RGB channels separately that will be lost from the averaging operation (for example, certain part of an animal's body has different color than others but is equal-luminous).

This averaging is only performed once, when loading ImageNet weights into our models. This occurs before training begins. The user never needs to do this because they will use our models that are pretrained on Kinetics700, rather than ImageNet.

For clarification, the ImageNet weights have 3 channels; red, green, and blue. Our flow classifier model has 20 channels; δ-X and δ-Y each for 10 frames. Therefore, we average across Red, Green, and Blue channels in the ImageNet weights and replicate this value 20 times. However, these weights are then changed by the training procedure, so there is no information lost due to averaging. We have updated the text to clarify this.

– It is not intuitive why simple average pooling is sufficient for fusing the spatial and flow streams. It can be speculated that classification of certain behavior will benefit much more from optical flow features while other behaviors benefits from still image features. I'm curious to see whether an additional layer at the fusing stage that has behavior-specific weights could improve performance.

We thank the reviewer for this interesting suggestion. In theory, this weighting of image features and flow features could happen implicitly in the fully connected layer prior to average pooling. Specifically, the biases on the units in the fully connected layer are per-behavior parameters for both the spatial and the flow stream that will affect which one is weighted more highly in the average fusion. We have also experimented with concatenation pooling, in which we fuse the two feature vectors together before one fully connected layer. However, this adds parameters and did not improve performance in our experiments. We have not experimented with more elaborate fusion schemes.

– Since computational demands is one of the major concern in this article, I'm wondering whether exploiting the sparse nature of the input images would further improve the performance of the algorithm. Often times the animal of interests only occupies a small number of pixels in the raw images, and some simple thresholding of the images, or even user-defined masking of the images, together with use of sparse data backends and operations should in theory significantly reduce the computational demands for both the spatial and flow feature extractor networks.

This is an interesting and insightful comment. For most of our datasets, the relevant information is indeed sparse, and the models could be made faster and perhaps more accurate if we could focus on the animal themselves. The difficulty is in developing a method that automatically focuses on the relevant pixels across datasets. For some datasets, thresholding would suffice, but for others more elaborate strategies would be required. Because we wanted DeepEthogram to be a general-purpose solution without requiring coding by the end user, we did not adopt any of these strategies. To make a general-purpose solution to focus on the relevant pixels, we could make a second version of DeepEthogram that implements models for spatiotemporal action detection, in which the user specifies one or more bounding boxes, each with their own behavior label. However, this would be a significant extension that we did not perform here.

Reviewer #3:The paper by Bohnslav et al., presents a software tool that integrates a supervised machine learning algorithm for detecting and quantifying behavior directly from raw video input. The manuscript is well-written, the results are clear. Strengths and weaknesses of the approach are discussed and the work is appropriately placed in the bigger context of ongoing research in the field. The algorithms demonstrate high performance and reach human-level accuracy for behavior recognition. The classifiers are embedded in an excellent user-friendly interface that eliminates the need of any programming skills on the end of the user. Labeled datasets can even be imported. We suggest additional analyses to strengthen the manuscript.

We thank the reviewer for this positive feedback and for their constructive suggestions.

Concerns:1) Although the presented metrics for accuracy and F1 are state of the art it would be useful to also report absolute numbers for some of the scored behaviors for each trial, because most behavioral neuroscience studies actually report behavior in absolute numbers and/or duration of individual behaviors (rears, face grooms, etc.). Correlation of human and DEG data should also be presented on this level. This will speak to many readers more directly than the accuracy and F1 statistics. For this, we would like to see a leave-one-out cross-validation or a k-fold cross-validation (ensure that each trial ends up exactly once in a cross validation set) that enables a final per-trial readout. This can be done with only one of the DEG types (e.g "fast"). The current randomization approach of 60/20/20% (train/validate/test) with a n of 3 repeats is insufficient, since it a) allows per-trial data for at most 60% of all files and b) is susceptible to artefacts due to random splits (i.e one abnormal trial can be over or under represented in the cross validation sets).

We thank the reviewer for these suggestions. We have added multiple new analyses and plots that focus on the types of data mentioned. In the new Figure 4A-C, we show measures of bout statistics, with dots reporting values for individual videos in the test set. In addition, in Figure 5A-B and Figure 5F-H, we show measures of bout statistics benchmarked against human-human agreement. We agree with the reviewer that these bout statistics are likely more interpretable to most readers and likely closer to the metrics that end users will report and care about in their own studies. The remainders of Figures 4 and 5 provide summaries of these bout statistics for each behavior in each dataset. In these summaries, each dot represents a single behavior with values averaged across train/validation/test splits of the data, corresponding to a measure similar to averages across videos. We used these summaries to report the large number of comparisons in a compact manner.

We did not employ leave-one-out cross validation because we base our model’s thresholds and learning rate changes on the validation set. We therefore require three portions of the data. K-fold cross-validation is more complex with a split into three groups, and we also require that each of the train, validation, and test sets contains at least one example from each behavior, which might not happen with K-fold cross validation. However, to address your concerns, we repeated the random splitting 5 times for all datasets (except Sturman-EPM, which only has 3 videos with all behaviors). We found that empirically 68% of all videos are represented at least once in the test set. While this does add noise to our estimates of performance, it is not expected to add bias.

2) In line with comment 1) we propose to update Figure 4, which at the moment uses summed up data from multiple trials. We would rather like to see each trial represented by a single data-point in this figure (#bouts/#frames by behavior). As alternative to individual scatterplots, correlation-matrix-heatmaps could be used to compare different raters.

In response to this suggestion, we have completely revised and extended the old Figure 4, dividing it into two separate main figures (Figures 4 and 5). In Figure 4, we now report bout statistics first on an individual video basis (Figure 4A-C) and then averaged across train/validation/test splits of the data for each behavior as a summary (Figure 4D-F). In the new Figure 5, we benchmark the performance of the model on bout statistics by comparison to human performance for the datasets for which we have multiple human labelers. In the new Figure 5A-B, we examined this benchmarking on the basis of single videos. Then in Figure 5C-E, we report the averages across data splits for each behavior as a summary across all datasets and all behaviors. This analysis led to an interesting finding. On an individual video basis, humans are more accurate than DeepEthogram (Figure 5A-B). However, on averages for each behavior, the accuracy of the model and humans is statistically indistinguishable (Figure 5C-E). This occurs because the model predictions are noisier than humans on single videos, but when averaging across multiple videos, this noise is averaged out, resulting in similar values between the model and humans.

3) Direct benchmarking against existing datasets is necessary. With many algorithms being published these days, it is important to pick additional (published) datasets and test how well the classifiers perform on those videos. Their software package already allows import of labeled datasets, some are available online. For example, how well can DeepEthogram score…a. grooming in comparison to Hsu and Yttri (REF #17) or van den Boom et al., (2017, J Neurosci Methods)b. rearing in comparison to Sturman et al., (REF #21)c. social interactions compared to (Segalin et al., (REF #7) or Nilsson et al., (REF #19))

We thank the reviewer for this recommendation. In response, we have done two things. First, we have tested DeepEthogram on publicly available datasets, in particular from Sturman et al. We greatly appreciate that Sturman et al., made their datasets and labels publicly available. We plan to do the same upon publication of this paper. This allowed us to confirm that DeepEthogram performs well on other datasets collected by different groups.

Second, we have benchmarked our method against two other approaches. First, we tried the method of Hsu and Yttri (B-SOiD) on the Mouse-Openfield dataset (Supplementary Figure 21). Specifically, we performed DeepLabCut to identify keypoints and then used B-SOiD on these keypoints to separate behaviors into clusters. The method worked well and created separable clusters in a low dimensional behavior space. We then tried to line up these clusters with researcher-defined behaviors of interest. However, it was challenging to find a good correspondence. As a result, DeepEthogram substantially outperformed this B-SOiD-based method. We think this is not surprising given that B-SOiD is designed for unsupervised behavior classification, whereas the problem we are trying to solve is supervised classification. While it is possible that clusters emerging from unsupervised methods may line up nicely with researcher-defined behaviors of interest, this does not have to happen. It therefore makes sense to utilize a supervised method, like DeepEthogram, for the problem of identifying pre-defined behaviors of interest. We now discuss this point and present these results in depth in the text.

We also compared the performance of DeepEthogram and the ETH DLC-Analyzer method from Sturman et al. Specifically, we used DeepLabCut to track body parts in our Mouse-Openfield dataset and then used the architecture from Sturman et al. to classify behaviors (the multilayer perceptron). Note that we had to re-implement the code for the Sturman et al. approach for two reasons. First, our datasets can have more than one behavior per timepoint, which necessitates different activation and loss functions. Second, we wanted to ensure the same train/validation/test splits were used for both DeepEthogram and the ETH DLC-Analyzer classification architecture (the multilayer perceptron). We also included several training tricks in this architecture based on things we had learned while developing DeepEthogram, aiming to make this the fairest comparison possible. We found that DeepEthogram had significantly better performance in terms of accuracy and F1 on some behaviors in this dataset. The accuracy of the bout statistics was similar between the two modeling approaches. We conclude that DeepEthogram can perform better than other methods, at least in some cases and on some metrics, and hope that these new benchmarking comparisons help lend credibility to the software.

We did not perform further benchmarking against other approaches. Such benchmarking takes substantial effort because it takes a lot of work to set up and validate each method. We agree with the reviewer that it would be valuable to have more extensive comparisons of methods, but in terms of timing and the multiple directions asked for by the reviewers (e.g. also new datasets), it was not feasible to try more methods. In addition, not all the datasets mentioned were available when we checked. For example, last we checked, the dataset for Segalin et al., is not publicly available, and the authors told us they would release the data when the paper is published. Also, to our knowledge, the method of Segalin et al., is specific to social interactions between a white and a black mouse, whereas the social interaction dataset that is available to us has two black mice.

4) In the discussion on page 19 the authors state: "Subsequently, tens to hundreds to thousands of movies could be analyzed, across projects and labs, without additional user-time, which would normally cost additionally hundreds to thousands of hours of time from researchers." This sentence suggests that a network trained on the e.g. the open field test in one lab can be transferred across labs. This key issue of "model transferability" should be tested. E.g. the authors could use the classifier from mouse#3 and test is on another available top-view recording dataset recorded in a different lab with different open-field setup (datasets are available online, e.g. REF #21).

The original intent of this comment was that videos with similar recording conditions could be tested, such as in behavior core facilities with standardized set ups. We ran a simple test of generalization by training DeepEthogram on our Mouse-Openfield dataset and testing it on the Sturman-OFT dataset. We found that the model did not generalize well and speculate that this could be due to differences in resolution, background, or contrast. We have therefore removed this claim and added a note that DeepEthogram likely needs to be retrained if major aspects of the recordings change.

5) Figure 5D/E: Trendline is questionable, we would advise to fit a sigmoid trendline, not an arbitrarily high order polynomial. Linear trend lines (such as shown in Figure 4) should include R^2^ values on the plot or in the legend.

We apologize for the lack of clarity in our descriptions. The lines mentioned are not trendlines. The line in the old Figure 5D-E is simply a moving average that is shown to aid visualization. Also, the lines in the scatterplots of the old Figure 4 are not linear trend lines; rather, they are unity lines to help the reader see if the points fall above or below unity. We now explain the meaning of these lines in the figure legends. We have also added correlation coefficients to the relevant plots.

6) In the discussion, the authors do a very good job highlighting the limitations and advantages of their approach. The following limitations should however be expanded:a. pose-estimation-based approaches (e.g. DLC) are going to be able to track multiple animals at the same time (thus allowing e.g. better read-outs of social interaction). It seems this feature cannot be incorporated in DeepEthogram.

We thank the reviewer for noting our attempts to provide a balanced assessment of DeepEthogram, including its limitations. We have now expanded the discussion to even more clearly mention the cases in which DeepEthogram might not work well. We specifically mention that DeepEthogram cannot track multiple animals in its current implementation. We hope the new discussion material will help readers to identify when and when not to use DeepEthogram and when other methods might be preferable.

b. Having only 2 human raters is a weakness that should briefly be addressed. Triplicates are useful for assessing outlier values, this could be mentioned in light of the fact that the F1 score of DeepEthogram occasionally outperforms the human raters (e.g. Figure 3C,E).

We have added a third labeler for our Mouse-Ventral1 dataset. We have also included results from Sturman et al., each of which have 3 human labelers. Unfortunately, we were not able to add a third labeler for all datasets. For those in which we included a third labeler, the results are similar to using only two labelers. We think this is because all of our labelers are experts who spent a lot of time carefully labeling videos.

c. Traditional tracking measures such as time in zone, distance moved and velocity cannot be extracted with this approach. These parameters are still very informative and require a separate analysis with different tools (creating additional work).

We agree these measures are informative and not currently output from DeepEthogram. We have added a sentence in the discussion noting that DeepEthogram does not provide tracking details.

d. The authors are correct that the additional time required for behavior analysis (due to the computationally demanding algorithms) is irrelevant for most labs. However, they should add (1) that the current system will not be able to perform behavior recognition in real time (thus preventing the use of closed-loop systems, which packages such as DLC have made possible) and (2) that the speed they discuss on page 16 is based on an advanced computer system (GPU, RAM) and will not be possible with a standard lab computer (or provide an estimate how long training would require if it is possible).

We have added sentences in the discussion and results that state that the DeepEthogram is not suitable for closed-loop experiments and that the speed of inference depends on the computer system and might be substantially slower for typical desktop computers. We have also updated the inference evaluation by using a 1080Ti GPU, which is well within the budget of most labs. Furthermore, during the revision process, we significantly improved inference speed, so that DeepEthogram-fast runs inference at more than 150 frames per second on a 1080Ti GPU.

[Editors' note: further revisions were suggested prior to acceptance, as described below.]

The reviewers all felt the manuscript was improved, and thank the authors for the additional datasets and analysis. We would just like to see two items before the publication is accepted fully.(1) Both reviewer #1 and #2 note the new data is great, but lacks human ground truth. Both for comparison, and releasing the data for others to benchmark on, it would be please include the data. We also understand that obtaining ground truth from 3 persons is a large time commitment, but even if there is one person, this data should be included for all datasets shown in Figure 3.

We have performed addition labeling to get human levels of performance for benchmarking. For eight datasets, we now have labels from multiple researchers, which allows calculation of human-level performance. We have updated all the relevant figures, e.g. Figure 3. The results are consistent with those previously reported, in which DeepEthogram’s performance is similar to human-level performance in general. The only dataset for which we did not add another set of human labels is the Fly dataset. This dataset has more than 3 million video frames, and the initial labeling took months of work to label. Note that this dataset still has one set of human labels, which we use to train and test the model, but we are unable to plot a human-level of performance with only one set of human labels. We feel that comparing DeepEthogram’s performance to human-level performance for 8 datasets is sufficient and goes beyond, or is at least comparable to, the number of datasets typically reported in papers of this type.

(2) Please include links for the raw videos used in this work; it is essential for others to benchmark and use to validate the algorithm presented here (see Reviewer 3: "raw videos used in this work (except the ones added during the revision) are – it appears – not accessible online").

We have posted a link to the raw videos and the human labels on the project website (https://github.com/jbohnslav/deepethogram). We have noted in the paper’s main text and methods that the videos and annotations are publicly available at the project website.

Lastly, reviewer 3 notes that perhaps, still, some use-cases are best suited for DeepEthogram, while others more for pose-estimation plus other tools, but this of course cannot be exhaustively demonstrated here; at your discretion you might want to address in the discussion, but we leave that up to your judgement.

We agree with Reviewer 3 on this point. After reviewing the manuscript text, we feel that we have already included a balanced and extensive discussion of the use-cases that are best suited for DeepEthogram versus pose estimation methods. We feel that the relevant points are mentioned and, for brevity, do not feel it would be helpful to add even more explanations as these points are already addressed extensively in both the Results section and the Discussion section.

Reviewer #1:I thank the authors for the revisions and clarifications, and I think the manuscript is much improved. Plus, the new datasets and comparisons to B-iOD and R-Analyzer (Sturman) are a good additions.

Thank you for the positive feedback and the constructive comments. Your input has greatly improved the manuscript.

One note is that is not clear which datasets have ground truth data; namely, in the results 5 datasets they use for testing are introduced:"Mouse-Ventral1""Mouse-Ventral2""Mouse-Openfield""Mouse-Homecage""Mouse-Social"plus three datasets from published work by Sturman et al.,and "fly"Then it states that all datasets were labeled; yet, Figure 3 has no ground truth for "Mouse-Ventral2" , "Mouse-Homecage" , "Mouse-Social" or 'Fly" -- please correct and include the ground truth. I do see that is says that only a subset of each of the 2 datasets in Figure 3 are labeled with 3 humans, but minimally then the rest (1 human?) should be included in Figure 3 (and be made open source for future benchmarking).It appears from the discussion this was done (i.e., at least 1 human, as this is of course required for the supervised algorithm too):"In our hands, it took approximately 1-3 hours for an expert researcher to label five behaviors in a ten-minute movie from the Mouse-Openfield dataset" and it appears that labeling is defined in the methods.

We apologize for the confusion on this point. All datasets had at least one set of human labels because this is required to train and test the model. To obtain a benchmark of human-level performance, multiple human labels are required so that the labels of one researcher can be compared to the labels of a second researcher. We have now performed additional annotation. Now 8 datasets have human-level performance reported (see updated Figure 3). The results show that the model achieves human-level performance in many cases.

The only dataset for which we do not have multiple human labelers is the Fly dataset. The Fly dataset has more than 3 million timepoints, and it took a graduate student over a month of full-time work to obtain the first set of labels. Given the size of this dataset, we did not add another set of labels.

We feel that having 8 datasets with human-level performance exceeds, or at least matches, what is typical for benchmarking of methods similar to DeepEthogram. We feel this number of datasets is sufficient to give the reader a good understanding of the model’s performance.

We have now made all the datasets and human labels available on the project website and hope they will be of use to other researchers.

Reviewer #2:The authors did a great job addressing our comments, especially with the additional validation work. My only concern is that some of the newly included datasets don't have human-labeled performance for comparison, hence making it hard to judge the actual performance of DeepEthogram. While I understand it is very time-consuming to obtain human labels, I think it will greatly improve the impact of the work if the model comparison can be bench-marked against ground truth. Especially it would be great to see the comparison to human label for the "Mouse-Social" and "Mouse-Homecage" datasets, which presumably represent a large proportion of use cases for DeepEthogram. Otherwise I think it looks good and I would support publication of this manuscript.

Thank you for the positive feedback and the valuable suggestions that helped us to extend and improve the paper. We have now added more human labels to obtain human-level performance for 8 datasets, including the Mouse-Social and Mouse-Homecage datasets. We have made the datasets and human labels publicly available at the project website, which we hope will assist other researchers.

Reviewer #3:The authors present a software solution (DeepEthogram) that performs supervised machine-learning analysis of behavior directly from raw videos files. DeepEthogram comes with a graphical user interface and performs behavior identification and quantification with high accuracy, requires modest amounts of pre-labeled training data, and demands manageable computational resources. It promises to be a versatile addition to the ever-growing compendium of open-source behavior analysis platforms and presents an interesting alternative to pose-estimation-based approaches for supervised behavior classification, under certain conditions.The authors have generated a large amount of additional data and showcase the power of their approach in a wide variety of datasets including their own data as well as published datasets. DeepEthogram is clearly a powerful tool and the authors do an excellent job describing the advantages and disadvantages of their system and provide a nuanced comparison of point-tracking analyses vs. analyses based on raw videos (pixel data). Also their responses to the reviewers comments are very detailed, thoughtful and clear. The only major issue is that the raw videos used in this work (except the ones added during the revision) are – it appears – not accessible online. This problem must be solved, the videos are essential for reproducibility.

We thank the reviewer for the positive comments and constructive feedback that has greatly helped the paper. We have now made all the videos and human annotations available on the projection website: https://github.com/jbohnslav/deepethogram

A minor caveat is that in order to compare DeepEthogram to existing supervised and unsupervised approaches, the authors have slightly skewed the odds in their favor by picking conditions that benefit their own algorithm. In the comparison with point-tracking data they use a low resolution top-view recording to label the paws of mice (which are obstructed most of the time from this angle). In the comparison with unsupervised clustering, they use the unsupervised approach for an application that it isn't really designed for (performed in response to reviewers requests). But the authors directly address these points in the text, and the comparisons are still valid and interesting and address the reviewers concerns.

We agree with these points. In particular, we agree that the comparison of supervised and unsupervised methods is challenging and not an apples-to-apples comparison, but we included this comparison because other reviewers felt strongly that it was a necessary analysis for benchmarking the model’s performance. We include text that highlights the caveats mentioned by the reviewer and have tried to provide a thorough and balanced assessment of cases in which DeepEthogram might work best and cases in which other approaches may be preferred and will provide additional information. In the head-to-head comparisons, we have highlighted the caveats of the comparisons, including those mentioned by the reviewer. Our goal is for the reader to understand when and when not to use DeepEthogram and what type of information one can obtain from the software’s output. We hope our extensive discussion of these points will help the reader and the community.